# The *Bothriolepis* (Placodermi, Antiarcha) material from the Valentia Slate Formation of the Iveragh Peninsula (middle Givetian, Ireland): Morphology, evolutionary and systematic considerations, phylogenetic and palaeogeographic implications

**Vincent Dupret**[1]*, **Hannah M. Byrne**[1], **Nélia Castro**[2], **Øyvind Hammer**[2], **Kenneth T. Higgs**[3], **Johan A. Long**[4], **Grzegorz Niedźwiedzki**[1], **Martin Qvarnström**[1], **Iwan Stössel**[5], **Per E. Ahlberg**[1]

**1** Department of Organismal Biology, Uppsala University, Uppsala, Sweden, **2** Natural History Museum, University of Oslo, Oslo, Norway, **3** School of Biological, Earth and Environmental Sciences, University College Cork, Cork, Ireland, **4** College of Science and Engineering, Flinders University, Adelaide, South Australia, **5** Department of Earth Sciences, ETH Zürich, Zürich, Switzerland

* vincent.dupret@ebc.uu.se

**Data Availability Statement:** Data is available at https://doi.org/10.57804/8y8t-kr41.

## Abstract

Material of the antiarch placoderm *Bothriolepis* from the middle Givetian of the Valentia Slate Formation in Iveragh Peninsula, Ireland, is described and attributed to a new species, *B. dairbhrensis* sp. nov. A revision of the genus *Bothriolepis* is proposed, and its taxonomic content and previous phylogenetic analyses are reviewed, as well as the validity of morphologic characteristics considered important for the establishment of the genus, such as the shape of the preorbital recess of the neurocranium. A series of computerised phylogenetic analyses was performed, which reveals that our new species is the sister taxon to the Frasnian Scottish form *B. gigantea*. New phylogenetic and biogeographic analyses of the genus *Bothriolepis* together with comparisons between faunal assemblages reveal a first northward dispersal wave from Gondwana to Euramerica at the latest in the mid Givetian. Other Euramerican species of *Bothriolepis* seem to belong to later dispersal waves from Gondwana, non-excluding southward waves from Euramerica. Questions remain open such as the taxonomic validity and stratigraphic constraints for the most ancient forms of *Bothriolepis* in China, and around the highly speciose nature of the genus.

## Introduction

The Valentia Slate Formation is renowned for its Middle Devonian fossil tetrapod tracks, which are the second oldest known record of tetrapod locomotion apart from the ones in Zachełmie, Poland (Givetian vs. Eifelian [1–3]). Little is known of the associated vertebrate body remains from these deposits, apart from the seminal work by Russell [4], who attributed

**Funding:** This work is funded by a Wallenberg Scholarship from the Knut and Alice Wallenberg Foundation and an ERC Advanced Grant (ERC-2020-ADG 101019613 "Tetrapod Origin"), both awarded to P.E.A. Except for K.H. who personally funded the physical preparation of the material in Switzerland, the funders had no role in study design, data collection and analysis, decision to publish, or preparation of the manuscript.

**Competing interests:** The authors have declared that no competing interests exist.

the most abundant bone remains to the antiarch placoderm *Bothriolepis* sp. Russell carefully avoiding a finer taxonomic assignment because of the paucity of identifiable and diagnostic material, and also because of the poor preservation of the material which is often chloritised within low grade metamorphosed rock.

Although the generic attribution of part of the Iveragh material to the genus *Bothriolepis* has never been challenged nor contradicted since Russell's publication, it is noteworthy that this genus is universally accepted as occurring in Euramerican deposits only in younger strata, being confined to the Late Devonian; in other words, *Bothriolepis* is only known in Gondwana and China in pre-Frasnian times [5–7]. The occurrence of *Bothriolepis* in Middle Devonian terranes of the northern supercontinent is therefore a significant discovery.

In this article we describe new specimens along with the original material published by Russell [4]. New specimens were collected by teams from University College Cork and Uppsala University on field trips between 1995 and 2019. The larger quantity of material and access to new technologies such as micro-CT imaging allow a more detailed description of most of the dermal armour, leading to the erection of a new species. This study also presents a new phylogenetic analysis of the genus *Bothriolepis* [8], the taxonomy of which is generally regarded as extremely complicated (see for example [9]:339). A correlation between taxonomic, phylogenetic, palaeobiogeographic and stratigraphic data suggest that a first dispersal wave of the genus *Bothriolepis* occurred from Gondwana to Euramerica as early as the middle Givetian, pushing back this episode by a few million years. Until now, the earliest occurrence of *Bothriolepis* in Euramerica was known in the lower Frasnian of Latvia, Lithuania and Russia [10].

## Material and methods

### Note on the preservation and deformation of the fossils: Taphonomy, diagenesis and metamorphism

The Middle Devonian deposits of the Valentia Slate Formation outcrop predominantly on Valentia Island and parts of the mainland on the Iveragh Peninsula in County Kerry, situated in the southwest of the Republic of Ireland (Fig 1). These deposits consist of fine-grained purple and green siltstones, which are interpreted as fluvial deposits with evidence of soil formation. Intercalated pebbly sandstone-rich layers and several conglomerates are found in-between the siltstones, and these are interpreted as distal spread of gravel from small-radius alluvial fans from the North. Grain size becomes finer southward [1, 11–14]. Several tuff levels allow for a radiometric dating. The strata underwent significant Variscan deformation such as folding, cleavage formation and low-grade metamorphism [1].

The fossil material was subjected to "sub-green schist facies metamorphism" [15, 16], related to Variscan deformation due to the subduction zone related to the dipping of the Rheic Ocean under Euramerica. This metamorphism led to zones of mineralization, as well as elemental exchange between bone and matrix (see below).

Most of the specimens are disarticulated indicating a certain degree of post-mortem transport. Some are better preserved and partly or completely articulated, indicating a quieter depositional environment and/or shorter transportation after the death of these individuals. The overall palaeoenvironment is interpreted as flood plains crossed by channels [1]. Shorter transportation is also correlated with grain size distribution: bigger bone remains are often associated with finer sediment grains, and coarser grain rocks bear smaller and more abraded elements.

Both skull roof (Figs 2–5) and thoracic armour (Figs 6–11) have been found. The fossils are unevenly preserved, sometimes within a single block. Some elements are unmodified, some are flattened, some are "rolled up" on themselves, some are compressed like an accordion. The

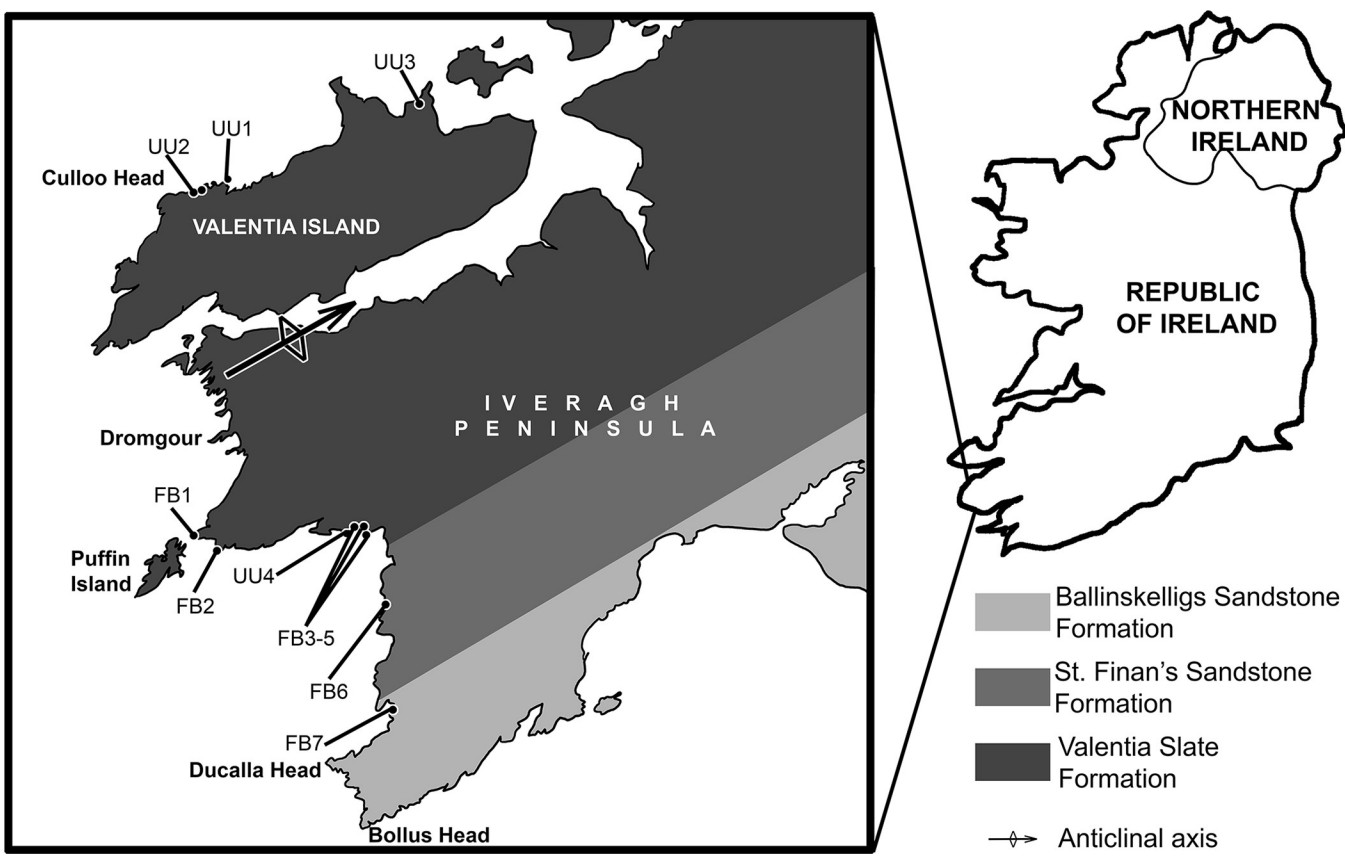

**Fig 1. Geographic context and main geological ensembles of the Iveragh Peninsula, with location of *Bothriolepis* localities.**

best example showing all three kinds of preservation in one single specimen is BMP 59677 (Fig 4). The bone vacuities do not appear compressed in any dominant direction, and thus seem to contradict any compression. It seems obvious that the degree of deformation varies a lot with the orientation of the element in relation to cleavage, often expressed as pressure solution cleavage. This implies that on a small scale, deformation is not entirely uniform. There seems to be a difference in "hardness" (rheological behaviour) of the bone fragments compared to the surrounding slate. The slate might have been more ductile during deformation, resulting in stronger deformation in the slate than in the bones. This might have occurred before the bone was transformed to the chloritic mineral assemblage which it represents today. The lithology, the location of the fossil in the slab, the location of the slab in the tectonic scale (i.e. thrust slice), and possibly the shape of the original element and its "ability" to be rotated or "involuted/rolled up" rather than flattened have influenced the uneven deformation of the fossils. Besides, a variation of the degree of metamorphic mineral reactions, in relation with depth and circulation of fluids, took place and influenced the local chemical situations.

It is impossible to assign separate plates to one individual unless found in articulation, because of the aforementioned tectonic deformations, but also because of the violent winter storms which can displace rocks over great distances in the localities. A few specimens were found *in situ* and can be related to their original taphonomical and sedimentological context.

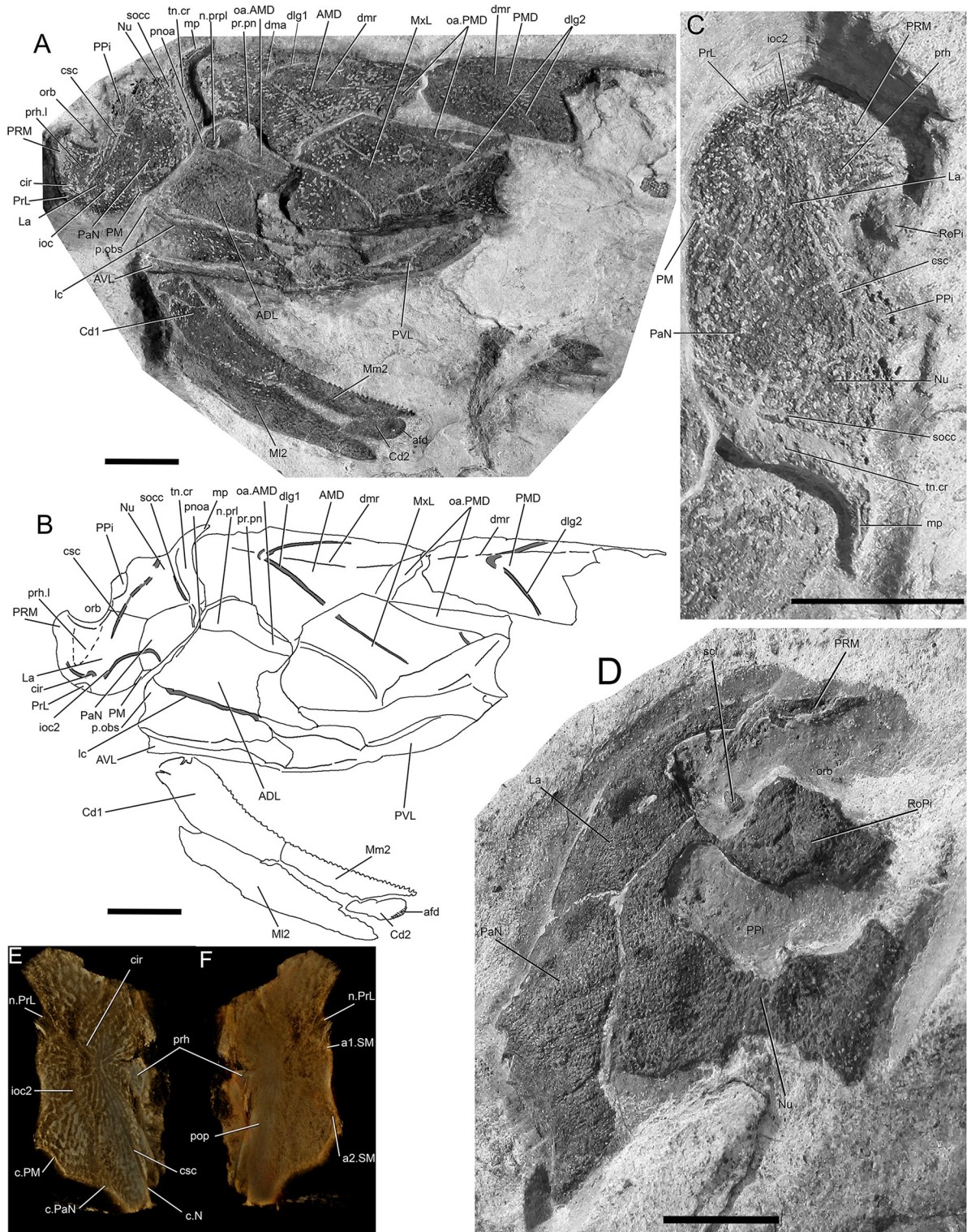

**Fig 2. *Bothriolepis dairbhrensis* sp. nov.** A-C. Subcomplete articulated head and body in left lateral view, NMING:F35201 (A, photograph, B, interpretative drawing, C focus on head, rotated 90° clockwise). D. incomplete skull roof in dorsal view, NMING:F35202. E-F Left lateral plate in external (E) and internal (F) views, NMING:F35203 (image generated with Dristhi 2.6.6). Scale bars 10 mm.

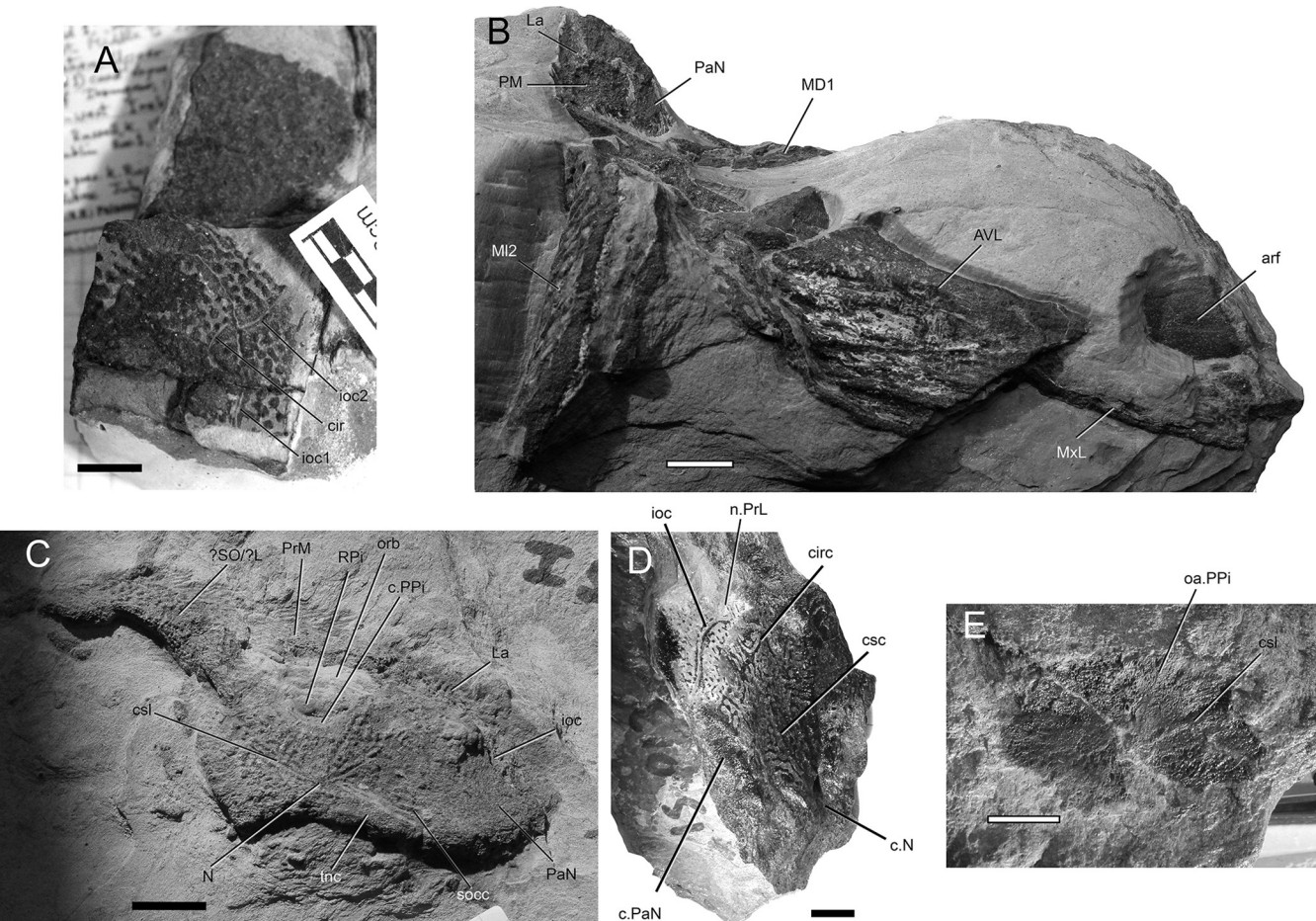

**Fig 3. *Bothriolepis dairbhrensis* sp. nov.** A. Undetermined and incomplete right lateral plate in external views, NHM P 59678. B. Incomplete lateral portions of the skull roof and of the plastron and fin in internal views, NMING:F35204. C. Incomplete skull roof in external view, NMING:F35205. D, left lateral plate in external view, NMING:F35206. E. Incomplete nuchal plate in dorsal view, NMING:F35207. Scale bars 10 mm.

## Anatomical abbreviations

See S1 Table.

## Institutional abbreviations

NHM P: Natural History Museum, London, U.K.; NMING: National Museum of Ireland—Natural History, Dublin, Ireland; TCD: Trinity College Dublin, Republic of Ireland.

## Material physical preparation

Part of the material was mechanically prepared by Christian Obrist who used a microjackhammer, sandblasting (carbonate powder, low pressure) and hand-preparation using gramophone needles.

## Material elemental Scanning Electron Microscopy

A lighter halo is easily noticed around some fossils: it corresponds to element migrations between the fossil bone and the matrix, as evidenced by the elemental analysis performed with

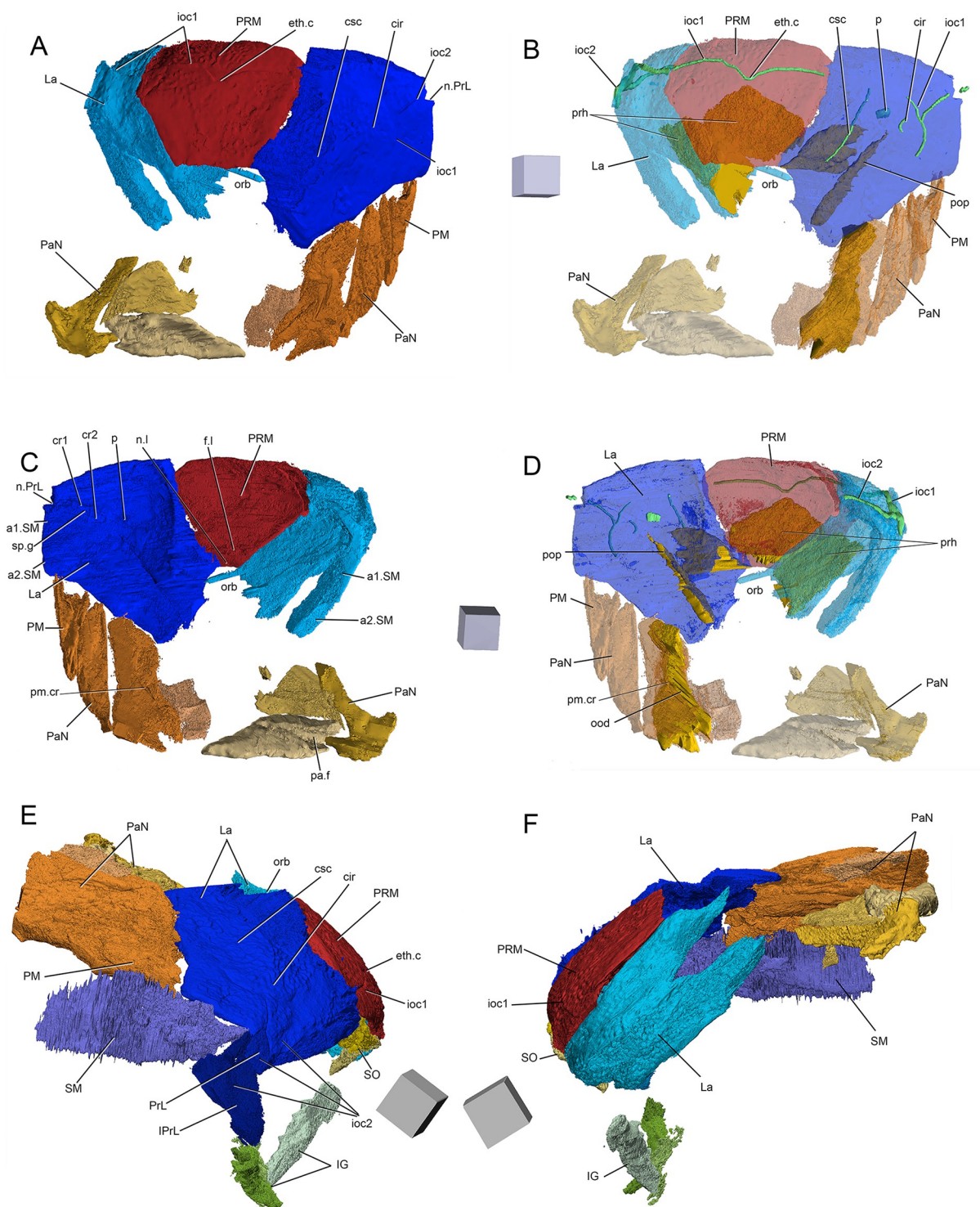

**Fig 4. *Bothriolepis dairbhrensis* sp. nov.** Incomplete head elements of specimen NHMP 59677, restored in anatomical position specimen after CT-scanning and segmentation (images generated with Adobe Acrobat (A-D) and Materialise 3-matic (E-F). A, C. skull roof in dorsal view. D, E. skull roof in ventral view (C and E with dermal plates rendered semi-transparent). E-F. Skull roof in right lateral (E) and left lateral (F) views. Edges of scale cube 10 mm.

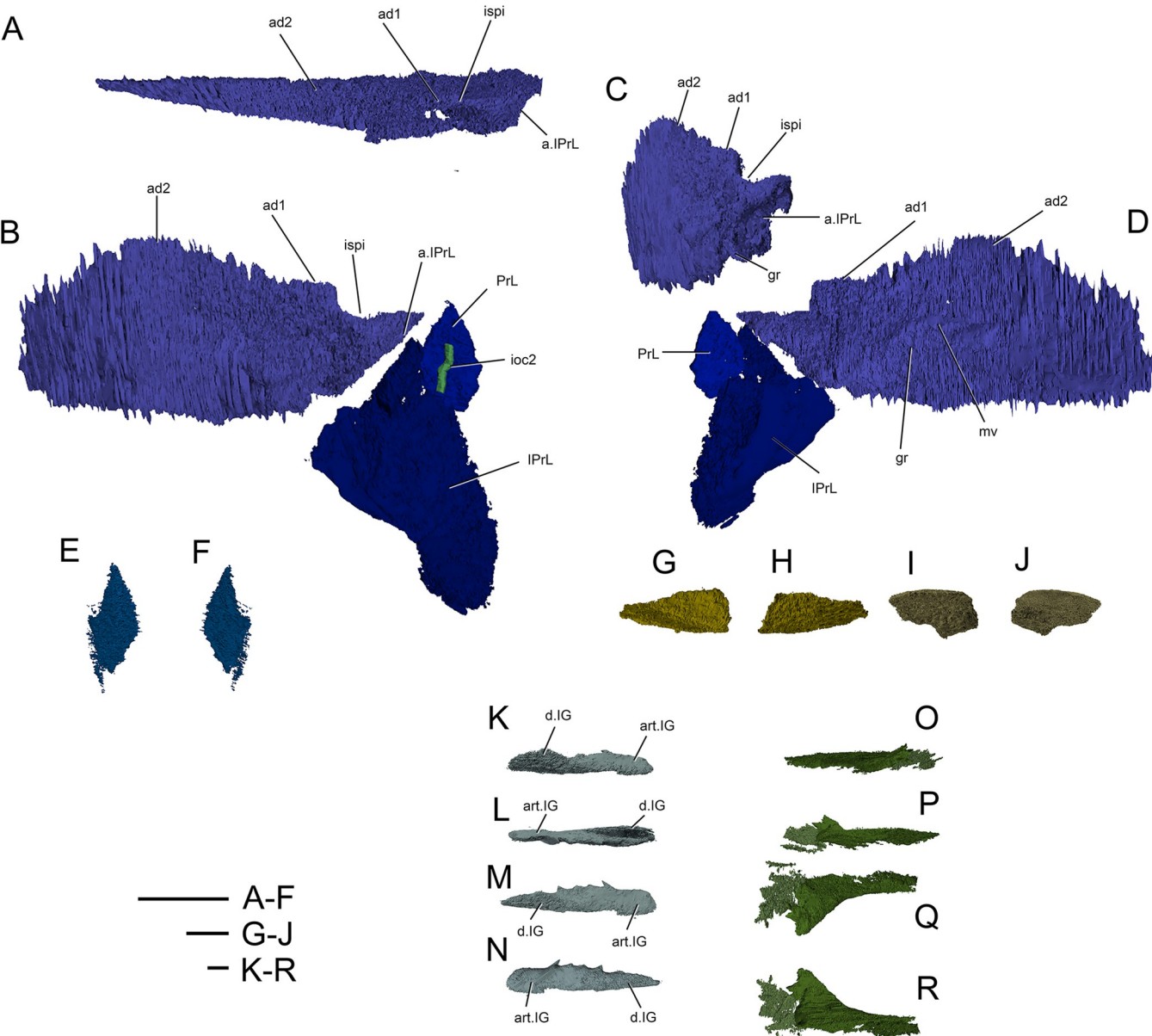

**Fig 5. *Bothriolepis dairbhrensis* sp. nov.** Opercular and mouth elements. A-D. right submarginal plate in dorsal (A) and anterior (C) views, with prelateral and infra-prelateral plates restored in anatomical position in anterolateral (B) and posteromesial (D) views. E-F, left prelateral plates in lateral (E) and mesial (F) views. G-J. Suborbital plates (G, right suborbital in internal view, H in external view; I, right suborbital plate in external view, J in internal view). K-R, infragnathal plates (K, left infragnathal in anterior view, L in posterior view, M in dorsal view, N in ventral view; O, right infragnathal in anterior view, P, in posterior view, Q in dorsal view, R in ventral view). Images generated with Materialise 3-matic, v. 13.0. Scale bars 10 mm.

a Hitachi S–3600N Scanning Electron Microscope (SEM) by Nélia Castro at the Natural History Museum, University of Oslo, Norway. The matrix composed of quartz, chlorite, micas, and feldspar crystals is rich in silicon and aluminium, moderately rich in potassium, and poor in magnesium and iron. The fossil s are rich in iron, and poor in magnesium, potassium, silicon and aluminium. Interestingly, the surrounding halo is richest in magnesium and iron in its median part, but poorer in silicon, aluminium and potassium close to the fossil (S1 Fig in S1 File). This distribution of elements impacts the absorption of X-rays during tomography,

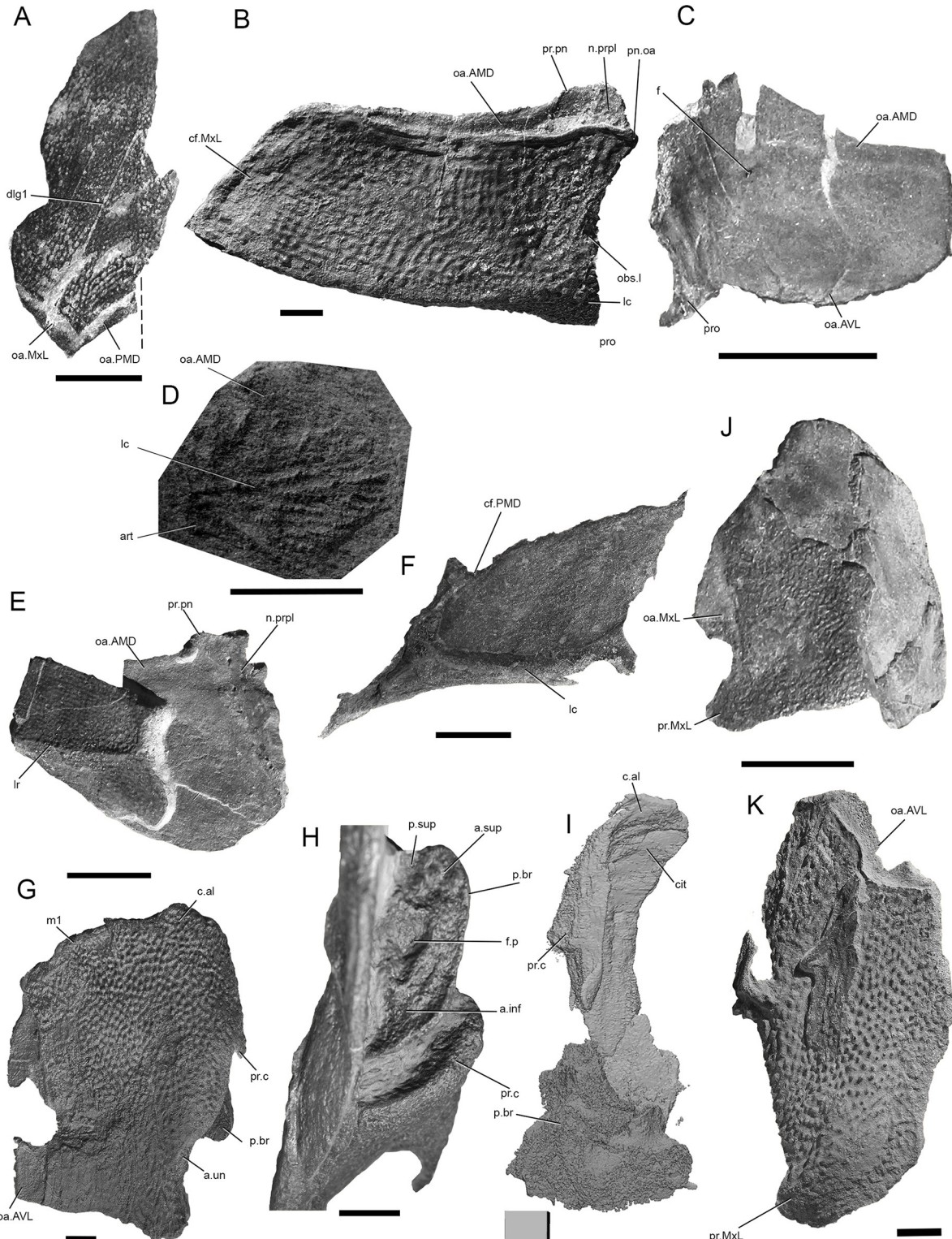

**Fig 6. *Bothriolepis dairbhrensis* sp. nov.** Disarticulated elements of the body armour. A. Anterior median dorsal plate in dorsal view, TCD 19618-FB1xxx (no precise number registered). B. Right anterior dorsolateral plate in external view, NMING:F35208. C. right anterior dorsolateral plate in internal view, NMING:F35209. D. Left anterior dorsolateral plate of a very young individual (NMING:F35210). E. Right anterior dorsolateral plate in external and internal views, NMING:F35211. F. Right mixilateral plate in internal view, NMING:F35212. G-H. Left anterior ventrolateral plate, NMING:F35213 (F in ventral external view, G focus on the brachial process). I. Left anterior ventrolateral

plate in dorsal internal view, NHMP 59677 (digital rendering from Materialise 3-matic). J-K. Left posterior ventrolateral plates in external views (I, NMING:F35214; J, NMING:F35215). Dashed line indicates midline. Scale bars A-C, E-K and edges of scale cube 10 mm, scale bar for D 5 mm.

and of course the resulting tomograms, so consequently segmentation becomes more difficult especially since the distribution is not even. This prevents automated segmentation of the scans by the software (Mimics v. 23).

## Material photography

The material was photographed with a Canon EOS 4000D DSLR Camera with an EF-S 18-55mm f/3.5–5.6 III lens, except the small ADL NMING:F35210 which was photographed with a Nikon DS-Fi1 camera, connected to a Leica MZ9.5 microscope, and with NIS-Elements 4.00.06 image integration software.

## Material X-Ray scanning

The details for each scan are given at the root of each folder holding the digital data.

In London, the material was CT-scanned with a Nikon HMX ST 225 system (Nikon Metrology, Leuven, Belgium) with a tungsten reflection target. Detailed settings files were adjusted for each specimen and are given in S2 Table. Two specimens required stitching performed with an in-house NHM script written in Octave.

In Oslo, the scanning was carried out with a Nikon Metrology XT H 225 ST microfocus CT instrument at the Natural History Museum, University of Oslo, at 160–220 kV, 1 s exposure time, 3016 projections, and with tin filters of varying thickness. Voxel resolution was 75 μm or better. Detailed settings are given in S2 Table.

## Digital data treatment

The contrast between the bone and the sediment is not constant throughout each data set, as exemplified in NHM P 59677. This is due to the migration of some elements from one into the other (see above). Consequently, a sharp segmentation showing ornamentation and/or smooth internal surfaces could not be performed for some of the plates, so only their general shape could be determined. The anatomical identification of some elements was difficult, so relied on the general shape of the element and on the spatial relationships between the elements (especially for the cranial, cheek and mandibular elements), although a high degree of deformation and local transport may have occurred.

Scans were studied using Drishti Import (v. 2.6.4) and Drishti (v. 2.6.4; [17, 18]), or Materialise Mimics (v. 23). In Mimics, each individual structure corresponding to a mask was used to generate a high quality 3D object, itself transformed into an STL file. Each 3D object was duplicated and relocated in the 3D space; once again STLs were generated. The STL files were then imported into Materialise 3-matic (v. 15.0) and were reorganised into anatomical ensembles (head, fin, etc.) in both original and correct positions. Each STL surface was then simplified and corrected using the following functions: Filter Sharp Triangles with filter distance 0.01; Filter Small Edges with filter distance 0.01; Improve Mesh with shape quality high/medium, maximum geometrical error 0.07, and maximal edge length 20.0. A 3D pdf file was finally generated and open in Adobe Acrobat to take pictures used in figures of the current article.

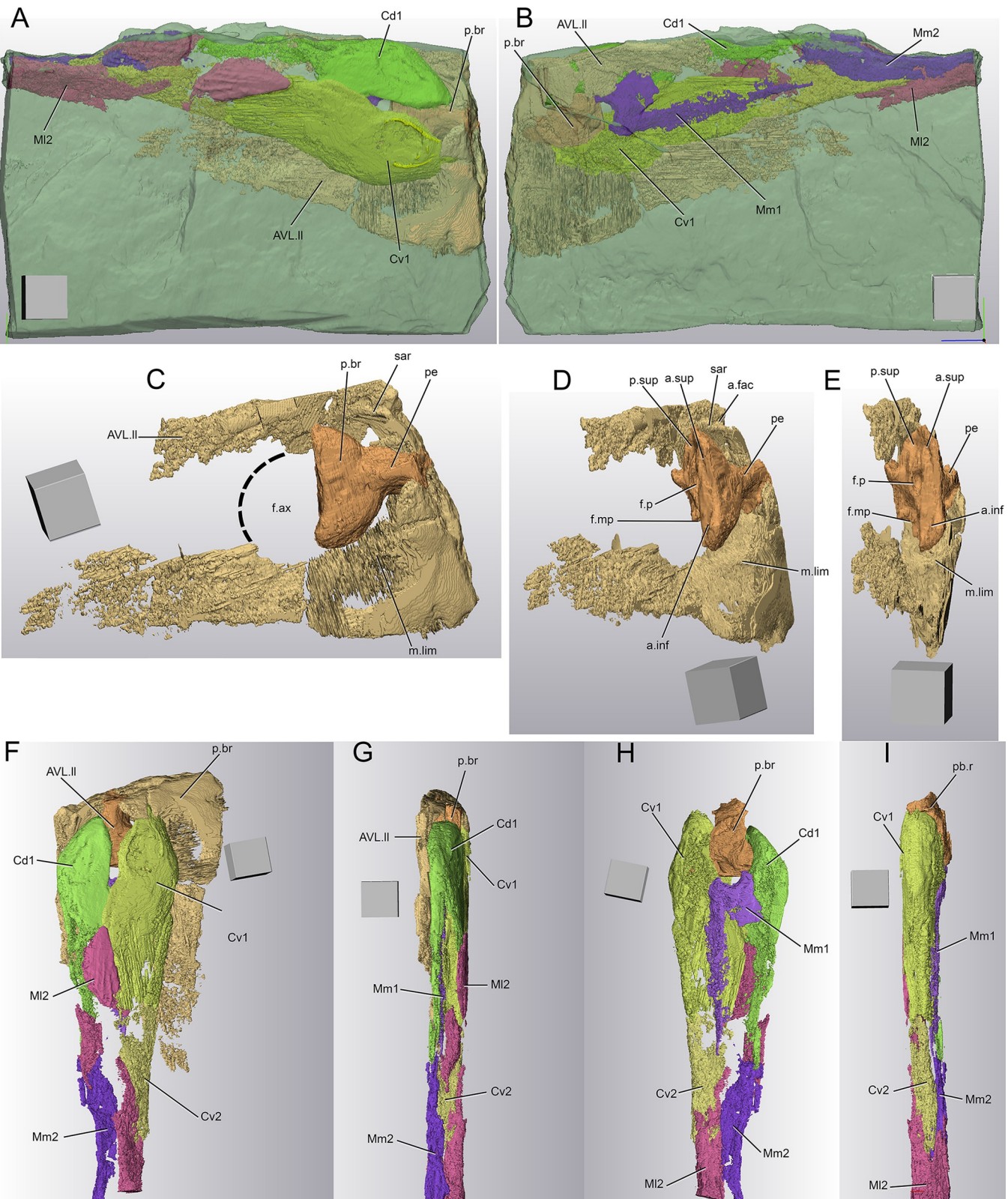

**Fig 7. *Bothriolepis dairbhrensis* sp. nov.** Digital rendering of an incomplete right pectoral fin, NMING:F35216 (images generated with Materialise 3-matic 13.0). A-B. Complete specimen with matrix semi-transparent (A, lateral view; B, median view). C-E. Lateral lamina of the anterior ventrolateral plate and

brachial process (C, lateral view; D, posterolateral view; E, posterior view). F-I. Right proximal segment of the pectoral fin (F, lateral view; G, dorsal view; H, medial view; I, ventral view; F and G with anterior ventrolateral plate). Scale cube edges 10 mm.

## Systematic assignment and its *a priori* justification

The subordinal, familial and generic diagnoses mentioned below are from Long, 1983 (318, ref. [19]). Although almost 40 years old, they still stand (see Discussion section).

The material belongs to the suborder Bothriolepidoidei Miles, 1968 [20], according to the following characteristics diagnosed for the suborder. The postorbital portion of the headshield is shorter than that of the other antiarchs. The postpineal plate is separated from the lateral plates. The anterior median dorsal margin is broad anteriorly (although not broadest at this location; see below). A mixilateral plate replaces the posterior dorsolateral and posterior lateral plates. However, the prelateral plates are present (found in specimen NHMP 59677, identified by the clear indentation in the lateral plates).

The identified material belongs to the Family Bothriolepididae Cope, 1886 [21], according to the following characteristics diagnosed for the family and visible in the incomplete specimen NMING:F35201 (Fig 2A–2C) and to a lesser extent in NMING:F35202 (Fig 2D) and BMP59677 (Figs 4, 5 and 11). The postpineal plate (PPi) is small and does not connect to the lateral plates (La) because of the anterolateral processes of the nuchal plate (Nu), which forms part of the posterior margin of the orbitonasal fenestra (orb). The preorbital recess is developed (prh). The obstantic process of the anterior dorsolateral plate (p.obs, ADL), although not obvious, is robust. Although incomplete and disarticulated, it is doubtless that the pectoral fin extended at least to the posterior edge of the trunk shield (or at least it can be said that it is not as short as in the other families of the Antiarcha). The dorsal central 2 (Cd2) plate is separated from central dorsal plate 1 (Cd1) by the mesial and lateral marginal plates 2 (Mm2 and Ml2).

The material corresponds to the genus *Bothriolepis* Eichwald, 1840 [8], according to the combination of following characteristics diagnosed for the genus and observed in one single specimen NMING:F35201 (Fig 2A–2C). The anterior median dorsal plate (AMD) is broadest at its lateral corners, overlaps the anterior dorsolateral plate and is overlapped by the mixilateral plate (MxL). The mixilateral plate is broadest at its dorsal corner and does not seem to contact the anterior ventrolateral plate (AVL).

## Fossil collecting and curation of new material

New specimens additional to those described by Russell [4] were found during geological and palaeontological field work organised by Cork University and Uppsala University between 1995 and 2019. All this new material (temporary specimen prefixes UU, PSFB, IS) will be deposited in the collections of the Natural History Division of the National Museum of Ireland, Dublin, Republic of Ireland (prefix NMING:F).

## Phylogenetic analyses

**Construction of the data matrix.** Previous phylogenies of the genus *Bothriolepis* were done using selected synapomorphies to denote nodes, rather than composed from computer algorithms (Fig 12; see further). A new taxon*character data matrix was constructed in Mesquite v. 3.61 [22], based on the data published in [10]. The initial set of 20 characters was slightly modified and expanded to 59 characters (see Supplementary Information), and the original range of 12 Euramerican *Bothriolepis* species (+ *Grossilepis* as outgroup) was extended to 54 species, collectively of global distribution. Depending on their completeness or conflicts, some taxa or characters were removed in order to obtain a final data matrix composed of 44

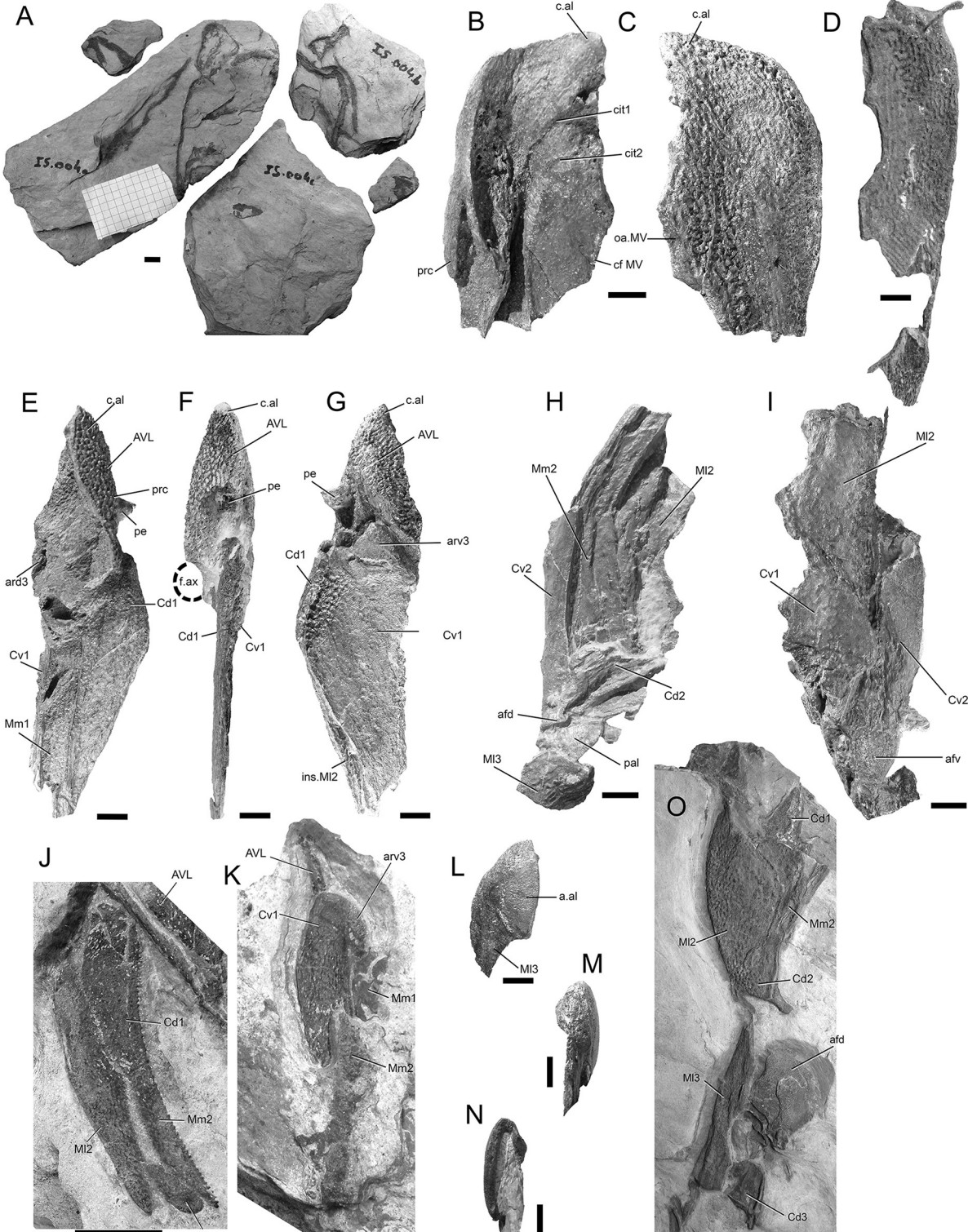

**Fig 8.** *Bothriolepis dairbhrensis* **sp. nov.** A. Elements of the trunk armour and the pectoral fins in dorsal view (NMING:F35217a,b,c,d). B-C. Left anterior ventrolateral plate in internal, NMING:F35218 (dorsal view, B) and in external (ventral view, C). D. Left posterior ventrolateral plate TCD 19618 FB4033 (ventral view). E-G. Incomplete right anterior ventrolateral plate and proximal segment of the fin, NMING:F35219 (E, dorsal view; F, lateral view; G, ventral view). H-I. Incomplete left proximal segment of the fin, NMING:F35220 (H, dorsal view; I, ventral view). J-K. Subcomplete proximal segments of the pectoral fins of specimen NMING:F35201 (J, left fin; K, right fin on

other side of slab). L-N. Articular condyle, NMING:F35221. O. Incomplete proximal and distal segments of the pectoral fin (NMING: F35222). Scale bars 10 mm.

taxa * 50 characters; 43 characters could be coded effectively for *Bothriolepis dairbhrensis* sp. nov. from the Valentia Slate Formation (completeness 73%; see S3 Table). Minor modifications in the coding and the labelling of characters and their states are included in the matrix file and in Supplementary Information). All characters are unordered, unpolarised, and to be of equal weight *a priori*. More characters would be necessary to resolve the relationships between the different coded species of *Bothriolepis*; we must hence keep in mind that polytomies and lower consistency indices (CI) are to be expected (addition of taxa decreases the CI more than the addition of characters in random generated matrices; [23]).

All characters were re-arranged according to a morphological logical sorting in the data matrix. Similarly, all taxa were re-arranged by convenience using the following iterative rules: 1) their outgroup vs. ingroup status, 2) their palaeogeographic origin (Gondwana, China, then Euramerica), and 3) alphabetical order. It should be noted that this sorting is only intended for user convenience, and has no impact on the results (it should be however kept in mind that TNT always uses the first taxon as outgroup).

Once constructed, a survey of the matrix could reveal the completeness of the taxa and of the characters (i.e. the percentage of actually coded information vs. "missing data"; see S3 Table). The following characters and taxa were removed from later phylogenetic analyses because their completeness was lower than 25%:

- Character 5 (0%): ratio length anterior tip / width between lateral corners of the skull roof; this character was difficult to assess regarding the 3D vs. flattened preservation and available reconstructions in literature; besides, the range of values was quite consistent throughout all species, with ratios ranging between 0.6 < L/W < 0.9 (see Supplementary Information).

- Character 7 (24%): bone thickness on orbital edges of premedian (PRM) and lateral plates (in [25]:text-fig. 68, character 27o; in [10]:char 8).

- Character 18 (24%): separate ventrally facing attachment surface for prelateral (PrL) on lateral plate (in [25]:text-fig. 68, character 27m).

- Character 25 (5%): shape of nuchal plate #3 (short or long posterolateral corners; in [25]: text-fig. 68, character 27p; split into three characters in the present analysis).

- Character 34 (15%): anterior submarginal (SM) plate attachment on lateral plate as a transverse ridge (in [25]:text-fig. 68, character 27j).

- Character 35 (7%): anterior portion of submarginal attachment covers spiracular groove (sp. g; in [26]:fig. 10, character 9).

- Character 57 (20%): pectoral pit-line traced on the central ventral plate 1 (Cv1) continuing on the central ventral plate 2 (Cv2; in [10]:character 5).

- Character 58 (2%): dermal ornamentation (in [25]:text-fig. 68, character 27a).

- Character 59 (0%): general size and ornamentation (in [25]:text-fig. 68, character 27q).

- *Bothriolepis obesa* (0%)

- *B. paradoxa* (7%)

- *B. stevensoni* (2%)

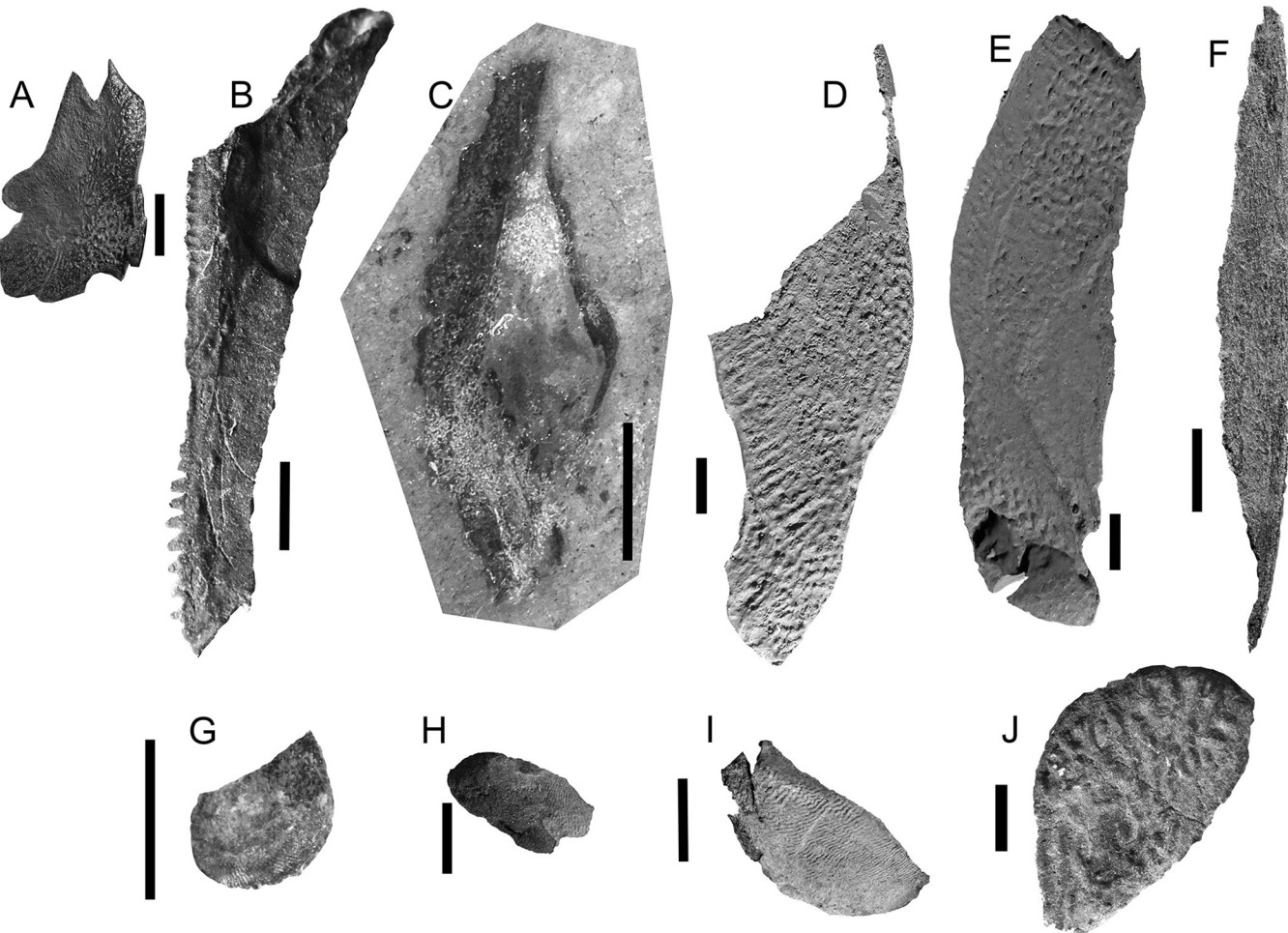

**Fig 9. *Bothriolepis dairbhrensis* sp. nov.** Elements of the proximal segment of the pectoral fin elements. A. Mesial marginal plate 1, NMING:F35230. B. Mesial marginal plate 2, NHMP 59683. C. Mesial marginal plate 2, NMING:F35223. D-E. Mesial lateral plate 2, NMING:F35224 (D in lateral view, E in mesial view). F. Mesial lateral plate 2, NMING:F35225. G. Central dorsal plate 2, TCD19618 FB7052. H. Central dorsal plate 2, NHRMP 69862. I. Central dorsal plate 2, NHRMP 59680. J. Central dorsal plate 2, NHMP 59688 Scale bars 10 mm.

- *B. coloradensis* (17%)

- *B. darbiensis* (24%)

- *B. nielseni* (19%)

- *B. alexi* (5%)

A similar survey revealed the degree of polymorphism coded for each taxon and character; these values can also be seen in the S3 Table. 44 taxa include polymorphic coding (81%); 34 characters include polymorphic coding (58%).

The following taxa are removed from the final analysis because they are considered as junior synonyms of *B. nitida* (see discussion between [27–29]: *B. virginiensis* (63%), *B. coloradensis* (17%), *B. nielseni* (19%).

*Bothriolepis sinensis*, although initially considered, proved to be too poorly preserved and was removed at an early stage of the study (see [30]).

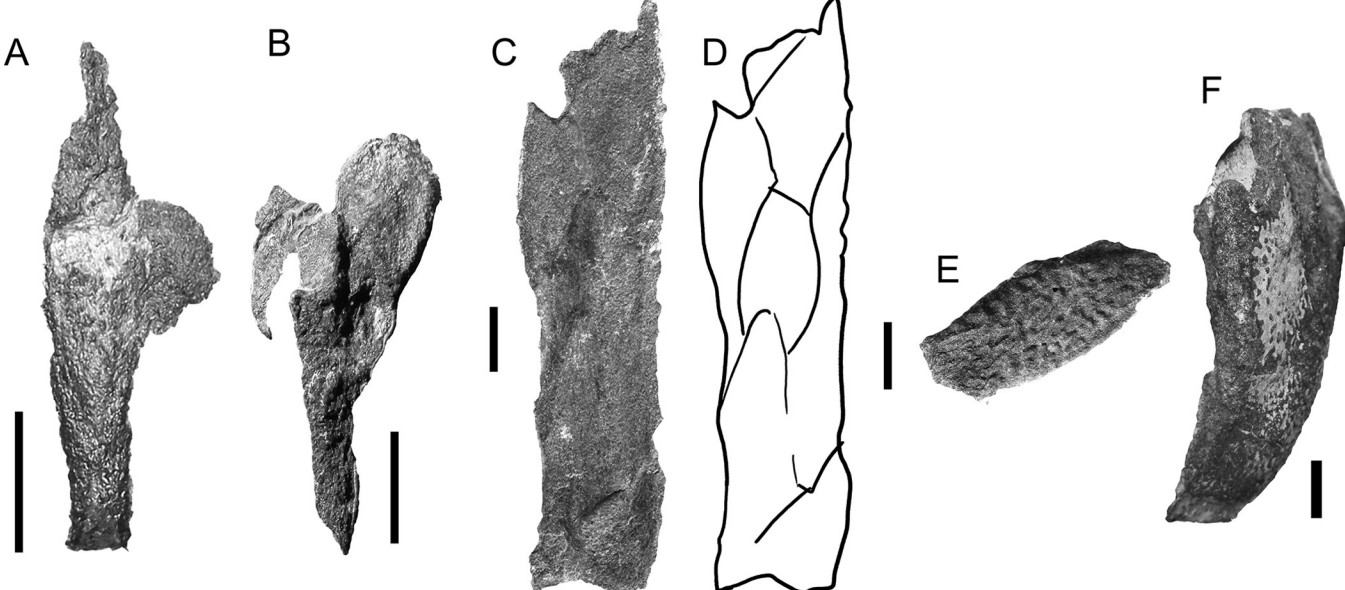

**Fig 10. *Bothriolepis dairbhrensis* sp. nov.** Elements of the distal segment of the pectoral fin elements. A. Central dorsal plate 3, NMING:F35226. B. Central ventral plate 3, NMING:F35227. C-D. Incomplete distal segment of the pectoral fin, NMING:F35228 (D, interpretation drawing). E. Mesial marginal plate 3, NHMP 59689. F. Terminal element of distal segment of pectoral fin (TCD 19618 FB1013). Scale bars 10 mm.

## Selected outgroups for character polarisation

Two antiarchs, *Remigolepis* sp. (Asterolepididae) and *Grossilepis* sp. (Bothriolepididae) were originally coded as outgroups. *Grossilepis* was selected because of its belonging to the same family, as it can also be resolved as sister-group to the genus *Bothriolepis* (see [25]:text-fig. 68A; [31]:fig. 14). *Grossilepis* is often used as such in previously published phylogenies and cladograms, but it should be kept in mind that until now all *Bothriolepis* phylogenies were based on hierarchies of characters assembled manually by the authors, rather than the product of computer-generated algorithms (see Discussion). *Remigolepis* is selected because it belongs to another family and is very well known.

*Remigolepis* and *Grossilepis* were however removed from our final analysis for several reasons. TNT considers only the first taxon as outgroup (i.e. *Remigolepis*) and *Grossilepis* was resolved within the ingroup (S1 File, S9-S14 Figs in S1 File); this result is likely related with the long-branch attraction phenomenon (see Results and Discussion parts; see also [32] for a review of the long-branch attraction phenomenon). *Dianolepis* was also considered as an outgroup, in conjunction with *Remigolepis* and *Grossilepis* or alone, but the results were considered non reliable (see S27-S32 Figs in S1 File). Chinese species of *Bothriolepis* were also considered as outgroups but the results proved inconsistent (see Discussion). The outgroup selected for the analysis described and discussed in details is *Bothriolepis askinae* (see Discussion).

## Treatment of the data matrix

The data were analysed in TNT v. 1.5 [33] using the Zephyr module v. 3.11 [34, 35] in the Mesquite v.3.61 build 927 interface [22]. Index values given further were retrieved from Mesquite. A New Technology search was performed (using the default sectorial search, ratchet, drift and tree fusing options, 1000 random addition sequences, random seed 1), with the most parsimonious trees found ($n_{MPT}$ = 96, $L_{MPT}$ = 210 steps, $CI_{MPT}$ = 0.333, $RI_{MPT}$ = 0.509) then used for a

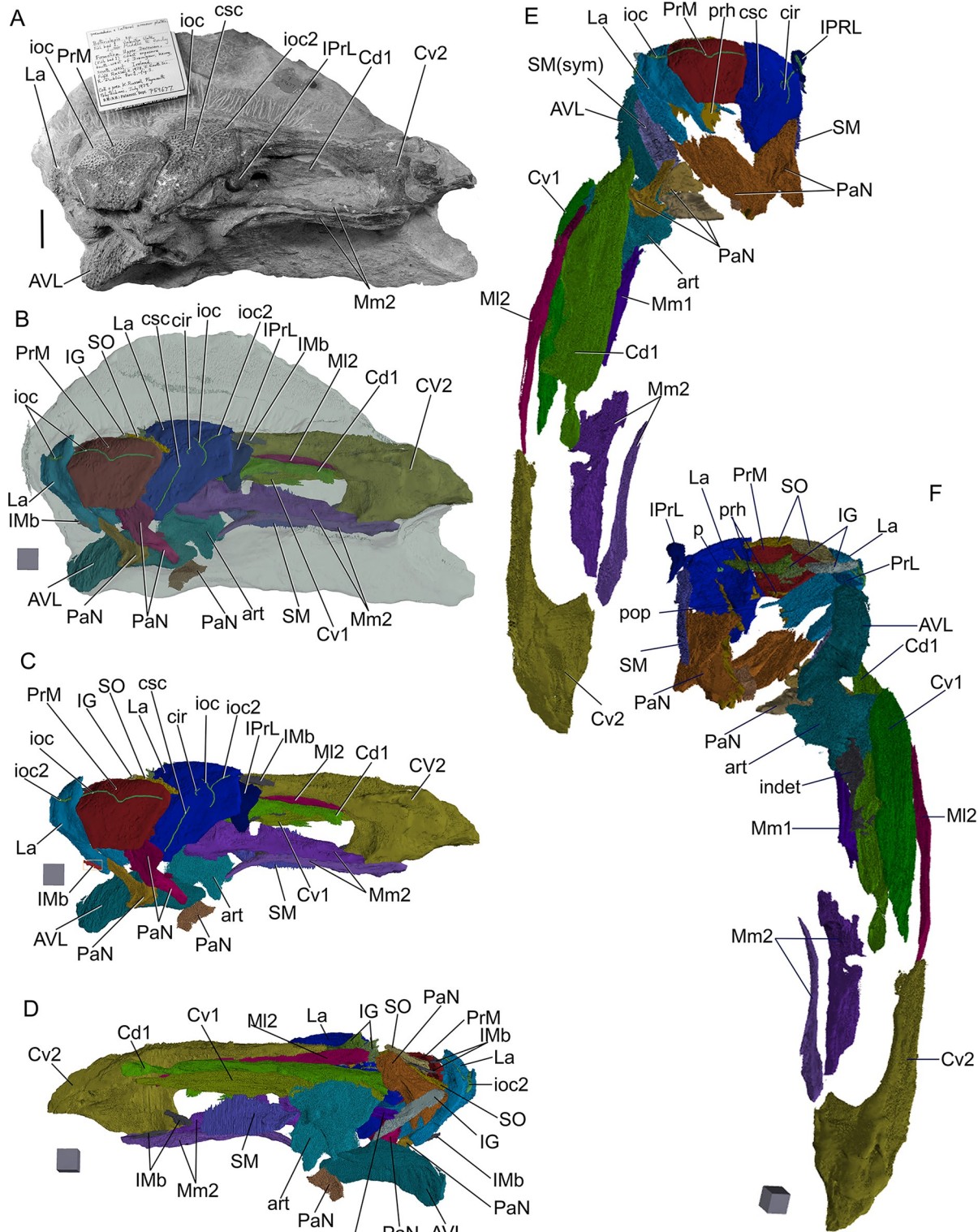

**Fig 11.** *Bothriolepis dairbhrensis* **sp. nov.** NHMP 59677. A. Photograph of specimen. B. Digital specimen with matrix rendered semi-transparent. C-D. Fossil bone material visible in the specimen with matrix rendered transparent (C in frontal view; D in opposite view). E-F. Reconstruction of plates in life position (as much as possible because of deformation) in dorsal (E) and ventral (F) views. Scale bar and edges of scale cube are 10mm.

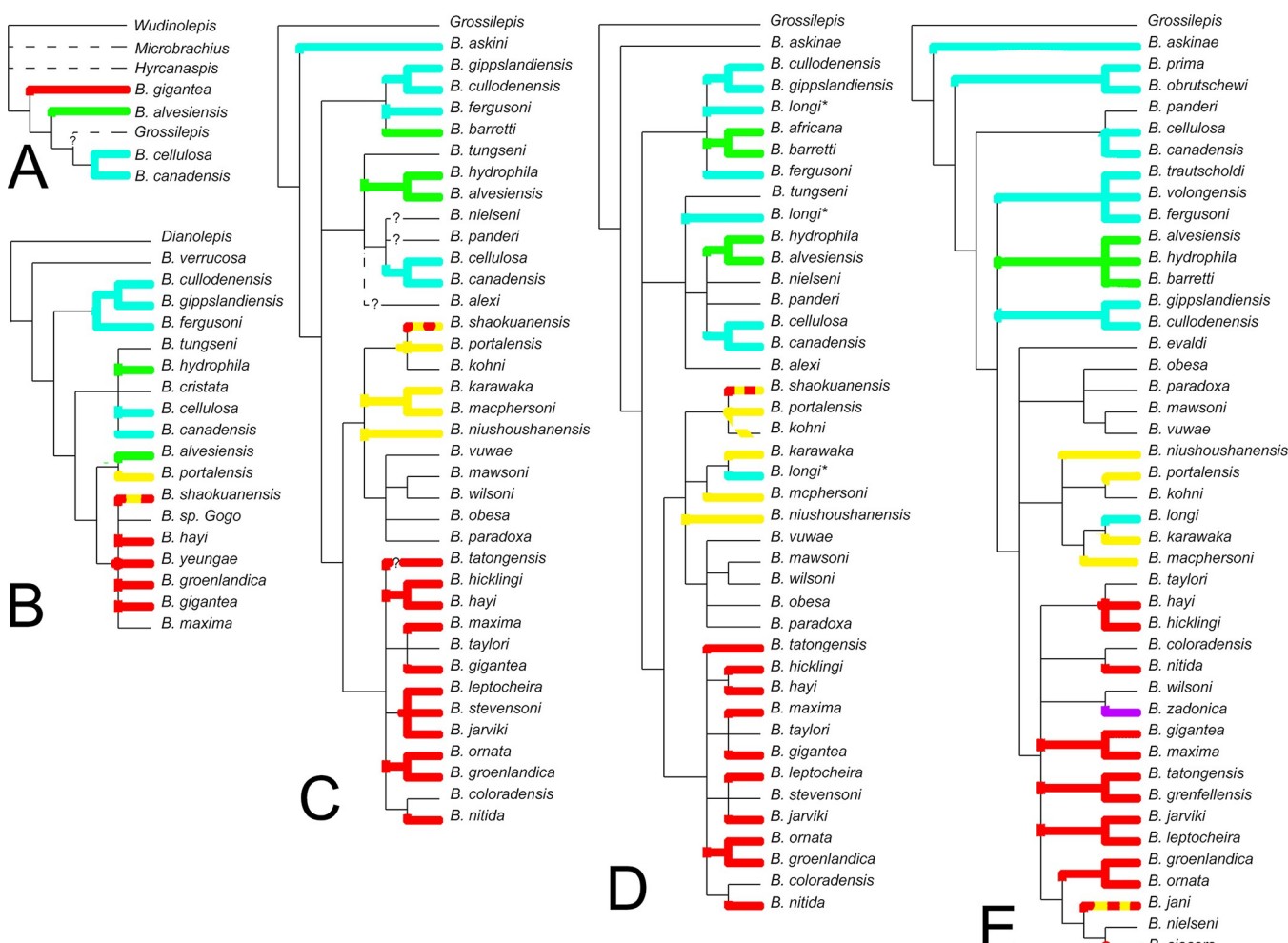

**Fig 12.** Historical overview of the relationships within the genus *Bothriolepis* according to: A. Janvier and Pan ([24]:fig. 12); B. Long ([19]:text-fig. 13); C. Young ([25]:text-fig. 68B); D. Johanson and Young ([26]:fig. 10); and E. Lukševičs ([10]:fig. 83). Colours indicate the shape of the preorbital recess for species in which it is known (see Fig 13).

Traditional Search (tree bisection–reconnection (TBR) algorithm), from which a majority rule consensus tree was calculated (Figs 13–15). The strict consensus tree is $L_{CS}$ = 253 steps long, has a $CI_{CS}$ = 0.277, and a $RI_{CS}$ = 0.358; the majority rule consensus tree is $L_{C50\%}$ = 251 steps long, has a $CI_{C50\%}$ = 0.279, and a $RI_{C50\%}$ = 0.365. No robustness analysis (Bremer nor Bootstrap) was performed. The data matrix, trees (original and consensus) and indices (trees and character) can be obtained with the Mesquite file in Supplementary Information and S4 Table. Some of the obtained trees were imported in PAUP version 4.0a169 [36] in order to access the listed history of character changes.

Further analyses involving the reweighting of the first character shape of the preorbital recess) were performed (using Mesquite, Zephyr and TNT). Indices and trends are shown in the Results section.

## Nomenclatural acts

The electronic edition of this article conforms to the requirements of the amended International Code of Zoological Nomenclature, and hence the new names contained herein are

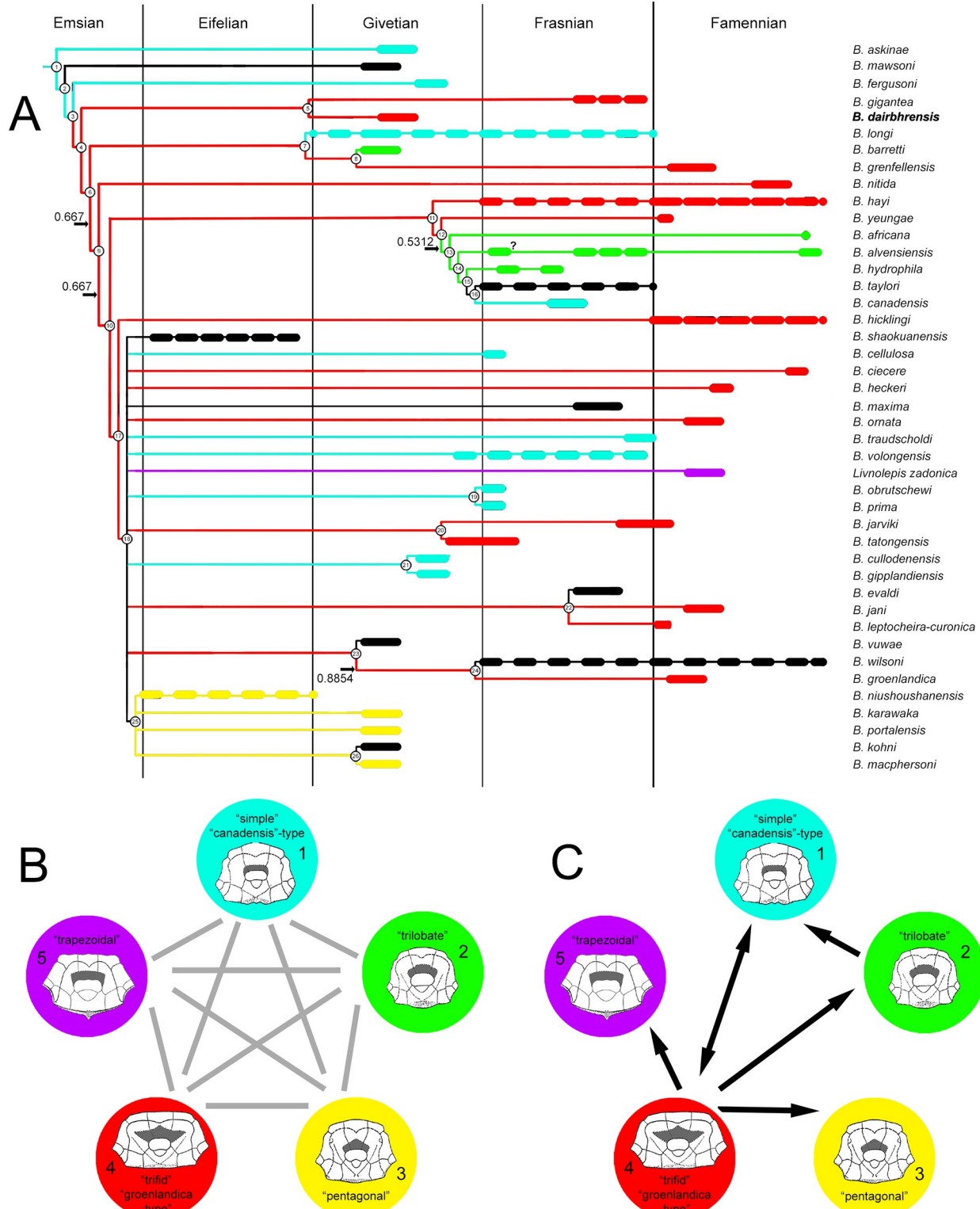

**Fig 13. Phylogenetic analysis of the best known species of the genus *Bothriolepis* [8], with special emphasis on the shape of the preorbital recess.** A. Stratigraphy correlated majority rule (50%) consensus tree (n = 96; $L_{50\%}$ = 251; $CI_{50\%}$ = 0.279; $RI_{50\%}$ = 0.365; indices for polytomies in the strict consensus tree are given on the branches; uncertain ranges indicated with longer dashes). B. *A priori* unordered and unpolarised relationships between the different states. C. Transformation sequence obtained from consensus tree. Colours in cladogram and diagrams correspond to the shape of a preorbital recess: light blue is simple, semi-circular recess ("*canadensis*-type"; outline after *B. askinae* in [25]), green is

trilobate state (outline after *B. barretti* in [25]), yellow is pentagonal state (outline after *B. portalensis* in [25]), red is trifid state ("*groenlandica*-type"; outline after '*B. nitida*' in [25]), purple is trapezoidal state (outline after *B. zadonica* in [37]), black is unknown state. *Bothriolepis* skull roof outlines not at scale.

available under that Code from the electronic edition of this article. This published work and the nomenclatural acts it contains have been registered in ZooBank, the online registration system for the ICZN. The ZooBank LSIDs (Life Science Identifiers) can be resolved and the associated information viewed through any standard web browser by appending the LSID to the prefix http://zoobank.org/. The LSID for this publication is: urn:lsid:zoobank.org:pub:0438-B3D4-AEAA-4000-A1CB-B5DA0ECA466F. The electronic edition of this work was published in a journal with an ISSN, and has been archived and is available from the following digital repositories: LOCKSS [author to insert any additional repositories].

## Systematic palaeontology

Antiarcha Cope, 1885 [39]

 Bothriolepidoidei Miles, 1968 [20]

 Bothriolepididae Cope, 1886 [21]

 *Bothriolepis* Eichwald, 1883 [8]

 *Bothriolepis dairbhrensis* sp. nov. urn:lsid:zoobank.org:act:B577F075-1EF6-48A5-8C7C-E64E36E2BC4E

 *Bothriolepis* sp. in [4]

 Type specimen: NHM P 59677, an adult incomplete skull roof showing the neurocranial preorbital recess and postorbital processes, thoracic armour and proximal segment of the left pectoral fin is proposed as holotype.

 Type locality: FB1 in [4], Puffin Sand Fish Bay, West of Iveragh Peninsula, Ireland

 Type horizon: Valentia Slate Formation, middle Givetian

 *Derivatio nominis*: after "Dairbhre", the Irish name of Valentia Island and Valentia slate Fm. where the fossils come from.

 Diagnosis: *Bothriolepis* species reaching large sizes (estimated approximately up to 50 cm long). Presence of a trifid preorbital recess combined with the following characters: submarginal plate with large elongated ovoid shape posterior and shorter and pointed anteriorly, with a distinct spiracular notch in between. Presence of an articular/contact facet on the anterior edge of the submarginal plate for the infraprelateral plate. Anterior edge of posterior median dorsal plate has angle of between 110˚ and 120˚. Presence of a first oblique sensory line groove on anterior median dorsal plate and of a second on posterior median dorsal plate. Mesial lateral plate 2 with unornamented lateral edge. The lengths ratio [skull roof / proximal segment] is very low and estimated around 0.33 for an adult.

 Remarks. The maximum estimated size mentioned in the diagnosis relies on some big specimens, such as one large posterior ventrolateral plate (PVL) of 14 cm long (TCD19618-FB4033; Fig 8D), or the width of the skull roof of NHMP59677 estimated at 10 cm. The ratio between the length of the skull roof and the proximal segment of the pectoral fin was calculated using NMING:F35201 as an articulated juvenile (ratio = 0.73) and digital reconstruction of the semi articulated adult specimen NHMP 59677 (ratio = 0.33). By comparison, adults of *B. yeungae*, *B. canadensis* have a ratio value of 0.47, *B. fergusoni* of 0.60 or *B. africana* of 0.66 (based on reconstructions in [9, 19, 40, 41]).

 The previously unrecorded geographic and stratigraphic occurrence of *B. dairbhrensis* sp. nov. contributes with its morphological autapomorphies and character combinations to justify the erection of a new species. One can compare the shape of the submarginal plates with that

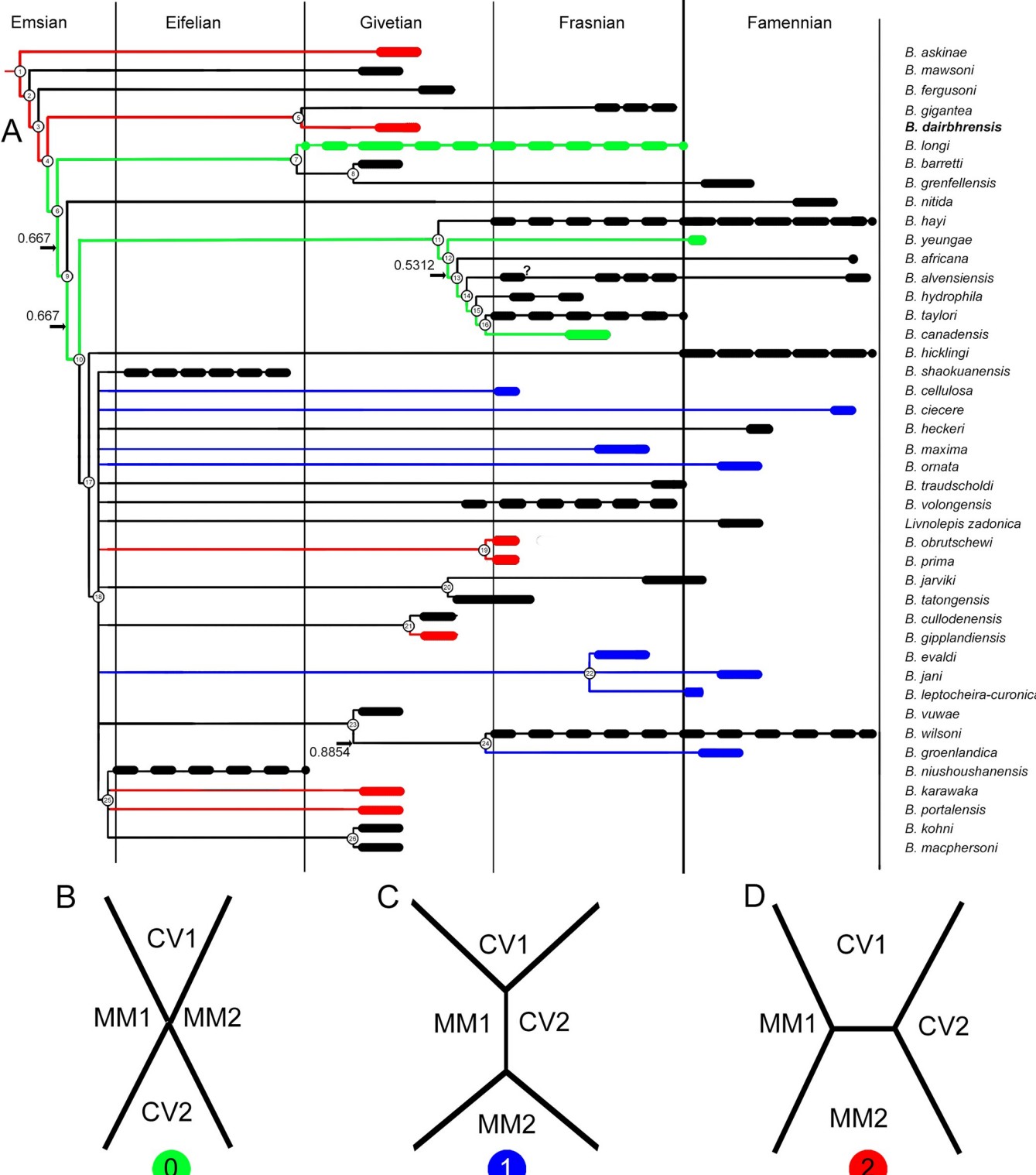

**Fig 14. Phylogenetic analysis of the best known species of the genus *Bothriolepis* [8], with special emphasis on the contact between plates of the proximal segment of the pectoral fin.** A. Stratigraphy correlated majority rule (50%) consensus tree (n = 96; $L_{50\%}$ = 251; $CI_{50\%}$ = 0.279; $RI_{50\%}$ = 0.365; indices for polytomies in the strict consensus tree are given on the branches; uncertain ranges indicated with longer dashes). B. CV1, CV2, MM1 and MM2 connect in one point (state 0, green). C. MM1 and CV2 connect and forbid CV1 and MM2 to contact (state 1, blue). D. CV1 and MM2 connect and forbid MM1 and CV2 to contact (state 2, red).

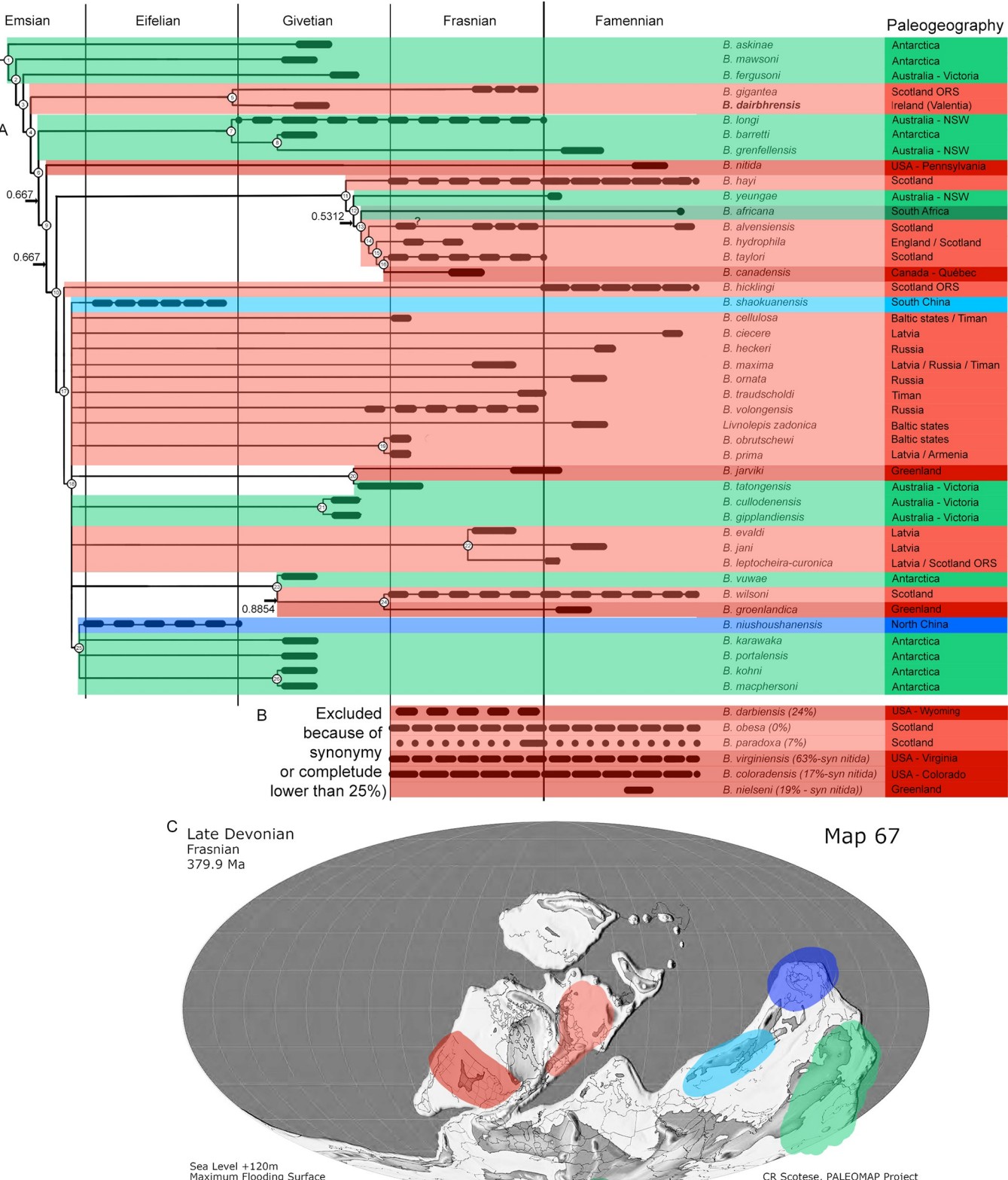

**Fig 15. Phylogenetic analysis of the best known species of the genus Bothriolepis [9], with special emphasis on their palaeogeographic distribution.** A. Stratigraphy correlated majority rule (50%) consensus tree (n = 96; L50% = 251; CI50% = 0.279; RI50% = 0.365; indices for polytomies in the strict consensus tree are given on the branches; uncertain ranges indicated with longer dashes). B. Stratigraphic range and palaeogeographic distribution of taxa not included of

phylogenetic analysis because of incompleteness or synonymy. C. Average distribution of Bothriolepis species of the phylogenetic analysis plot on a late Devonian palaeomap (modified after [38]; oceans in dark grey, shallow and epicontinental seas in light grey, emerged continents in white). Colours in cladogram and palaeomap correspond to the following palaeogeographic zones: Euramerica in red (western part darker than eastern, boundary approximated at Caledonian Mountains), Gondwana in green (western part darker than eastern), Chinese palaeoblocks in blue (darker for Northern, lighter for Southern). For period-wise distribution of the complete Bothriolepididae, please refer to S1 File. ORS: Old Red Sandstone; NSW: New South Wales.

of other taxa (see [9]:fig. 11 for review of plate shapes). The submarginal plate of *B. dairbhrensis* sp. nov. shows a distinctive outline (highly convex dorsal edge just behind the middle of the plate; rather horizontal ventral edge) from that of all other known *Bothriolepis* submarginal plates. It resembles most the Gondwanan forms such as *Bothriolepis* sp. from Gogo [42] and *B. yeungae* [9] with the low anterior process and the notch just posteriorly to it. A clear articulation facet for the prelateral plate seems unique. The angle of the anterior edge of the posterior median dorsal plate is sharper than the one observed in *B. askinae* from Antarctica, and would also suggest a Gondwanan affinity.

## Results

### Description

**Ornamentation.** The ornamentation consists of rounded tubercles that may join up to form coalescent thick ridges in the larger individuals (e.g. ventral lamina of the posterior ventrolateral plate TCD19618-FB4033, Fig 8D). In many specimens the external ornament is worn or poorly expressed due to weathering and preservation (e.g. Fig 2A, 2C and 2D). The smaller specimens, and hence considered younger individuals, display a more nodose or reticulated ornament. This change of ornamentation through ontogeny is common in the genus *Bothriolepis* [9, 43–46].

**Neurocranium.** The shape of the neurocranium in antiarchs is only known from impressions on the visceral surface of the head shield. A portion of the outline of the dorsal surface of the neurocranium is accessible from only one specimen (NHMP59677, Fig 4). It can be restored in its anterior half only with certainty anteriorly and laterally from the orbital fenestra, where it is overlaid by the premedian and lateral plates, and with a lesser degree of confidence in its posterolateral portion.

As described above/below, the infilling of the preorbital recess reveals unambiguously a trifid preorbital recess ("*groenlandica* type" of [20, 25, 43]). The median branch of the preorbital recess is triangular, flat, and is located below the posterior half of the premedian plate. The two lateral branches of the preorbital process are more pointed than the median one; they are flat at their tip but thicker posteromesially at the level of the orbital fenestra. The axis of the right lateral branch of the preorbital recess is directed towards the internal pit (p, Fig 4B and 4C). The angle between the axes of the median and the left branches is 65˚, between the axes of the median and right branches is 69˚. These measured values are considered accurate enough, considering the very different deformation for either lateral plate (inrolled for the left one, flattened for the right one) and very limited deformation for the premedian plate.

The depression for the postorbital process (pop, Fig 4B and 4D) is located under the pocket containing the right lateral branch of the preorbital recess. It is very slender and its tip extends anteriorly as far as the lateral branch of the preorbital recess, and thus more anteriorly than the orbital fenestra. Its axis is directed toward the internal pit.

The infilling of the otico-occipital depression (ood, Fig 4D) of the inner side of the right paranuchal plate reveals a square posterolateral corner. It is bounded laterally by the paramarginal crista (pm.cr, Fig 4C and 4D).

**Skull roof.** The skull or skull elements are preserved in 9 specimens (NMING:F35202, NMING:F35205, NMING:F35206, NMING:F35207, NMING:F35204, NMING:F35201, NHM PV P 59677, NHM PV P 59678, NMING:F35203). It is restored from nearly all elements being preserved.

The premedian plate is trapezoidal, and is larger anteriorly than on its orbital edge (PRM, Figs 2A–2D, 3C and 4). It is widest between the lateral extremities of the sensory line groove. It is best seen on NHM V P 59677 (Fig 4) as it is incomplete and preserved in dorsal view. The ratio (Width max/Width orbital) is 2.0 in NHM V P 59677 (23.62mm*2/23.51mm). The orbital edge is straight in large individuals such as NHM V P 59677, but could be slightly concave in smaller individuals such as NMING:F35201 (in which it is difficult to assess because of the lateral compression of the specimen; Fig 2A–2C). A very shallow pentagonal depression in the midline of the posterior half of the plate corresponds to the underlying preorbital recess. The ethmoid commissure (ethm.c, Fig 2A and 2B) corresponds to a posteriorly directed "pinch" of the infraorbital sensory line groove on the external side, the posterior point of which corresponding to the median anterior apex of the underlying preorbital recess wall. The ventral side of the premedian plate is concave, and clearly reveals the nasal and facial laminae of the preorbital recess in its posterior half (n.l, f.l, prh, Fig 4). The facial lamina is part of the main component of the plate. The left lateral side shows the contact surface with the adjacent lateral plate better than the right side, both on the ornamented part and the nasal and facial laminae. The nasal lamina looks slightly sheared to the left in ventral view compared to the dermal part of the plate. As noted above, the premedian component of the preorbital recess is pentagonal and can be filled digitally in order to reveal its complete shape (see above/further; Fig 4). The preservation of the material and its scan cannot illustrate the presence or not of a rougher area just anterior to the preorbital recess in ventral view.

Only three specimens indicate size and shape of the orbital fenestra, although at different degrees of preservation, and compression orientation (NMING:F35201, NMING:F35202, NHMP59677; orb, Figs 2A–2D and 4). The orbital fenestra appears proportionally larger in the skull roof in younger specimens than older individuals. There is unfortunately no information relative to the size and shape of the suborbital fenestra internally. NMING:F35201 reveals anterolaterally a millimetre difference between the orbital fenestra and the suborbital fenestra (orb, s.orb, Fig 2C).

The rostropineal plate is observed on NMING:F35202 but does not present any obvious nor describable feature (RoPi, Fig 2D).

The postpineal plate is preserved as an incomplete part in specimen NMING:F35201 (PPi, Fig 2A–2C) but its lateral and posterior outline can be seen on NMING:F35202 (Fig 2D). It is slightly concave anteriorly and more strongly convex posteriorly.

The external surface of the lateral plate harbours the main and lateral portions of the infraorbital sensory line groove (ioc1, ioc2, Figs 2–4); a semi-circular groove (cir, Figs 2–4), the concavity of which is oriented anterolaterally, connects mesially to the intersection of the main and lateral branches of the infraorbital sensory line groove. A weakly defined notch on the anterolateral corner of the plate, at the tip of the lateral extension of the infraorbital sensory line groove, indicates the insertion area of the prelateral plate (n.PrL, Fig 4C). The central sensory line (csc, Figs 2–4) groove extends anterolaterally between the contact with the nuchal plate posteriorly and the semi-circular sensory line groove anteriorly but without reaching it. In the largest individual the central sensory line groove is located closer to the triple point between the lateral, nuchal and paranuchal plates (NHMP 59677, Fig 4), whereas in a smaller individual it is located closer to the orbital fenestra (NMING:F35201; Fig 2A–2C), unless the latter would be related with a compression of the specimen on its lateral side (i.e. median part closer to orbit shortened). These grooves are best visible on the right lateral plate of NHM P

59677 (La, Fig 4). A triangular area lower than the rest of the headshield on the lateral plate of NHMP 59677 indicates the position and shape of the underlying lateral branch of the preorbital recess (Figs 4A and 11A). The internal surface of the lateral plate reveals the flat lateral branches of the preorbital recess wrapped by the facial and nasal laminae. The postorbital process is situated laterally to the lateral side of the nasal lamina. An internal pit of moderate size is visible anteriorly to the tip of the lateral nasal lamina, as well as the oblique spiracular groove bordered by the anterior and posterior crests anterolaterally to the ensemble (sp.g, cr1, cr2, Fig 4C). The lateral edges of the lateral plates of NHMP59677 reveal in ventral view the two attachment areas for the submarginal plate (a1.SM, a2.SM, Fig 4C). The anterior insertion area is short, round and located just posteriorly to the notch for the prelateral plate (n.PrL, Fig 4C); the second one is located a bit posteriorly from the first one, and is elongated anteroposteriorly. Specimen NMING:F35203, although incomplete, shows contact surfaces for the adjacent nuchal and paranuchal plates (c.N, c.PaN, c.PM, Fig 2E and 2F). This specimen thus shows that the postmarginal plate contacts both the lateral and paranuchal plates. The internal side of the same specimen shows a paired attachment area located behind the notch for the prelateral plate, in a pattern similar to that observed in NHM P 596777.

The nuchal plate is observed on NMING:F35201 in lateral view and in NMING:F35202 for the anterior half in dorsal view (Fig 2A–2D). It produces a pair of anterolateral processes avoiding a contact between the lateral and postpineal plates. The posterior edge appears thicker because of the transverse nuchal crista of the internal side, laterally continued on the paranuchal plate. Its shows a well pronounced median process (mp, Fig 2A and 2B). Just anteriorly to the transverse nuchal crista thickening a transverse supraoccipital sensory line groove can be distinguished, also continued on the paranuchal plate (tn.cr, socc, Figs 2A–2C and 3C). An unidentified plate lying just posteriorly to the right lateral plate of NHMP 59677 could be a portion of the nuchal plate. A more complete specimen (NMING:F35205; Fig 3C) reveals a clear concave contact edge for the postpineal plate, a central sensory line, a posterior supraoccipital sensory line, and a thick transverse nuchal crest continued on the right paranuchal plate, although very weathered; the posterior median crest is damaged and not preserved.

The paranuchal plate is visible in lateral view on NMING:F35201 and in dorsal view on NMING:F35202 (Fig 2A–2D). Its posterior edge harbours mesially the continuation of the supraoccipital sensory line groove from the nuchal plate. It is crossed anteroposteriorly by the infraorbital sensory line groove, which shows a short and sharp posterolateral curve in the posterolateral corner of the plate. Two very damaged elements are identified as possible portions of the left and right paranuchal plates on NHMP 59677. The remains of the left would correspond to the lateral and posterolateral edges of the plate showing the dermal articulation area with the anterior dorsolateral plate. The right element is bigger, displaced under the premedian plate, and folded in a similar way that what occurred to the left lateral plate; the element can be identified as a paranuchal plate because of the otico-occipital depression and pocket for the posterolateral edge of the neurocranium (ood, Fig 4D). The supraoccipital thickening is not preserved. When repositioned next to the right lateral plate, the contact edges fit somehow in our model. The most anterior part of this element may belong to another plate, possibly the submarginal plate if those elements were really connected. A very poorly preserved element can be interpreted as a left paranuchal plate and can be observed in NHMP 59677; it was broken in two parts; the lateral edge of the plate remained connected to the lateral plate, while the more mesial part was folded and rotated next to the previous element. Reconstructed, the dorsal side shows a shallow and wide groove for a supraoccipital sensory line. The lateral edge shows a slender overlap area for the anterior dorsolateral plate. Ventrally, a transverse depression can be interpreted as the para-articular fossa (pa.f, Fig 4C), or as a simple break. NHMP 59677 also shows a right paranuchal plate, displaced under the premedian and left lateral plates

and folded on itself, with its ventral otico-occipital lamina underlying the otico-occipital process, which is broken and displaced. Once digitally unfolded, it accommodates the lateral and submarginal plates. The posteromesial edge of the mesial lamina shows indentations to fit the nuchal plate.

The postmarginal plate is partly visible in NMING:F35201, still attached to the skull roof, and maybe as a detached plate in NHMP69677 (PM; Fig 2A–2C), but its outline remains uncertain. It seems to be a small quadrangular element (the latter corner is missing in NMING:F35201). In NMING:F35201 it is aligned with the orbital fenestra. In the larger individual NHMP 59677, it overlies the right paranuchal plate but is very compressed and appears triangular; it however contacts the lateral and paranuchal plates, and its ventral side accommodates the posterodorsal edge of the submarginal plate.

**Elements of the cheek.** The prelateral plate (PrL, Figs 2A–2C and 5B–5F) is a small tear drop-shaped element, the top of which inserts in the notch of the lateral plate. It is known only in NMING:F35201 and NHM P 59677, in which only the right one shows a shallow groove for the second branch of the infraorbital sensory line.

The infraprelateral plate is known only in NHM P 59677 (IPrL, Fig 5B and 5D). Its external surface shows a very shallow and curved groove for the second branch of the infraorbital sensory line (ioc2, Fig 5B). Its outline is uncertain, but appears more similar to the rounder homologous element in a *Bothriolepis* sp. from Gogo [25, 42] than to the quadrangular one of *B. canadensis* [43, 47]. The posterodorsal edge of the plate is straight and accommodates the anterior edge of the submarginal plate. The posteroventral edge of the infrapreorbital plate is sigmoid and presents an overlap area to accommodate the anterior edge of the anterior ventrolateral plate. The anterior edge is straight, and the ventral one is rounded. A faint groove interpreted as the second infraorbital sensory line groove can be distinguished on the external surface of the plate (ioc2, Fig 5B); vertical dorsally, it bends anteriorly in the lower third of the plate. A dorsal process of the infraprelateral plate penetrates into a space between the prelateral and submarginal plate; this pattern is unknown in other members of the genus *Bothriolepis*, but we have to keep in mind that the cheek elements are seldom preserved.

The submarginal plate is found only in NHM P 569677 (SM, Fig 5A–5D). Although the ornamentation cannot be observed because of the preservation state, its outline, articular surfaces are clear and indicate that it is a right submarginal plate. It is oblong in shape, and slightly higher posteriorly than anteriorly. The dorsal edge of the plate is straight, at the exception of a small and rounded notch between the anterior and posterior articulation areas with the lateral plate, corresponding to the infraspiracular incisure (ad1, ad2, ispi, Fig 5A–5D). The anterior facet ad1 articulates on the lateral plate posteriorly to the spiracular groove, somewhat resembling the condition in *B. mcphersoni* in which the articulation area a1.SM is located in the spiracular groove rather than anteriorly to it as in most *Bothriolepis* species. Unlike *B. macphersoni*, the articulation area is oriented longitudinally along the edge of the lateral plate rather than transversally and more mesially (see [25]:text-fig. 66C). A longitudinal groove can be seen under the posterior lateral articular facet on the anterior part of the external side of the plate, shorter to what is observed in Young ([25]:text.-fig. 55). The posterodorsal edge is slightly curved and accommodates the postmarginal plate of the skull roof. The anterior edge of the plate shows an articular facet to accommodate the infraprelateral plate (a.IPrL, Fig 5A–5C). Mesial to this articular facet lies a groove extending ventrally and posteriorly on the internal side of the plate, and underlying a mesial vault (gr, m.v, Fig 5C and 5D). This vault tapers posteriorly into a low thickening.

**Jaw apparatus.** The suborbital plates (SO, Fig 5G–5J) are subquadrangular elements with rounded corners, wider than high. The symphysial edge is much shorter than the lateral one. They are strongly tuberculated, and do not show any sensory line groove. Both left and right

plates are preserved in NHMP 59677, although the ventral masticatory blades are not preserved. It is however noteworthy that the ventral edge of the internal side of the left suborbital plate is smooth, and could indicate a very low and abraded masticatory blade, advocating for a very old individual, comforted by the large size of this specimen.

The infragnathal plates are elongated (IG, Fig 5K–5R). The plates are naturally folded so that the masticatory median and articular lateral flanges are not in the same plane (d.IG, art. IG, Fig 5K–5N). The masticatory flange is tuberculated externally; the rest of the plate is smooth. Neither plate is complete.

Six elements are encountered in the material, next to the skull. They could be considered as hyomandibular and ceratohyals, but as their shape is undiagnostic they could be other elements as well. They can be seen on the 3D pdf available as supplementary material.

**Sensory lines.** Their description is given above in the relevant dermal plates. Despite being considered non diagnostic within the genus *Bothriolepis* [9], we use some characters related to their course (see phylogenetic analysis sections), keeping in mind the possibility of extreme intraspecific or even intra-population variation (see for example [48]).

**Trunkshield.** The anterior median dorsal plate is observed on two specimens (NMING: F35201, TCD 19681 FB1xxx; AMD, Figs 2A, 2B and 6A). It is visible in dorsal view as indicated by the posterior extension of the oblique sensory line groove as a vertical line, but the right lateral half is very compressed in NMING:F35201 (approximately 33% shorter). This is due to the lateral main axis of compression of the specimen during fossilisation. The ratio L/W must be slightly greater than 1.3 (width value calculated as the double of the measured left unaltered side of the plates in NMING:F35201 (ratio equals 1.35) and TCD 19681 FB1xxx (ratio equals 1.4)). The anterior edge of the anterior median dorsal plate that articulated to the skull roof is hidden under the posterior edge of the skull roof. The plate overlaps anteroventrally the anterior dorsolateral plate and is overlapped posteroventrally by the mixilateral plate and posteriorly by the posterior median dorsal plate. The posterior edge of the ornamented part of the plate is very clear as it displays a step pattern, and is followed by the overlap area for the posterior median dorsal plate (oa.PMD, Fig 6A). This posterior edge must have been a very wide angle open posteriorly, greater than the measured 145˚ value. The posterior oblique dorsal sensory line groove on the anterior median dorsal plate is an inverted V-shaped structure (dlg1, Figs 2A, 2B and 6A). Its anterior tip is located at the first third of the length of the plate, which may correspond to the tergal angle (dma, Fig 2A). The groove extends posterolaterally onto the mixilateral plate. A low median ridge is observed in the second third of the plate, between the tergal angle and the level of the articulation of the mixilateral plate (dmr, Fig 2A and 2B). The internal side of the anterior median dorsal plate is not available.

The posterior median dorsal plate is observed on one specimen (NMING:F35201, PMD, Fig 2A and 2B). In NMING:F35201, it is visible in left lateral view and although the right half is very compressed, a portion of the right mesial lamina is visible. It overlaps both the anterior median dorsal and the mixilateral plate. The measured angle of the anterior edge is of 89˚, and obviously does not fit the value of the posterior edge of the anterior median dorsal plate; this discrepancy is related to its compression during the fossilisation process. Because the anterior edge is not very damaged, we are confident in its completeness. A calculated angle taking into account only a lateral compression gives a value of 117˚ ([145+89]/2). In the anterior median dorsal plate TCD 19681 FB1xxx, the value of the half angle between the midline and the posterior edge of the plate is 55˚; the total value of the angle for the overlap area for the posterior median dorsal plate (and thus the angle of its anterior edge) is 110˚. The posterior edge of the plate is broken, but the right lateral edge is as long as the anterior median dorsal plate. A low posterior median ridge extends along the second third of the plate (dmr, Fig 2A and 2B). A

second oblique dorsal sensory line groove, inverted V shaped is observed and continues onto the mixilateral plate (dlg2, Fig 2A and 2B).

The anterior dorsolateral plate (ADL, Figs 2A, 2B and 6B–6E), is known in two specimens: the subarticulated NMING:F35201 and NMING:F35208. It is roughly quadrangular and longer than high. One specimen shows a clear lateral ridge (lr, Fig 6E). The plate is overlapped mesially by the anterior median dorsal plate (oa.AMD, Figs 2A, 2B and 6B–6E) and overlaps posteriorly the mixilateral plate (cf.MxL, Fig 6B). The overlap area for the anterior median dorsal plate is wider anteriorly, is located behind the low postnuchal ornamented corner of the plate and extends between the notch for the external postlevator process and the internal postlevator process of the anterior dorsolateral plate (pn.oa, n.prpl, pr.pn, Fig 6B). A low and smooth lip is visible at the boundary between the ornamented and unornamented areas. The dorsal lamina is much wider than the lateral one is high; this is more marked in adult specimens than in younger ones. The obstantic process is not preserved in its entirety as only a short buttress is visible on NMING:F35201 (p.obs, Fig 2A and 2B). However, the articulated specimen NMING:F35201 shows that the dermal articulation occurred with the postmarginal and the paranuchal plate. A very small anterior dorsolateral plate most likely belonging to a very young individual was discovered in the material (NMING:F35210, Fig 6D). Its ornamentation consists of straight tuberculated ridges, oblique against the sensory line groove.

The mixilateral plate (MxL), is visible in two specimens: in external view with the semi articulated NMING:F35201 (Fig 2A and 2B) and in internal view in NMING:F35212 (Fig 6F). It is broken in two parts: the posterolateral portion broke off and overlaps slightly the dorsal lamina posteriorly. The dorsal lamina is overlapped by the anterior dorsolateral plate anteriorly and by the posterior median dorsal plate dorsoposteriorly, and overlaps the anterior median dorsal plate anterodorsally (oa.ADL, oa.PMD, Figs 2A, 2B and 6F); this overlapping pattern is observed in all *Bothriolepis* species. The overlap area for the posterior median dorsal plate is slightly higher in its posterior end. The mesial edge of the ornamented dorsal lamina is straight at the exception of the posterior end indented by the posterior median dorsal plate. Similarly, the lateral edge of the ornamented area of the lateral lamina is straight except near the posterior end where the posterior ventrolateral plate indents the mixilateral plate dorsally. The posterior edge of the lateral lamina is concave. The dorsal lamina is crossed in its anterior half by the posterior oblique dorsal sensory line groove continued from the anterior median dorsal plate (dlg1, Figs 2A, 2B and 6A), and in its most posterolateral part by the second posterior oblique dorsal sensory line groove continued from the posterior median dorsal plate; this feature is present in all *Bothriolepis* species. The main lateral sensory line groove cannot be recognised due to the damage of the lateral lamina in its anterior part.

The anterior ventrolateral plate (AVL, Figs 6G–6I, 7A–7G, 8B and 8C) is known from specimens NMING:F35213, NMING:F35218, NMING:F35219, IS073, IS075, NHMP 596777 and NMING:F35216 (although only digitally for the two latter ones). It is known from both internal and external surfaces with both the ventral and lateral laminae preserved. The lateral lamina is highest above the pectoral fin attachment area. The ventral lamina of NMING:F35213 is wide and displays tubercles anteriorly and anastomosed ridges posteriorly. It overlaps posteriorly the posterior ventrolateral plate (oa.AVL, Fig 6K). The ventral lamina extends anteriorly under the head as a subcephalic division, overlapped by the submarginal plate, visible in NHMP59677 (c.al, Fig 6G and 6I). The prepectoral corner (pr.c, Fig 6G) is well marked and protects the brachial process. The overlap area for the semilunar and the other anterior ventrolateral plate are visible in NMING:F35213 (m1, oa.AVL, Fig 6G and 6K). There is a small unornamented area on the ventral lamina of the plate immediately posteriorly at the articular fossa, as observed in Antarctica forms (a.un, Fig 6G; see [25]:23). The diameter of the axillary foramen is estimated as wide as the length of the brachial process (Fig 7C). A wide and low *crista*

*transversalis interna anterior* is observed on the internal side of NHMP 59677 (cit, Fig 6I). It looks more like a thickening than a crest, and can be related to either the poor preservation or the adulthood of the specimen, or both.

The brachial process is large (in some individuals it can be 24 mm in diameter as in modelled NMING:F35216; NHMP 59677; p.br, Figs 6G–6I and 7). It is rectangular rounded at its corners (although the rectangular aspect is largely due to compression during fossilisation) and pierced by a large funnel pit (f.p, Figs 6H and 7D–7E). The process faces posteriorly. The anterior dorsal, anterior ventral and posterior dorsal muscle insertions are located above and below the funnel pit respectively (a.sup, a.inf, p.sup, Fig 6H, 6D and 6E). The *pars pedalis* is short and extends horizontally (pe, Fig 6C–6E). The triangular protractor area is triangular and observed on the mesioventral part of the condyle on NMING:F35216 (f.mp, Fig 6D and 6E). The poorly preserved *margo limitans* is located anteriorly from the brachial process, and extends dorsally to form a supra-articular ridge (m.lim, sar, Fig 7C–7E). This ridge thickens slightly anteriorly and shows what could be interpreted as an auxiliary articular facet (a.fac, Fig 6D and 6E). The dorsal and ventral edges of the axillary foramen can be distinguished, though poorly preserved (f.ax, Fig 6C). If our reconstruction is correct, the foramen is as large as the whole brachial process.

The posterior ventrolateral plate is known from NMING:F35201, NMING:F35214, NMING:F35215 and TCD 19618 FB4033 (Figs 2A, 2B, 6J, 6K and 8D). NMING:F35214 and NMING:F35201 correspond to small and juvenile specimens, as witnessed by anastomosed ridged ornament in NMING:F35214. NMING:F35214 is preserved as bone posterolaterally and as internal mould in the rest of the plate. NMING:F35215 belongs to a much larger and adult individual, and harbours a tuberculated ornamentation. The lateral lamina shows a typical posterior buttress indenting the mixilateral plate. The ventral lamina is as wide as the lateral is high in the juvenile specimen.

No **semilunar** nor **median ventral plate** were identified in the material.

**Pectoral fins.** The elements of the pectoral fins can be found dissociated, in chunks or still attached to the armour. The fins are long, and are reconstructed as extending far beyond the posterior edge of the armour (confirming the belonging to the family Bothriolepididae). On NMING:F35201, the left proximal segment (in anatomic connection with the armour) is almost as long as the armour itself (lateral corner of PVL); the right element is observed on the other side of the specimen (Fig 2A and 2B). NHM P 59677 corresponds to an assemblage of head, trunk armour and pectoral fin elements that belonged to one adult individual, indicating a [headshield length / proximal pectoral fin length] of 0.33.

The fins are better known from the proximal segments than the distal segments, but isolated bones of the latter occur in our material (Fig 10). The proximal segments appear gently bowed, convex laterally and concave medially in younger forms (see NMING:F35201, Fig 2A and 2B). In larger specimens, the lateral edge seems straighter, while the medial side keeps its concavity.

The first central dorsal and first central ventral are the most proximal elements (Cd1, Cv1, Figs 2A, 2B, 7, 8E-8K and 11), and they articulate with the brachial process of the anterior ventrolateral plate. When preserved, these articulation areas appear narrow (ar3d, ar3v, Fig 8E and 8G). Tentative reconstructions show that these plates did not enclose completely the brachial process; on the contrary two slits, the mesial one being larger, allowed for some proximolateral fin movements. The lateral marginal plate 2 (Ml2) contacts the central dorsal and central ventral with an acute angle.

The first mesial marginal plate (Mm1, Figs 7B, 7H, 7I, 8K, 11E and 11F) is seldom complete (but see NMING:F35230, Fig 9A). It separates the first central dorsal and central ventral plates approximately at the same level than the latter two connect. The proximal edge of the mesial marginal plate is rounded, located slightly behind the brachial process, and this associated

with the axillary foramen of the anterior median ventral plate; this is particularly well visible in NMING:F35216 (Fig 7). Based on the shape and relationships of the other plates of the proximal segment of the fin, it is likely that that the mesial marginal plate 1 does not contact the central ventral 2, because of the median proximal process of the lateral marginal plate 2 (see other side of NMING:F35201; Fig 8K).

The lateral marginal plate 2 (Ml2, Figs 8O, 9D and 9E) is the most striking bone of the fin; it wraps the proximal segment of the fin laterally, and is gently curved; its curvature may increase in very large individuals (NMING:F35224, Fig 9D and 9E). Although ornamentation varies throughout the size and hence the age of each individual, so is the case of the lateral edge of the plate.

The mesial marginal plate 2 is another large element of the fin, visible on both the ventral and mesial side. Younger and older individuals show a single row of bigger protrusions on the angle ventromesial edge, with the median ones fused to each other and the most distal ones being separated (see NHMP 59683, Fig 9B). The mesial edge of the plate is gently concave. Central ventral 1 and median marginal 2 contact each other.

The second central dorsal plate is a small rounded element, elongated proximally, and terminated distally by the narrow articular area for the distal segment of the pectoral fin (Cd2, afd, Figs 8O, 9G–9J). The second central ventral plate is a much larger element, although not connecting anteriorly with the first mesial marginal plate (Cv2, Figs 7, 8H and 8I).

Few elements can be attributed with certainty to the distal segment of the fin. A central dorsal 3 and a central ventral 3 plates are recognized in respectively NMING:F35226 and NMING:F35227 (Cd3, Cv3; Fig 10A and 10B). A series of elements, possibly preserved in medial view, are recognised in NMING:F35228a. A large pointed element is tentatively assigned to a terminal element (Fig 10C and 10D).

The posterior part of the body was likely smooth and not covered with bony scales, as none is encountered in our material, even in the articulated ones (see NMING:F35201, Fig 2A).

## Preliminary phylogenetic analyses

Several sets of analyses were attempted, which are described below; they are sorted depending on the selected outgroup (see also Table C in S1 File for summary of indices). Only the last one is thoroughly described. It should be kept in mind as a disclaimer that addressing the phylogenetic resolution of a speciose genus such as *Bothriolepis* is difficult, even more with a rather small and simple set of recoverable morphological characters (especially compared to other Placodermi groups such as Arthrodira or Acanthothoraci). Besides, because it is exclusively fossil and very ancient, molecular and DNA approaches cannot be undertaken. This genus records also a remarkable longevity, spanning from late Emsian to late Famennian. Moreover, although impossible to test, if hybridization occurred, it would have contributed to blur the relationships. Lastly, the fossil record is by definition incomplete and of variable preservation, so phylogenetic reconstructions are likely to evolve with further discoveries.

**Outgroup = *Remigolepis* + *Grossilepis*.**   A first set of analysis includes the Asterolepididae *Remigolepis* sp. and the Bothriolepididae *Grossilepis* sp. as outroups (50 characters * 47 taxa). But as TNT can take into account as outgroup the first taxon only (i.e. *Remigolepis*), *Grossilepis* was resolved within the ingroup, as sister-group to *B. cellulosa*; this poor result, contradictory with the conception of a separate genus for *Grossilepis* is likely related to long-branch attraction phenomenon (see [49]:277; [32]) but also reminds Janvier and Pan's hypothesis [24] (see also Fig 12A). The most basal species are Euramerican in the obtained topologies, highly contradicting previously considered scenarios (topologies and indices are given in S9-S12 Figs in S1 File).

**Outgroup = *Grossilepis*.** *Remigolepis* is removed from the data matrix and *Grossilepis* is considered as single outgroup (50 characters * 46 taxa). The most basal species are again Euramerican in the obtained topologies, highly contradicting previously considered scenarios (topologies and indices are given in S13, S14 Figs in S1 File).

**Outgroup = *Bothriolepis niushoushanensis*.** Non *Bothriolepis* taxa (i.e. *Remigolepis* and *Grossilepis*) are removed from the data matrix, and *Bothriolepis niushoushanensis* is taken as outgroup (50 characters * 45 taxa). *B. niushoushanensis* is considered as one of the most ancient species of the genus, as it occurs in the Eifelian of Ningxia (Northern China Palaeoblock). The topology of the consensus tree reveals a high number of polytomies and is poorly resolved (topologies and indices are given in S15, S16 Figs in S1 File). It is noteworthy that above the outgroup an ensemble of four Gondwanan species (*B. karawaka*, *B. portalensis*, *B. kohni* and *B. macphersoni*) is in polytomy with a cosmopolitan clade. The resolution is however considered non-satisfactory enough for further consideration.

**Outgroup = *Bothriolepis shaokuanensi*.** *B. shaokuanensis* is also considered as one of the most ancient species of the genus, as it occurs in the Eifelian of Guangdong (Southern China Palaeoblock). The topology of the consensus tree reveals an irresolution above the outgroup (topologies and indices are given in S17, S18 Figs in S1 File).

## Main phylogenetic analysis (outgroup = *Bothriolepis askinae*)

**Tree topology.** *Bothriolepis askinae*, from the Givetian of Antarctica, is taken as outgroup (50 characters * 45 taxa). This species is selected preferentially because it is resolved as sister group to all other *Bothriolepis* species since 1988 (misspelled *B. askini* in [25]:text-fig. 68B). Another reason to consider this taxon as outgroup is related to the presence of a "simple" preorbital recess (type 1), considered as the primitive condition since the first phylogenetic reconstructions attempts. The research led to 96 most parsimonious trees of 210 steps each ($n_{MPT}$ = 96, $L_{MPT}$ = 210 steps, $CI_{MPT}$ = 0.333, $RI_{MPT}$ = 0.509) then used for a Traditional Search (tree bisection–reconnection (TBR) algorithm), from which a strict consensus tree was calculated (topologies and indices are given in S23, S24 Figs in S1 File). The strict consensus tree is $L_{CS}$ = 253 steps long, has a $CI_{CS}$ = 0.277, and a $RI_{CS}$ = 0.358; the majority rule consensus tree is $L_{C50\%}$ = 251 steps long, has a $CI_{C50\%}$ = 0.279, and a $RI_{C50\%}$ = 0.365 (Figs 13–15; S25, S26 Figs in S1 File). Robustness analysis was performed ($1 \leq$ Bremer Index $\leq 5$; see further).

The majority rule consensus tree is reasonably well resolved, especially at its base. The Givetian Irish *B. dairbhrensis* n.sp. is sister group with the Frasnian Scottish *B. gigantea* (node 5, Fig 13). This clade (node 4) sits in a paraphyletic sequence composed at its base and respectively by the Gondwanan *B. askinae*, *B. mawsoni* and *B. fergusoni* (nodes 1, 2, 3) and above with a clade (node 6) divided into a Gondwanan clade containing *B. longi*, *B. baretti* and *B. grenfellensis* (node 7)and a more cosmopolitan clade (node 9) described below. From node 9, the species *B. nitida* is sister group of two clades (node 10), the first one being Euramerican (node 11) with a short Gondwanan grade (nodes 12, 13) in-between, and a second clade (node 17) showing *B. hickingli* resolved as sister group of a very poorly resolved clade containing Euramerican, Chinese and Gondwanan taxa (node 18).

**Description of selected clades.** A complete list of character transformation generated from PAUP is provided in Supplementary Information. We focus below on major clades of the majority rule consensus in Fig 13A.

The clade(*B. gigantea*, *B. dairbhrensis* sp. nov.) at node 5 is characterised by a ratio of the (anterior edge/mesial constriction width) of the premedian plate < 1.0 ($CI_{12}$ = 0.222, non-ambiguous transformation), a flat anterior edge of the postpineal plate ($CI_{17}$ = 0.286,

ambiguous transformation), a long supra-occipital sensory line groove ($CI_{42}$ = 0.200, non-ambiguous transformation).

The clade at node 6 is characterised by an axillary foramen longer than high ($CI_{38}$ = 0.5), a common contact point between the central ventral plates 1 and 2 and the mesial marginal plates 1 and 2 (Cv1, Cv2, Mm1 and Mm2) in the pectoral fin ($CI_{39}$ = 0.4, non-ambiguous transformation), and a curved central sensory line groove ($CI_{48}$ = 0.143, non-ambiguous transformation).

The clade (*B. longi* (*B. barretti*, *B. grenfellensis*) at node 7 is characterised by a position of lateral corner of skull roof at the same level as the junction between the premedian, lateral and paranuchal plates ($CI_{07}$ = 0.250, non-ambiguous transformation) and by large pits on the internal side of the lateral plate ($CI_{15}$ = 0.500, ambiguous transformation). This clade is reminiscent of the one suggested by Long [19].

The clade (*B. hayi* (*B. yeungae* (*B. africana* (*B. alvesiensis* (*B. hydrophila* (*B. taylori*, *B. canadensis*)))))) at node 11 is characterised by a round shape of the skull roof ($CI_{03}$ = 0.429, ambiguous transformation) and a light posterior indentation of the junction of the infraorbital sensory line groove on the premedian plate ($CI_{46}$ = 0.286, non-ambiguous transformation). The first character transforms again in *B. africana* (quadrangular shape of the skull roof, non-ambiguous transformation) and the second for the clade at node 15 ((*B. hydrophila* (*B. taylori*, *B. canadensis*))) (straight junction of the infraorbital sensory line groove on the premedian plate, ambiguous transformation).

The clade (*B. niushoushanensis*, *B. karawaka*, *B.* portalensis, (*B. kohni*, *B. macphersoni*)) at node 25 is characterised by a pentagonal preorbital recess ($CI_{01}$ = 0.333, non-ambiguous transformation), a posterior postorbital process extending anteriorly from the orbital fenestra ($CI_{02}$ = 0.286, non-ambiguous transformation), a lag between the mesial and ventral laminae of the lateral and paranuchal plates ($CI_{06}$ = 0.333, ambiguous transformation), a lateral corner of the skull roof anterior to the junction between the premedian, lateral and paranuchal plates junction point ($CI_{07}$ = 0.250, ambiguous transformation), a ratio of the [width of the interomesiolateral edge / width interoposterolateral edge) of the nuchal plate > 1 ($CI_{23}$ = 0.2502, ambiguous transformation), a quadrangular postmarginal shape ($CI27$ = 167, non-ambiguous transformation), a minor crest on the median dorsal plate ($CI_{32}$ = 0.250, ambiguous transformation), a straight contact between the anterior median dorsal and anteriordorsolateral plates ($CI_{33}$ = 0.111, ambiguous transformation), a contact between the central first ventral and the second mesial marginal plates in the pectoral fin ($CI_{39}$ = 0.400, non-ambiguous transformation). The mesial and ventral laminae of the lateral and paranuchal plates are at the same level (ambiguous transformation), ratio of the [width of the interomesiolateral edge / width interoposterolateral edge] of the nuchal plate = 1 (ambiguous transformation) in *B. karawaka*. The postmarginal plates becomes pentagonal in *B. karawaka* and the clade (*B. kohni*, *B. macphersoni*) (ambiguous transformations). The lateral corner of skull roof becomes posterior to the junction point between the premedian, lateral and paranuchal plates, there is no dorsal crest on the median dorsal plates and the contact between the anterior median dorsal and anterior dorsolateral plates becomes sinuous in *B. niushoushanensis* (ambiguous transformations). This grouping is reminiscent of the one suggested by Young [25] although this author envisaged a paraphyletic ensemble at the base of a "trifid" preorbital recess rather than a monophyletic group.

**Character history.** Below is described and discussed the distribution of character states deemed important by previous authors.

The character "shape of the preorbital recess" (character 1) is divided into 5 states (character 1; state 0: simple; 1: trilobate; 2: pentagonal; 3: trifid; 4: trapezoid) and is a priori unordered and unpolarised (Fig 13B). The simple shape of the preorbital recess is considered

plesiomorphic as it is also present outside the genus *Bothriolepis* (but poorly developed and restricted to the premedian plate), and *de facto* as it is the pattern of the chosen outgroup (*B. askinae*). The character has a consistency index of 0.333, and its retention index is of 0.5.

The previous character state transformation sequence for this character was thought as follows [10, 25]: 1←0→2→3 [trilobate←simple→pentagonal→trifid], with a strong probability according to which the trifid state is derived from the pentagonal one, rather than two independent states. It is however noteworthy that the group supported by a pentagonal preorbital recess appears monophyletic (see topology in Young, 1988 [25]:text-fig. 68B, node defined by character i). or possibly paraphyletic (see unresolved topology in Lukševičs, 2001 [10]:fig. 83). As for the fifth state "trapezoid" identified by Moloshnikov [37], it would be the most derived state (see topology in [10]:fig. 83, despite it was not coded as trapezoid). It is nevertheless noteworthy that in Young's scheme, a clade based on the trifid preorbital recess appears one node more coronally in the tree at node r; Young thus considered the pentagonal state as intermediate, or he would have used it to characterise the sister clade of the one defined by r. Last, the shape of the preorbital recess is unknown in seven out of the eleven species in the "pentagonal–non-trifid" clade have a known preorbital (node defined by characters j and k in [25]).

The current phylogenetic analysis based on the strict or majority rule consensus trees reveals a completely different transformation sequence. The distribution of the states reveals only one unresolved clade supported by the pentagonal recess (node 24: *B. karakawa*, *B. portalensis*, *B. niushoushanensis*, *B. macphersoni* and *B. kohni*; Fig 13A). This clade is in polytomy with an assemblage of Gondwanan, Euramerican and Chinese forms, showing simple, trifid and trapezoid shaped preorbital recesses (node 18).

Another important result is that the trifid state is not resolved as the most apomorphic state, but rather seems to constitute an important grade between the simple shaped recess basally, and the simple, trilobate, pentagonal and trapezoid more coronally. The states are however not scattered randomly in the tree, as evidenced by the character consistency index (CI = 0.333; RI = 0.500).

Last, the distribution of the states in the tree reveals the *a posteriori* ordering and polarisation of this character: 0↔3, 3→1, 3→2, 3→5 (Fig 13C). The central position of the trifid preorbital recess questions its previously assigned apomorphic status, but rather pledges for a plesiomorphy instead, and hence echoes the original hypothesis by Janvier and Pan [24]. This fits with the stratigraphic record, as the earliest occurrence of *Bothriolepis* is *B. shaokuanensis* from the Eifelian of South China showing a trifid preorbital recess.

The presence of large pits on the internal side of the lateral plate (character 15) was proposed by Long ([19]:text-fig. 13) to support a clade of three Gondwana forms: *B. cullodenensis*, *B. gippslandiensis* and *B. fergusoni)*. More taxa were added later to this clade based on this same character: Long & Werdelin [45] added *B. tatongensis*, Young ([25]:text-fig. 68B) added *B. barretti*, Johanson ([9]:fig. 12) added *B. africana*, and Johanson & Young ([26]:fig. 10) considered the possible inclusion of *B. longi* (among other positions). This character is not listed in Lukševičs's matrix and cannot consequently become a synapomorphy ([10]:fig. 83, appendix). The clade of previous authors is not retained; the species previously listed are scattered into three clusters (*B. cullodenensis* + *B. gippslandiensis*; *B. barretti*; *B. longi*) or not being included in the study (*B. africana*, *B. yeungae*). Our study added the species *B. grenfellensis* [50]. We did not retrieve a clade based on large lateral pits either in our analysis. The consistency index for this character in our majority rule consensus tree is CI = 0.500, and the retention index is RI = 0.666. The derived state is distributed in two distant monophyletic clusters: *B. longi*, *B. barretti* and *B. grenfellensis* basally, and *B. cullodenensis* and *B. gippslandiensis* more apically.

Characters of the cheek have been considered. An anterior submarginal plate attachment supported by a separate ridge in the transverse lateral groove is suggested by Young ([25]:text-fig. 68) to support an unresolved and cosmopolitan clade, to which Johanson ([9]:fig. 12) proposed the possible addition of *B. yeungae*, and Johanson & Young ([26]:fig. 10) that of *B. longi*. These last authors also considered a smaller Gondwanan and South Chinese clade devoid of Euramerican and other Gondwanan species ([26]:fig. 11). This last smaller option is the one retained by Lukševičs ([10]:fig. 83; inclusion of *B. macphersoni*, *B. karawaka*, *B. longi*, *B. kohni*, *B. portalensis*, *B. niushoushanensis*; exclusion of *B. vuwae*, *B. mawsoni*, *B. paradoxa*, *B. obesa*). Originally coded in our data matrix, the character was finally removed based on a poor completeness (15%; character 34 in original matrix; see S3 Table). The shape of the submarginal plate is however taken into consideration (character 29, 31% completude).

Regarding the fin elements (character 39), Young [25] considered that the contact between the mesial marginal 1 and central ventral 2 plates, separating the central ventral 1 and mesial marginal 2 plates, supported a clade within the simple preorbital recess grade, consisting of the Gondwanan *B. alexi*, the Euramerican *B. canadensis*, *B. cellulosa*, *B. panderi*, *B. nielseni*, *B. alvesiensis*, *B. hydrophila* and the South Chinese *B. tungseni*). This clade was not retained in Lukševičs' tree [10], the state being scattered in two different branches of his tree (fig. 83). This cannot be related to the slight difference of the chosen taxa (*B. alexi* and *B. tungseni* are not part of Lukševičs's study), but rather to the assumption that *B. nielseni* is considered as belonging to a trifid preorbital recess group and is resolved very apical compared to the other forms which remain closer to the root. In our analysis, this character is modified from two to three states: 0: all plates contact in one point; 1: contact between the mesial marginal 1 and central ventral 2 plates; 2: contact between the mesial marginal 2 and central ventral 1 plates. This refining of the character results in a decrease in its consistency index (CI = 0.400; RI = 0.666). No clade nor grade can be identified immediately based on this character. However, three interesting observations can be made, but rely on the optimisation of missing data. 1) A contact between the mesial marginal 1 and central ventral 2 plates is distributed as a basal grade and present in the coronal polytomy. 2) All four plates contacting in one point are distributed among an intermediate grade between the base and the coronal polytomy. 3) The contact between the mesial marginal 1 and central ventral 2 plates is distributed in the apical polytomy only (Fig 14A–14D).

**Best indexed characters (CI or RI ≥ 0.666).**   Ten characters have a Consistency or Retention Index equal to or greater than 0.666. They are discussed below.

A shorter antorbital region (character 5) is primitive and present in the most basal forms of *Bothriolepis* (*B. askinae*, *B. fergusoni*; CI = RI = 1).

The character 15 involving the size of the lateral pits has a CI = 0.500 and RI = 0.666. See above for discussion.

A lateral corner of the skull roof posterior to the level of the contact between the postpineal plate and the orbital margin (character 16) is resolved as the primitive state. A lateral corner at the same level or anterior is observed only in the apical polytomy (node 18) and in *B. grenfellensis*; one occurrence of a posterior lateral corner is observed in this clade with *B. shaokuanensis* (CI = 0.250; RI = 0.684).

Anterior flanges of the postpineal plate (character 18) are observed only in the basal form *B. fergusoni* (CI = 1, autapomorphy).

A nuchal plate as long as wide (character 20) is observed only in *B. tatongensis* (CI = 1; autapomorphy).

A squarish Nu with convex anterior division of the lateral margin and short posterolateral corners (character 25, [25]) is observed in *B. leptocheira-curonica* (CI = 1; autapomorphy).

A mixilateral plate not broadest at its dorsal corner in adults (character 35) was observed in *B. canadensis* (CI = 1; autapomorphy)

The character 39 involving the fin elements has CI = 0.400; RI = 0.666. See above for discussion.

The derived condition of a branch of infraorbital sensory line diverging on prelateral plate parallel to the rostral margin of the head-shield (character 43) is displayed only in B. maxima (CI = 1; autapomorphy).

The derived condition of a long branch of the infraorbital sensory groove/canal diverging on prelateral plate (character 45) is observed only in *B. heckeri* (CI = 1; autapomorphy).

**Iterative reweight of the character "shape of the preorbital recess".** A series of analyses following the same parameters as the one before (OG = *B. askinae*; 50 characters $^*$ 45 taxa) was performed but by changing iteratively the weight of the first character "shape of the preorbital recess". The weight was increased until a value of w = 8 (analyses with reweight values of 9 and greater were impossible to perform due to memory errors). We focus on the main indices of the trees: number, length, general CI and RI, CI and RI for the first character for the obtained trees and the majority rule consensus tree; these results are displayed in Table D in S1 File and figured in S33-S47 Figs in S1 File.

**Evolution of indices.** The number of retained trees increases when the reweight of the character equals w = 2 (n = 268), but then decreases until a reweight of w = 4 (n = 30), from which the number of trees remains somehow constant. The length of obtained trees increases gently with the reweight of the character (between L = 210 at w = 1 to L = 247 at w = 8); the length of the majority rule consensus trees is more fluctuant but remains stable ($L_{50max} = 251$ at w1; $L_{50min} = 232$ at w = 3; L50 = 251 at w = 8). The consistency and retention indices of the trees and of the first character for the obtained trees and the majority rule consensus trees increase slowly with the reweighing of the character, but not at the same rate. A reweight of w = 5 is necessary to obtain a CI = RI = 1 for the original trees, but w = 6 is required to obtain the same values for the majority rule consensus. None of the reweighing options solves the discrepancy between stratigraphy and phylogeny. From w = 4 and above, *L. zadonica* is resolved with an ensemble of trifid preorbital recess, thus avoiding this latter state to characterise a clade; consequently, there is no clade defined by a trifid preorbital recess in any of the reweight options.

**Evolution of topology relative to iterative reweighting.** At w = 2 (S34, S35 Figs in S1 File), the topology of the majority rule consensus tree differs greatly from the one previously described (w = 1). *B. canadensis* is resolved as intermediate within a basal Gondwanan grade; *B. dairbhrensis* remains sister group for *B. gigantea*, but this clade is in polytomy with Gondwanan and Euramerican species, and is not resolved as basally as in the previous analysis. The states of character are not consistently distributed, except for the pentagonal shape which characterises a clade in a trifid preorbital grade.

At w = 3 (S36, S37 Figs in S1 File), the character states are a better distributed, although the plesiomorphic state (simple preorbital process) characterises a basal grade and two more apical ensembles (one cosmopolitan grade and one Euramerican clade); the trilobate and trifid shapes are successive grades, and the pentagonal shape characterises one clade within the cosmopolitan trifid grade. *B. canadensis* is still basally resolved within a Gondwanan grade; *B. dairbhrensis* and *B. gigantea* are part of a partly resolved grade of Euramerican forms resolved above the basal Gondwanan grade.

At w = 4 (S38, S39 Figs in S1 File), *B. canadensis* is again resolved in a basal Gondwanan grade. *B. dairbhrensis* and *B. gigantea* are intergroup of *B. hayi* and *B. yeungae*, and this larger clade is at the base of a grade characterised by a trifid preorbital recess, located just apical to a trilobate grade. Better distributed, only the pentagonal shape characterises an apical clade

resolved above a simple cosmopolitan shape grade, itself resolved above a cosmopolitan trifid shape, itself resolved above the trilobate grade.

At w = 5 (S40, S41 Figs in S1 File), the majority rule consensus tree is less well resolved than at w = 4. The clade ((*B. dairbhrensis*, *B. gigantea*), (*B. hayi*, *B. yeungae*)) is unresolved in polytomy with many other Euramerican trifid preorbital recess forms. A clade characterised by the trilobate shape is resolved within the simple shape grade. The pentagonal shape corresponds to a clade resolved with an unresolved coronal ensemble of simple shape.

At w = 6 (S42, S43 Figs in S1 File), the state of characters are well distributed. The trilobate shape forms constitute a clade in the basal simple-shape grade. The pentagonal shape forms a clade sister group with a more apical clade of simple shapes. The clade ((*B. dairbhrensis*, *B. gigantea*), (*B. hayi*, *B. yeungae*)) is unresolved in polytomy with many other Euramerican trifid preorbital recess forms. The Chinese taxa *B. shaokuanensis* and *B. niushoushanensis* belong to an unresolved clade characterised by the pentagonal shape of the preorbital recess.

At w = 7 (S44, S45 Figs in S1 File), and w = 8 (S46, S47 Figs in S1 File), the topology is similar to the one observed in w = 6.

**Robustness of the tree: Bremer decay index.** We used Bremer Index (BI) to evaluate the robustness of the tree. We ran a series of searches for suboptimal trees corresponding to n = $n_{MPT}$ +x, $n_{MPT}$ being the number of steps in the most parsimonious trees, and x being the Bremer index value. The following code line was added to the original TNT search script:

subopt x; bbreak;

We then computed the consensus trees and observed which branches collapsed from the MPT topology presented above.

For BI = 1, the number of maximal trees retained was expanded to 100 000, and the research led to 12 629 trees. The strict consensus is poorly resolved (S48 Fig in S1 File), *B. askinae* remains sister taxa to a polytomy containing unresolved species, except a few sister group species: *B. gigantea* and *B. dairbhrensis*, *B. tatongensis* and *B. jarviki*, and a trichotomy between *B. hydrophila*, *B. taylori* and *B. canadensis*. The majority rule consensus tree topology is preserved, and is even slight better resolved than that of the most parsimonious trees (S49 Fig in S1 File): *B. karawaka* is sister group to *B. portalensis*, and *B. kohni* to *B. mcphersoni*. These two latter groups are in polytomy with *B. niushoushanensis*.

For BI = 2, the research had to be limited to 1 million trees. The strict consensus tree topology is reduced to a complete polytomy with *B. askinae* as outgroup (S50 Fig in S1 File). The majority rule consensus tree is still surprisingly well resolved, and only a few branches collapse (S51 Fig in S1 File). *B. yeungae*, *B. hayi* are in polytomy with a resolved clade containing ((((*B. canadensis*, *B. taylori*) *B. hydrophila*) *B. alvesiensis*) *B. africana*). *B. volongensis* is resolved in polytomy one node more basally than in BI = 1.

For BI = 3, the number of maximal trees retained is kept at 100 000 000. The strict consensus tree topology remains a complete polytomy with *B. askinae* as outgroup (S50 Fig in S1 File). The majority rule consensus tree is slightly better resolved than at BI = 2, as a clade reappears containing *B. niushoushanensis* in polytomy with two clades consisting of (*B. karawaka*, *B. portalensis*) and (*B. kohni*, *B. macphersoni*) (S52 Fig in S1 File).

For BI = 4, the number of maximal trees retained is kept at 100 000 000. The strict consensus tree topology remains a complete polytomy with *B. askinae* as outgroup (S50 Fig in S1 File). The majority rule consensus tree is again better resolved than at BI = 3, as a clade reappears containing *B. ornata* as sister group to a clade containing *B. ciercere* and *B. volongensis* (S53 Fig in S1 File).

For BI = 5, the number of maximal trees retained is kept at 100 000 000. The strict consensus tree topology remains a complete polytomy with *B. askinae* as outgroup (S50 Fig in S1 File). The majority rule consensus tree is again better resolved than at BI = 4. Several clades

appear in the apical polytomy sister group with *B. hickingli*: *B. niushoushanensis* becomes sister group with *B. kohni* and *B. macphersoni* (S54 Fig in S1 File), and the three of them is sister group with a clade uniting *B. karawaka* and *B. portalensis*; *B. traudscholdi* becomes sister group of the clade (*B. leptocheira-curonica* (*B. evaldi*, *B. jani*)); *B. maxima* becomes sister group of *B. panderi*. The clade sister group to the clade holding *B. hickingli* and the apical polytomy is also resolved: compared to BR = 4, *B. yeungae* and *B. hayi* are resolved as successive sister groups for the previously resolved (*B. africana* (B. alvesiensis (*B. hydrophila* (*B. taylori*, *B. canadensis*)))).

Summary of results

Several important pieces of information must be retained from the unweighted majority rule consensus tree. 1) The base of the tree is very well resolved and kept in the strict consensus tree, providing us a higher degree of confidence than previous analyses. 2) A short Euramerican clade containing the Givetian Irish species *B. dairbhrensis* n. sp. is quite basally resolved in a Gondwanan paraphyletic sequence. 3) Except at the base of the tree, there is no clear larger palaeogeographic clade or grade identifiable (i.e. which would have been totally Gondwanan, Euramerican or Chinese). 4) The character "shape of the preorbital recess" (character 1 of the current matrix) is not as consistently resolved as what it was assumed from previously published phylogenies (see distribution in dedicated column of Fig 12). 5) The most ancient forms are not the most basal in the tree (but this is *pro parte* related artificially to the choice of the outgroup, see previous attempts above); on the contrary they are resolved quite apically. These remarks invite for important palaeobiogeographic and taxonomic consideration undertaken in the discussion part. 6) The species *Livnolepis zadonica* is resolved within the genus *Bothriolepis*. 7) Even a slight reweight of a character modifies greatly the tree and characters indices and topologies.

## Discussion

### A brief history of systematics and classification

The diagnoses of the suborder Bothriolepidoidei Miles, 1969 [20], the family Bothriolepididae Cope, 1886 [21], and the genus *Bothriolepis* Eichwald, 1840 [8] have changed and been emended over the years by successive authors since the discovery of the first specimens. Some characters were added or deleted, while some were assigned to another rank; for example, the diagnosis of the suborder Bothriolepidoidei in Young and Gorter, 1981 [51] was subsequently considered as familial rank for the Bothriolepididae as soon as Long, 1983 [19]. It is noteworthy that Denison [5] did not consider suborders but rather three families within the Antiarcha (Bothriolepididae [21], Asterolepididae Traquair, 1888 [52] and Sinolepididae Liu, 1958 [53]; labelled respectively Bothriolepidae, Asterolepidae, Sinolepidae, but see van der Laan ([54]:3) for familial suffixes), and that he included Yunnanolepididae Gross, 1965 [55] (non Miles, 1968 [20] as published; see [54] into Bothriolepididae.

Suborder Bothriolepidoidei [20]

Long ([19]:318) proposed the following definition for the suborder Bothriolepidoidei: antiarch with a short postorbital region of the headshield; postpineal separated or almost separated from lateral plates; otico-occipital depression narrow relative to headshield breadth; prelateral plate present; anterior median dorsal plate with broad anterior margin; posterior dorsolateral plate and posterior lateral plates replaced by a single mixilateral; semilunar plate unpaired.

Moloshnikov emended this definition into the following ([47]:740): head shield hexagonal or rounded hexagonal; preorbital recess located on its internal side; postpineal adjoining laterals or separated from them by nuchal; head shield with posterior oblique pit-line grooves; united mixilateral present; semilunar unpaired; trunk shield with brachial process, ventral

notch absent; pectoral fins long, reaching posterior margin of carapace, divided into two segments; proximal segment longer than distal segment. One can notice the common agreement on the absence or presence of contact between lateral and postpineal plates at the subordinal level between these two authors.

Family Bothriolepididae [21]

The diagnosis for the family Bothriolepididae is the same as that of the suborder by Young and Gorter [51], before being amended by Long ([19]:318) into the following: Bothriolepidoidei with a small postpineal plate separated by the lateral by the nuchal, which forms part of the posterior margin of the orbital fenestra; preorbital recess well developed; obstantic process (of the anterior dorsolateral plate) strongly developed; adducted pectoral appendage reaching back beyond the trunk shield; dorsal central plate 2 small, and separated from dorsal central 1 by mesial and lateral marginal plates 2.

Lukševičs ([10]:505) displaced the presence of a mixilateral from the subordinal to the familial diagnosis. Moloshnikov's definition ([47]:740) is much shorter, as some characters were included in the suborder: head shield hexagonal [thus insisting on suborder character], with well pronounced corners; postpineal not bordered by laterals. All authors agree on the lack of contact between the lateral and postpineal plates for the family Bothriolepididae.

Moloshnikov was more restrictive in his proposed subordinal and familial diagnoses, but we suppose this is due to the inclusion of new genera *Livnolepis* Moloshnikov, 2008 [47], *Rossolepis* Moloshnikov, 2008 [47] and *Tubalepis* Panteleyev, 2003 (see [47, 56]; Moloshnikov even considers that *Tubalepis* could belong to another family). Moloshnikov ([57]:fig. 7) summarized the different morphologic characteristics of the different families within Bothriolepidoidei (Microbrachiidae Stensiö, 1931 [58], Dianolepididae Long, 1983 [19], Bothriolepididae Cope, 1886 [21] and Tubalepididae Moloshnikov, 2011 [57]), based on the course of the cephalic lateral sensory line relative to the posterolateral corner of the paranuchal plate, the shape of the anterior median dorsal plate, and the shape (or absence) of the medioventral plate. It is noteworthy that van der Laan ([54]:28) listed other families as synonyms of the former in the order Bothriolepiformes (Antiarcha being probably understood as super-order).

Genus *Bothriolepis* [8]

The genus *Bothriolepis* was first erected by Eichwald in 1840 [8] for some dermal body armour remains from the red sandstones of the Novgorod area, and was categorized as "fish" together with *Holoptychius* remains ([8]:78–79). As customary at this time, no formal diagnosis was given, nor any reliable description by today's standards can be retained. The oldest diagnosis for the genus that we could find in the literature is the one published by Stensiö in 1948 ([43]:223).

Stensiö noted a high number of species in the genus as early as 1948 [43], and provided the following diagnosis for *Bothriolepis*: Bothriolepinae forms of varying size; anterior median dorsal plate broadest at its lateral corners, overlapping the anterior dorsolateral plate and overlapped by the mixilateral plate; posterior median dorsal plate generally narrower anteriorly; mixilateral plate broadest through its dorsal corner, with its lateral lamina of greater extent to the lateral lamina of the anterior dorsolateral plate; ornament regularly reticular in immature individuals, less reticular and more short ridges and tubercles, without anastomose in mature individuals, and concentrically and radially distributed. Most of these characteristics were retained by later authors.

Miles ([20]:222) shortened drastically Stensiö's [43] diagnosis to retain only the dermal ornament characters to differentiate *Bothriolepis* from *Grossilepis*, although he acknowledged the necessity to revise it once the genera *Dianolepis* and *Wudinolepis* had become better known.

In 1978, Denison provided a wide definition reflecting the already speciose nature of the genus: ornament is reticular in juveniles, and may remain reticular in adults or be broken up into short ridges or tubercles; trunk armour may be flat or high, even crested; anterior median dorsal plate tapers towards both ends, normally overlaps the anterior dorso-laterals and is overlapped by the mixilaterals; posterior median dorsal narrows anteriorly. The author also lists a respectable number of generic synonyms ([5]:108).

Young and Gorter [51] provided a more precise diagnosis for the genus: Bothriolepididae in which the anterior median dorsal plate is broadest across its lateral corners, and normally overlaps the anterior dorsolateral, and is overlapped by the mixilateral plate; mixilateral plate broadest through its dorsal corner, with its lateral lamina of similar extent to the lateral lamina of the anterior dorsolateral plate, and not forming extensive contact with the anterior ventro-lateral plate. Young [25] completed the diagnosis by adding a posterior median dorsal plate narrowing anteriorly, strongly reticulate ornament in juveniles and mainly reticulate or broken up into short ridges and tubercles in adults. It is noteworthy that Lukševičs [10] followed the diagnosis of Young and Gorter [51].

The most recently amended diagnosis for the genus *Bothriolepis* was proposed by Moloshnikov ([47]:741). The genus is then diagnosed as displaying large postorbital process imprints on the inner side of the skull roof, almost parallel nasal and facial laminae for the preorbital recess, contact between the postmarginal and paranuchal plates, widest nuchal plates [of all species], aligned lateral corners of the anterior and posterior median dorsal plates, presence of a median ventral plate and possible presence of a dorsal median crest. This diagnosis differs greatly from its predecessors, mainly because he created new genera within the Bothriolepididae.

The legitimate question of a definition (i.e. both diagnosis and specific content) for a monophyletic genus *Bothriolepis* s.s. is discussed below.

Non-*Bothriolepis* genera

Among the following genera, the most striking characteristics in the diagnoses may rely in the absolute size of the taxon. Some differences with the genus *Bothriolepis* defined above relate with the contact between some plates (e.g. postpineal and lateral plates), their shape (e.g. width of various parts of the anterior or posterior median dorsal plates), as well as their relative overlapping. Consequently, some forms were later excluded from the family Bothriolepididae, although remaining in the Bothriolepidoidei. The position of the posterolateral corner of the skull relative to the posterior edge of the orbital fenestra, the paranuchal participation to the posterolateral corner can be considered relevant to justify the non-belonging to *Bothriolepis*, but closer look reveals that certain patterns occur in some species of *Bothriolepis*. A simple (semicircular) preorbital recess is observed in *Grossilepis* and *Wufengshania*; only *Livnolepis* displays an unusual trapezoid preorbital recess. Diagnoses of the following taxa may be copied from original publications without substantial modification deemed possible (N.B. for co-authors and editor: that's for plagiarism disclaimer/check).

*Grossilepis* Stensiö, 1948 was erected by Stensiö ([43]:523) who considered that the Latvian *B. tuberculata* described by Gross [59] differed "considerably from the other *Bothriolepis*-species". The diagnosis of *Grossilepis* differs from that of Bothriolepis in the following: anterior median dorsal plate of equal width beyond the external postlevator process; anterior median dorsal plate possibly overlapping the mixilateral plate; posterior median dorsal plate of "strikingly uniform" width; anterior dorsolateral and mixilateral plates of uniform height throughout their length; dorsal laminae of anterior dorsolateral and mixilateral plate of similar widths; ornament "delicate" (but remaining reticulated in juveniles and tuberculated and/or ridged in adults).

The genus *Wudinolepis* Chang, 1965 and the family Wudinolepididae were erected by Chang [60] with the following diagnosis: very small form with total length of head and body

armour not exceeding 20 mm; dorsal and lateral walls of armour with regular and symmetrical ridges; similar in *Bothriolepis* in length proportion of head shield, trunk-armour and pectoral fins. It is noteworthy that Chang considered the family belonging to Asterolepiformes, mainly by opposition with Yunnanolepiformes, than *contra* Bothriolepiformes as he compares *Wudinolepis* with *Bothriolepis* rather than with *Asterolepis*; he also considered the family Bothriolepididae belonging to the Asterolepiformes.

*Dianolepis* Zhang, 1965 is diagnosed by Zhang [61] by a moderate size; a short obstantic margin of the head shield, a large orbital fenestra; posterior lateral corner far beyond the posterior edge of the orbital fenestra; postpineal plate short and broad excluding the nuchal plate from contacting the orbit; postpineal plate contacting the lateral plates; anterior margin of nuchal plate very short limited to notch for postpineal plate; absence of median crista on visceral surface of nuchal; dorsal wall of trunk armour rather high; dorsal median ridge elevated into high crest; sutures between anterior median dorsal plates and anterior dorsolateral and mixilateral plates similar to those of *Bothriolepis*; posterior margin of posterior median dorsal plate strikingly broad; median dorsal division of crista transversalis interna posterior clearly convex; very small axillary foramen on the anterior ventrolateral plate; pectoral fins fairly robust; lateral and medial spines absent; ornamentation consisting of cycloidal tubercles in juveniles and adults. This genus was later assigned to its own family Dianolepididae Long, 1983 [19] (thus prevailing recent practice over Wudinolepididae Chang, 1965 [60], according to van der Laan [54]), as some of its diagnostic characteristics contradicted the Bothriolepididae diagnosis (notably the lack of contact between the postpineal and lateral plates). The subfamily Tenizolepidinae Moloshnikov, 2011 [62] is considered different from the Dianolepidinae by Moloshnikov [62]. The family Jiangxilepididae Zhang, 1991 [63] is considered synonym of Dianolepididae by van der Laan [54].

The genus *Monarolepis* Young, 1988 was created by Young [25] to differentiate the form *B. verrucosa* from other *Bothriolepis* species. He gave the following diagnosis: bothriolepid with a small axillary foramen on the anterior ventrolateral plate, a transverse crescentic ridge on the visceral surface of the anterior median dorsal plate, absence of the posterior oblique abdominal pit line groove of the trunk, poorly developed central sensory line of the head; lateral laminae of the anterior and posterior ventrolateral plates of uniform height; anterior dorsolateral plate of uniform width; median ventral plate of relatively large size; dermal ornament consisting of coarse tubercles.

The Australian and monospecific genus *Briagalepis* Long, Burrett, Pham Kim Ngan et Janvier, 1990 [64] was erected to exclude *B. warreni* from the genus *Bothriolepis*, on the basis of its very short overlap area for the anterior median dorsal and anterior dorsolateral plates, combined to its lack of reticulate ornament. Its diagnosis defines *Briagalepis* as a small bothriolepid with a trunk shield about 6 cm long; an anterior median dorsal plate with very short overlap areas for the anterior dorsolateral plate; a crescentic ridge on the visceral surface of the anterior median dorsal plate anterior to the small ventral median pit; anterior median dorsal plate with wide anterior margin; lateral plate with a short depression for the postorbital process of the neurocranium which does not extend anteriorly as far as the orbital notch; dermal ornament consisting of scattered tubercles.

The genus Vietnamaspis Long, Burrett, Pham Kim Ngan et Janvier, 1990 [64] is diagnosed as follows: bothriolepid antiarch having a trunk shield with a maximum estimated length of about 12 cm; posterior median dorsal plate approximately parallel-sided in its posterior half, and with W/L index under 60; anterior median dorsal plate with W/L index close to 80 with the anterior margin about one third the maximum width of the plate, and lacking a dorsal sensory-line groove. This diagnosis separates *Vietnamaspis* from *Bothriolepis* and *Monarolepis* which possess large posterior median dorsal plates with non-parallel lateral margins, and from

*Dianolepis* and *Grossilepis* by the narrow anterior margin of its anterior median dorsal plate ([64]:185).

*Livnolepis* Moloshnikov, 2008 [47] (type-species *Bothriolepis zadonica* [65]) was distinguished from the genus *Bothriolepis* within the Bothriolepididae mainly because of the presence of an unusually large preorbital recess: the angle between the nasal and facial laminae has a value comprised between 50 and 80° in *Livnolepis*, by opposition to the slit-like and parallel laminae in other antiarchs [47]. Later, the subfamily Livnolepidinae Moloshnikov, 2012 [62] was erected to contain the genera *Livnolepis* and *Rossolepis* Moloshnikov, 2008 [47], but was not considered valid by van der Laan ([54]:28). The preorbital recess condition is unknown is *Rossolepis*. Although not questioning the legitimacy of this character, we wonder whether the thin preorbital recess and its parallel nasal and facial laminae could not be a result of either compression or damaged specimen (the nasal lamina is indeed very fragile and seldom identified; see for example [9]:fig. 5C–5D). *Livnolepis* is diagnosed as follows: imprints of anterior postorbital processes of endocranium superficial, poorly pronounced; preorbital recess very large; facial and nasal laminae positioned at 60°–80°; paranuchal bordered by postmarginal; greatest width of nuchal, anterior and posterior mediodorsals observed in line with lateral corners (nuchal in line with lateral and posterolateral corners); trunk shield with medioventral; dorsal crest well developed, high. *Rossolepis* is diagnosed by "imprints of anterior postorbital processes of endocranium very poorly pronounced, superficial, very narrow at base; preorbital recess large, superficial.

The genus *Tubalepis* Panteleyev, 2003 [56] corresponds *pro parte* to the material previously assigned to *Bothriolepis extensa* by Sergienko [66] (see also [56, 67]). Although assigned to a different family (Tubalepididae Moloshnikov, 2011 [57]), we include *Tubalepis extensa* in our palaeomap illustrations (Famennian; S6 Fig in S1 File). The genus is diagnosed as follows: medium-sized bothriolepidid (trunk shield up to 10 cm long); anterolateral margin of the paranuchal plate absent; posterolateral angle of otico-occipital depression of skull roof almost straight; trunk shield typical of bothriolepidids, but median ventral plate absent; external sculpture of plates composed of narrow ridges forming network with poorly pronounced tubercles at points of intersection.

The genus *Houershanolepis* Lu, Tan et Wang, 2017 [68] was erected together with the type-species *H. zhangi* to address the problem of *nomen dubium* associated with the species *H. changi* which had been mentioned but never properly described nor diagnosed. The species is based on a single anterior median dorsal plate from the early Devonian of Houershan Mountain, in the Dushan County of Guizhou Province, south China. It is diagnosed as a small antiarch fish; anterior median dorsal plate longer than its breadth with a wide anterior margin which is nearly equal to its posterior margin; levator fossa short and post levator fossa present; the anterior ventral pit located at the first fifth of the anterior median dorsal plate length; overlap relationship between anterior median dorsal and neighbouring plates similar to that of *Bothriolepis*; ornamentation mainly composed of regularly displayed long ridges.

The genus *Wufengshania* Pan, Zhu, Zhu et Jia, 2017, was erected with the type-species *W. magniforaminis* [69]. It is diagnosed as a bothriolepid antiarch of small size, with length of skull roof less than 15 mm; breadth/length ratio of skull roof nearly 1.00; length ratio between orbital fenestra and skull roof about 0.52; preorbital recess semicircular in shape, and roof of preorbital recess convex; postpineal plate arched and large; breadth/length ratio of premedian plate about 1.20; dorsal surface of nuchal plate with X-shaped domed ridges; supraorbital sensory canal developed as a short groove disjoined from infraorbital sensory canal; occipital cross-commissure short; obtected nuchal area of skull roof developed; skull roof ornamented by irregularly arranged tubercles.

## Stratigraphic record of the genus *Bothriolepis*

The stratigraphic and palaeogeographic record of the genus *Bothriolepis* is summarised on the cladogram of Fig 15, as well as on a series of palaeomaps (S3-S8 Figs in S1 File). The most ancient forms occur in the Emsian and Eifelian of China (e.g. [70]). The genus then occurs later in Gondwana and Kazakhstan between the Givetian and Famennian (e.g. [62, 71]). Until now, the occurrence of *Bothriolepis* in Euramerica was confined to the Upper Devonian, but has now been lowered to the middle Givetian with *B. dairbhrensis* sp. nov. from the Valentia Slate Formation in Ireland (e.g. [5, 10]). The latest record of *Bothriolepis* is the terminal Famennian of South Africa and Belgium [41, 44].

It is noteworthy that some other genera of the family Bothriolepididae may display an earlier and wider occurrence than *Bothriolepis* species themselves.

## Comparison with previously published phylogenies of the genus *Bothriolepis*

It is important to keep in mind that all previous published phylogenies of the genus *Bothriolepis* were "hand-made"; these phylogenies used various "outgroups" to root their trees, including other members of the family Bothriolepididae, such as *Grossilepis* (e.g. [25]) or even *Dianolepis* (e.g. [19]). Most of these publications insist on the difficulty to obtain reliable results. Recent computerised cladistic analyses at a lower taxonomic level (i.e. within the Euantiarcha and not higher than generic level) confirmed the sister group relationship between *Grossilepis* and *Bothriolepis* (see [69]:fig. 9).

A preliminary reconstruction of the relationships within the genus *Bothriolepis* and with other antiarchs was proposed by Janvier & Pan ([24]:fig. 12; Fig 12A). Only four species of *Bothriolepis* were taken into account, but the most striking information in their cladogram is the potential inclusion of *Grossilepis* within *Bothriolepis*. The ensemble *Bothriolepis + Grossilepis* constitutes the family Bothriolepididae which has unresolved relationships with *Wudinolepis*, *Microbrachius* and *Hyrcanaspis*, forming altogether the Bothriolepida, itself a sister group of Asterolepida.

To our knowledge, the first larger cladogram was drawn by Long ([19]:315; 18 species of *Bothriolepis*; Fig 12B), who noted that the "bothriolepidid diversity [had given] rise to much parallelism". The genus *Bothriolepis* is mainly characterised by the lack of contact between the lateral and postpineal plates. Interestingly, although Long considered the species *B. verrucosa* [51] as the most primitive (because of the possession of a small axillary foramen, a weak median occipital crista and a central ventral 1 contacting the mesial marginal 2), he also considered that the opposite states of character united all other *Bothriolepis* species and *Grossilepis*. *B. verrucosa* was nevertheless sorted as the sister group for all other *Bothriolepis* species by Long ([19]:text-fig. 13), but was later assigned to the genus *Monarolepis* by Young [25] on the basis of the previously mentioned morphological characteristics. Long divided the possible shapes of the preorbital recess into three patterns: an "enlarged median division", an "enlarged lateral division" and a simple pattern (the latter considered primitive and unmentioned state in the tree). As often for cladistic analyses not relying on a data matrix, it is noteworthy that it is impossible to determine whether these patterns correspond to three different binary characters or to one multi-state character, nor whether the states were ordered and polarised. Long's cladogram however provided a start hypothesis for further palaeoichthyologists to work upon, as he considered this character essential to, if not create a cladistic classification (however self-admitted deemed impossible), at least identify morphotypes within the genus. Long proposed a simple preorbital recess as being the primitive condition, thus disagreeing with Janvier and Pan who thought the trifid shape was the most primitive [24]. It is noteworthy that the only

Chinese species in the cladogram (*B. shaokuanensis*) is not basally resolved despite its earliest occurrence, revealing that the author rightfully considered relationships based more on morphology rather than stratigraphy. This may also reflect a preconceived idea of Gondwanan origin for the genus, or at least [Gondwana + China].

The next important landmark in the history of the relationships within the genus *Bothriolepis* is the impressive study of the antiarchs from the Devonian Aztec Siltstone in Antarctica by Young [25]. As Long did, Young built a hand-made tree, based on a series of 18 apomorphies to define 12 nodes (Fig 12C). Four types of preorbital recesses are recognised, the plesiomorphic condition being the simple shape displayed in *B. askinae*, whereas clearly stated apomorphic conditions are trilobate, pentagonal and trifid ([25]:text-fig. 65); once again there is no indication regarding their coding as several characters or a multi-stated one. Following Long's decision to use the shape of the preorbital recess as the most important character to build a phylogeny but criticising a lack of resolution within a given morphotype group, Young put more emphasis on other characters such as those related to cheek attachment. The role of these characters was to support the consistency of groups based on preorbital recess-based morphotypes: two clades probably forming a paraphyletic ensemble related to the simple shape of the preorbital recess (apomorphies d and h, based on respectively "enlarged lateral pits" and "mesial marginal 1 contacting central ventral 2"), one clade corresponding to the pentagonal shape (apomorphies j and k, respectively "anterior SM attachment supported by a separate ridge in transverse lateral groove" and "axillary foramen longer than high") and one clade grouping "the most advanced species" possessing a trifid preorbital recess. The author concluded that this "proposed scheme [was] provisional", and awaited to be "tested by checking many uncertain aspects in many species". This tree however, appeared to be the final one upon which new species have been grafted upon over the next years (e.g. [9, 10, 26]; Fig 12D and 12E). In other words, a computerised matrix and test has never been successfully attempted. Other authors describing new material of *Bothriolepis* and naming (or not) new species carefully avoided the spiny problem of its systematics and postponed (for good reasons!) a cladistic analysis (e.g. [72–74]).

One exception has to be mentioned with the (aborted) computerised analysis by Lukševičs [10] who had built and published a proper data matrix for East European platform species (12 *Bothriolepis* species + *Grossilepis*). However, the tree obtained in PHYLIP [75, 76] was not retrieved in parsimony-using softwares (such as PAUP; [36, 77, 78]), and it was then decided with the editor to use Young's tree [25] and graft upon it (43 *Bothriolepis* species) instead of publishing the PHYLIP tree (Fig 12E). A last tree was published on-line only and not peer-reviewed as part of a more global phylogeny of living forms overview, and added the species missing in Lukševičs's tree (*B. africana*, *B. yeungae* and *B. heckeri*) as a basal polytomy [79].

The important information to remember from this very shortened history is that 1) previous phylogenies of the genus *Bothriolepis* were hand-made and not computerised, that 2) most proposed phylogenies relied on Young [25], who himself elaborated on Long [19], who both considered the preorbital recess as a reliable character to build consistent groups, that 3) only a small data matrix was built but the result was disappointing [10], that 4) a South Chinese origin is stratigraphically implied and acknowledged for *Bothriolepis*, but the most basally resolved species have consistently been Gondwanan (see [25]:text-fig. 68 A & B). Last, 5). Far from criticising former authors, we presently acknowledge the difficult nature of the genus *Bothriolepis* (including but not restricted to intraspecific variation, growth allometry, species nature, numerous taxonomic problems such as synonymy, etc.), and the lack of large computerised calculation until recently. We can quote the honesty of Z. Johanson who resumed excellently the state of things as early as 1998, as she wrote that "reanalyse phylogenetic

relationships of *Bothriolepis* [. . .] will require a considerable and painstaking search for the qualitative characters necessary" ([9]:339).

The new analysis we propose herein is not devoid of defaults and could only handle a handful of the numerous problems inherent to this genus. We hereby keep a humble attitude regarding previous authors who tried to tackle this taxon, and consider this new set of results and interpretations as new hypotheses to work on and test in the future.

## Phylogeny and characters

As mentioned above, certain characters have been considered as utmost important in *Bothriolepis* phylogeny. The most emblematic ingroup sorting is the shape of the preorbital recess.

**Preorbital recess.** Different preorbital recesses were first noted by Stensiö [43] who distinguished a "canadensis" (i.e. "simple, type 1) from a "groenlandica" type (i.e. trifid, type 4). Young [25] added two more types, a trilobate one (after *B. barretti*, type 2), and a pentagonal one (after *B. portalensis*, type 3). The latest addition of a fifth pattern consists in a very thick and trapezoid preorbital recess, hitherto known only in *Livnolepis* (*Bothriolepis*) *zadonica* [37].

As mentioned above in the discussion, the shape of the preorbital recess has been used to build morphotype-based groups and phylogenies [41]. In other words, the shape of the preorbital recess was considered to record the evolution of the genus *Bothriolepis* in a consistent and strict parsimonious way, no reversion or convergence being deemed possible. This comfortable *status-quo* remained unchanged and rarely challenged until today. Johanson indeed mentioned some problems of keeping phylogenies based on preorbital recesses: keeping this in mind while hand-building a tree inevitably may lead to unconscious over-weighing of the character (when all characters should have the same weight *a priori*; [9]:338–339). This was reflected in her attempts to place *B. yeungae* in the clades identified by Young ([25]: 3 different position hypotheses; [9]:fig. 12). This problem persisted later, as she and G. Young tried to resolve *B. longi*: 4 positions and 2 different tree topologies were suggested ([26]:figs 10 and 11). As a matter of fact, the authors mentioned clearly that "none of the [major clades previously recognized] should be regarded as strongly supported" ([26]:68). This poses the problem of a possible inherent homoplasy within the character "shape of the preorbital recess", but also all the other ones. This was tested in our own analysis, and it appears that the "preorbital recess shape" has some degree of homoplasy (CI = 0.333, RI = 0.500; see below/above).

In 1988, Young considered that the trifid recess unifies all northern hemisphere taxa [25]. This hypothesis is no longer supported in our cladogram, as it was already challenged by the occurrence of such characteristic in Gondwanan (*B. yeungae*) and Chinese forms (*B. shaokuanensis*).

Far from supporting a very apical clade as expected by Young, the trifid preorbital recess is resolved as potentially leading to other shapes (Fig 13C). Although the morphology-based cladistic analysis is disconnected from stratigraphic information, this result echoes with the fact that an Eifelian Chinese form (*B. shaokuanensis*) possesses a trifid preorbital recess. It is even worth remembering that the trifid recess was once considered plesiomorphic among Bothriolepidoidei because suggested possibly homologous with the preorbital depression of the Yunnanolepidoidei [24]. On the contrary, Long denied such homology on the basis that the yunnanolepidoideid preorbital depression was ornamented and thus could not hold the rhinocapsular ossification; consequently he considered the trifid process as apomorphic among Bothriolepididae [19]; Wang and Zhu confirmed the non-homology between the preorbital recess and the preorbital depression in antiarchs [80].

It is also noteworthy that the trifid shape is quite widespread, and that a common condition is often *a priori* considered as the primitive condition; this however relates to the knowledge

we have of the group (*a fortiori* based on incomplete fossil data) and the sampling (see [81, 82]). Moreover, the simple (semicircular) preorbital recess is encountered outside of the genus *Bothriolepis* (i.e. *Grossilepis* and *Wufengshania*); despite these genera contain very few species, such distribution of a simple recess can be considered even wider than a trifid one.

By extended outgroup comparison, and considering that a simple preorbital recess occurs outside *Bothriolepis* (although restricted to the premedian plate), it is difficult to conceive a morphologic continuous transformation from the trifid to pentagonal shape. The state distribution observed in the main majority rule consensus as well as in those issued from the reweights of the preorbital recess shape character invite for an exploration of Emsian and Eifelian terranes and taxa in order to possibly discover new forms which will help answer this ordination and polarisation problem, and consequently the importance of the character in the *Bothriolepis* phylogeny.

**Cheek plates.** Young considered the morphology of the anterior attachment area of the submarginal plate on the lateral plate, associated with the shape of the prelateral plate attachment area: a strong transverse anterior attachment for the submarginal plate, separate from the one for the prelateral plate would have occurred in the pentagonal group. Such pattern was also identified in the trifid form *B. yeungae* [9, 25]. Young also envisaged the modification of the shape of the anterior median dorsal plate throughout ontogeny as an relevant character [25]. As mentioned above, the poor completeness of these characters was not considered high enough to include the characters in the analyses (see S3 Table).

One character considering the shape of the submarginal plate is retained in our analysis (Character 29; completeness = 31%; CI = 0.500). A low and long submarginal plate is considered as the plesiomorphic state (present in *B. askinae*); only two species in polytomy in the most coronal unresolved clade (*B. ciercere* and *B. ornata*) display the apomorphic state (short and deep).

## Phylogeny and stratigraphy

It is well accepted that no stratigraphic information should be coded equally to morphological characters in a cladistic analysis, but rather considered as added value information *a posteriori*, once the trees are constructed. Rare are the trees in which phylogenetic sequences (of taxa or characters alike) follow gently a smooth geological timescale. Previous *Bothriolepis* phylogenies are no exceptions, as already noted by Long [19] or Johanson and Young [26]. A case in point is the non-basal and phylogenetically scattered position of the Chinese taxa, despite their earliest stratigraphic occurrence (S3 Fig in S1 File). This is partly related to the presence in these forms of either the pentagonal shape of the preorbital recess (although questionable in *B. niushoushanensis* according to [26]:71). These authors also questioned the actual biogeographic and biostratigraphic proximity between Australian and Antarctic fauna, arguing for a yet too incomplete fossil record of the genus *Bothriolepis* in East Gondwana ([26]:72). Stratigraphic correlations are indeed hard to assess for many reasons (e.g. erosion, physical access to outcrop, obvious incomplete fossil information recording), and rely among other things on vertebrate macroremains (e.g. Arthrodira Phyllolepida and *Wuttagoonaspis*, Antiarcha *Bothriolepis*) and microremains (e.g. *Turinia*). Authors have pledged for further taxonomic studies of these latter taxa (e.g. [26]).

Our current analysis is not free from this symptomatic stratigraphy-phylogeny discrepancy. It therefore invites for further work on the earliest and most basal forms of *Bothriolepis*.

## Phylogeny and taxonomy

**How many (valid) species of *Bothriolepis*?.** *Bothriolepis* is undoubtedly a speciose genus (i.e. related to the high number of species rather than a visually pleasing aspect; see [83]). The

increasing number of *Bothriolepis* species through time is plotted on a diagram, based on the taxonomic authority of each species, including synonyms; articles mentioning indeterminate species and redescribing previously named species are not taken into account (S2 Fig and Table A in S1 File). A different colour-code indicates whether a given species remained valid, became a junior synonym of another *Bothriolepis* species or was reassigned to another genus.

Obviously the publication of new species depends on many factors (e.g. access to localities, historical, anatomical and systematics comprehension of the material, etc.) and should be seen through the eyes of a natural science historian rather than through those of a data analyst. The discovery and publication of new species relies on many independent factors, such as access to fossil locality (sometimes extremely difficult, such as Antarctica), geopolitical context (some fossils could indicate terranes of economic importance considered non-disclosable and embargoed for a period of time), but also the technology available for excavation, preparation and study, as well as previously published works. However, a rapid overview is possible in order to get a general idea of the research dynamics associated with the different *Bothriolepis* species.

95 species are recorded, ranging from 1840 with the erection of *Bothriolepis ornata* [8], to *B. dairbhrensis* sp. nov. in 2021 Several articles published more than one new species at a time and suddenly increased the number of species and contributed to the then knowledge of the genus diversity. Chronologically, they are: 1) Agassiz, 1844 (3 species [84]), 2) Traquair, 1888 (2 species [52]), 3) Traquair, 1890–92 (1 species, but it is also associated with an other assemblage; i.e. *B. leptocheira curonica* [85]), 4) Gross, 1942 (2 species [86]), 5) Stensiö, 1948 (3 species [43]), 6) Miles, 1968 (6 species [20]), 7) Long, 1983 (4 species [19]), 8) Young, 1988 (10 species; 13 more indeterminate species are not taken into account because unnamed [25]) and Malinovskaya, 1988 (4 species, one of them was later reassigned to the genus *Tubalepis* [87]).

Many species are described on an incomplete set of dermal plates and endocranial structures (some taxa lack head or thoracic armour). The individual variability within a population and/or a species, associated with growth heterochronies and ontogenetic variations increase the taxonomic complexity of the task (e.g. plate shapes and sizes, ornamentation, sensory line system; e.g. [46, 48, 88]). Some species may have been erected based on stratigraphic and (palaeo-)geographic reasons rather than morphological, leading to the increase of taxonomic complexity of *Bothriolepis* (see for example *B. nitida* vs. *B. minor*, *B. leidyi*, *B. coloradensis*, *B. darbiensis* and *B. virginiensis*, although the latter is considered morphologically distinct and therefore a valid species by Weems, 2004; see [27–29]). As far as the current study is concerned, 77 species of *Bothriolepis* are considered valid, and 43 are retained for our final phylogenetic analysis (plus *Livnolepis* (*Bothriolepis*) *zadonica*).

**Could the genus *Bothriolepis* be paraphyletic?.** For decades, the monophyly of the genus *Bothriolepis* has remained essentially unquestioned. This is due to a combination of different factors: the number of species increasing with time (see above), the late access to cladistic methodology (Hennig's work was translated in 1966 in English, and 50 species of *Bothriolepis* were then recorded; [89–91]), and computing facilities and calculation power for numerous taxa and characters data matrices (the first phylogenetic software was PHYLLIP by Felsenstein in 1980; 59 species recorded then; [75]). To these technical limitations it should be added that the difficulty of data treatment (polymorphism, missing data, stratigraphic considerations, etc.), together with a morphologically ungrateful material, has contributed to the procrastination of a complete *Bothriolepis* overview study. A review of previous phylogenetic analyses is given above, and all were handmade (see also Fig 12).

The non-monophyly of *Bothriolepis* (rather its paraphyly) was suggested by Janvier & Pan ([24]:fig. 12) when they considered the possible inclusion of *Grossilepis* within the genus *Bothriolepis*. These authors reminded that *Grossilepis* differs only slightly from the then known

species of *Bothriolepis*. They even considered that the 'simple' preorbital recess shared by *G. tuberculata*, *B. canadensis* and *B. cellulosa* may be a synapomorphy for these forms.

The non-monophyly of *Bothriolepis* was later suggested in a two-step process: 1) the publication of Lukševičs' tree including *B. zadonica* ([10]:fig. 83), and 2) the exclusion of this species from the genus *Bothriolepis* and the erection of the genus *Livnolepis* by Moloshnikov [47]. The main reason to assign *B. zadonica* to a distinct genus relies on the thickness of the preorbital recess and the angle between its nasal and facial laminae. For the same reason, Moloshnikov assigned *B. heckeri* to *Livnolepis* [92]. Surprisingly, these authors have never acknowledged the phylogenetic status consequence of this change of genus.

Interestingly, our phylogenetic analysis recovers the idea of Janvier & Pan because *Grossilepis* is resolved within the genus *Bothriolepis* when *Remigolepis* is taken as outgroup. This can be explained by either long branch attraction phenomenon or parallelism in *Bothriolepis* or a real misinterpretation of *Grossilepis* as a separate genus; these hypotheses advocate for a non-paraphyly of *Bothriolepis*.

However, the non-legitimacy of the separate genus *Grossilepis* appears a very weak argument, and is hardly believable considering the diagnosis of this taxon, relying on clear morphologic differences with *Bothriolepis* since its erection in 1948 by Stensiö. We consequently consider that analytic biases rather than real biological and evolutionary trends explain the position of *Grossilepis* within *Bothriolepis*, and that *Grossilepis* [43] should remain a valid genus.

Similarly, *Livnolepis zadonica* is resolved within the genus *Bothriolepis* in all of our analyses. Moloshnikov erected the genus several years after describing extensively *B. zadonica* [37, 47]. The most striking feature of the genus is the thick trapezoid preorbital recess, which was originally considered as a fifth morphologic possibility in the genus *Bothriolepis* [37]. It is however noteworthy that Lukševičs resolved *B. zadonica* as sister species of *B. wilsoni*, on the shared possession of a large median dorsal crest ([10]:fig. 83), and both species were included in a clade based on a trifid preorbital recess. The question hence revolves on the anatomical distinction and taxonomic distribution of this pattern.

Morphologically speaking, the distinctiveness of the preorbital process in *Livnolepis (Bothriolepis) zadonica* is unquestionable: the wide lateral flanges of the recess cannot be mistaken with longer lateral protrusion of the classic trifid recess. The apparent thickness of the preorbital recess in *Livnolepis* pledges for the recognition of a real distinct process rather than a variation of the trifid type.

Our phylogenetic analysis resolves *Livnolepis* within the genus *Bothriolepis*, but we acknowledge the lack of any other genus of the Bothriolepididae in the current data set. A wider sampling of Bothriolepididae genera and closely related forms, including a number of *Bothriolepis* species should be analysed in an attempt to resolve the validity of the genus *Livnolepis* (and possibly others); such analysis is beyond the scope of this article.

The question remains open regarding either the paraphyly of the genus *Bothriolepis* or *a contrario* the validity of the genera *Grossilepis* and *Livnolepis* (which should then be considered as junior synonyms of *Bothriolepis*). Arguing these questions is beyond the scope of the present article and will be the topic of another study (in progress).

## Review of the stratigraphic distribution of *Bothriolepis*

Below we review the occurrences of *Bothriolepis* and Bothriolepididae in the Emsian to Famennian terranes (S3-S8 Figs and Table B in S1 File). This review is important as we advocate that the new Irish *B. dairbhrensis* sp. nov. is the most ancient occurrence of the genus in Euramerica, and that this taxon (with others) is used for important palaeogeographic implications.

Until recently, the most ancient occurrence of *Bothriolepis* in Euramerica was considered to belong to the late Givetian Gauja Formation (Gauja regional stage) from the Main Devonian Field of the north-western part of the East European Platform [10].

**Emsian.**   The earliest record of *Bothriolepis* occurs in the upper part of the Emsian of South China with two indeterminate species from the Jiucheng and the Chuandong formations in Wuding and Qujing respectively (Yunnan, Assemblage V in [70]; S3 Fig in S1 File). It is noteworthy that other Bothriolepididae were identified and described in the Emsian of Taemas Wee Jasper in eastern Australia (*Monarolepis verrucosa*, previously assigned to the genus *Bothriolepis*; see [25, 51]) and of South China (*Wufengshania magniforaminis*, *Houershanolepis zhangi*, and *Wudinolepis weni*; see [60, 68–70]).

**Eifelian.**   The Eifelian record of the genus *Bothriolepis* is restricted to the Chinese palaeoblocks: *B. niushoushanensis* is encountered in the north in Ningxia [70, 93], while *B. shaokuanensis*, *B. sinensis*, *B. tungseni*, *B. yunnanensis*, and *Bothriolepis* sp. occur in the southern palaeoblock in Yunnan, Guangxi and Guangdong [30, 60, 70, 94, 95], as is the Bothriolepididae *Dianolepis liui* in Yunnan [61, 70]; the latter species is also encountered in the Givetian of Yunnan [96], although not listed as such by Zhu who considers the species present only in the Eifelian of the Haikou and Wuding Formations ([70]:378) (S4 Fig in S1 File).

**Givetian.**   The Givetian record of the genus *Bothriolepis* starts getting more cosmopolitan with evidence in the South China Palaeoblock, in Eastern Gondwana, in Euramerica and Kazakhstan (Figs 16 and 17, S5 Fig in S1 File). *B. lochangensis* and *B. kwangtungensis* are encountered in the Dahepo Formation in Lechang, Guangdong, and Anfu, Jiangxi [70, 97, 98], and *Bothriolepis* sp. is encountered in the Laohuao Group of Taishan, Guangdong [70, 98]. It is also present in eastern Gondwana, especially in the Antarctic Aztec siltstone with *B. antarctica*, *B. askinae*, *B. kohni*, *B. portalensis*, *B. barretti*, *B. alexi*, *B. karawaka*, *B. macphersoni*, *B. mawsoni*, *B. vuwae* and 13 indeterminate species [25, 99, 100], but also in eastern Australia with *B. longi*, *B. bindarei*, *B. tatongensis*, the latter possibly belonging to early Frasnian [19, 26, 45]. *Bothriolepis babichevi* and *B. kassini* occur in the Konyr Formation of central Kazakhstan, which corresponds to the end of Givetian or early Frasnian [62, 87, 101, 102].The earliest evidence of *Bothriolepis* in Euramerica comes from the mid-Givetian Valentia Slate Formation in Ireland, described in the present article. Other Bothriolepididae occur around the Givetian world: *Dianolepis liui* in the Haikou Formation of Yunnan ([96]; although not in [70]), *Vietnamaspis trii* from the Givetian–Frasnian of Vietnam [64], in eastern Australia with *Briagalepis warreni* (originally belonging to *Bothriolepis* and assigned to a new genus by Long, 1990 in [64]). Finally, *Grossilepis spinosa* is encountered in the Givetian Gauja Formation according to Sallan and Coates [96], but Lukševičs [10] considers this taxon as middle Frasnian (Pamūšis regional stage) (Figs 16 and 17, S5 Fig in S1 File).

**Frasnian.**   The genus *Bothriolepis* becomes really cosmopolitan in the Frasnian with fossil evidence in South China, eastern, northern and western Gondwana, eastern and western Euramerica (on either side of the Appalachian—Caledonian mountains) and in Kazakhstan (S6 Fig in S1 File). *Bothriolepis* sp. is observed in the Shetianqiao Formation of Taojiang, Hunan [70, 98]. Its presence in the early Frasnian of Mount Howitt, Victoria, Australia can be evidenced with *B. longi* and *B. tatongensis* (but see comment on possible Givetian age in former paragraph; [19, 26, 45]). *Bothriolepis* sp. is also present in the late Frasnian Gogo Formation in Western Australia [96]. As for the northern margin of Gondwana, *Bothriolepis* sp. would occur in the lower Frasnian of Iran (Chahriseh and Kerman; [96]). As for the north-western part of Gondwana, *B. perija* is encountered in western Venezuela [103, 104], and *Bothriolepis* sp. in the Cuche Formation of Columbia [105, 106]. Eastward from the Caledonian Mountains in Euramerica, the genus is present in the mid-late Frasnian of the Alves Beds of Scotland with *B. gigantea*, *B. alvesiensis*, *B. macrocephala*, *B.* major, *B. obesa*, *B. paradoxa*, *B. stevensoni*, *B.*

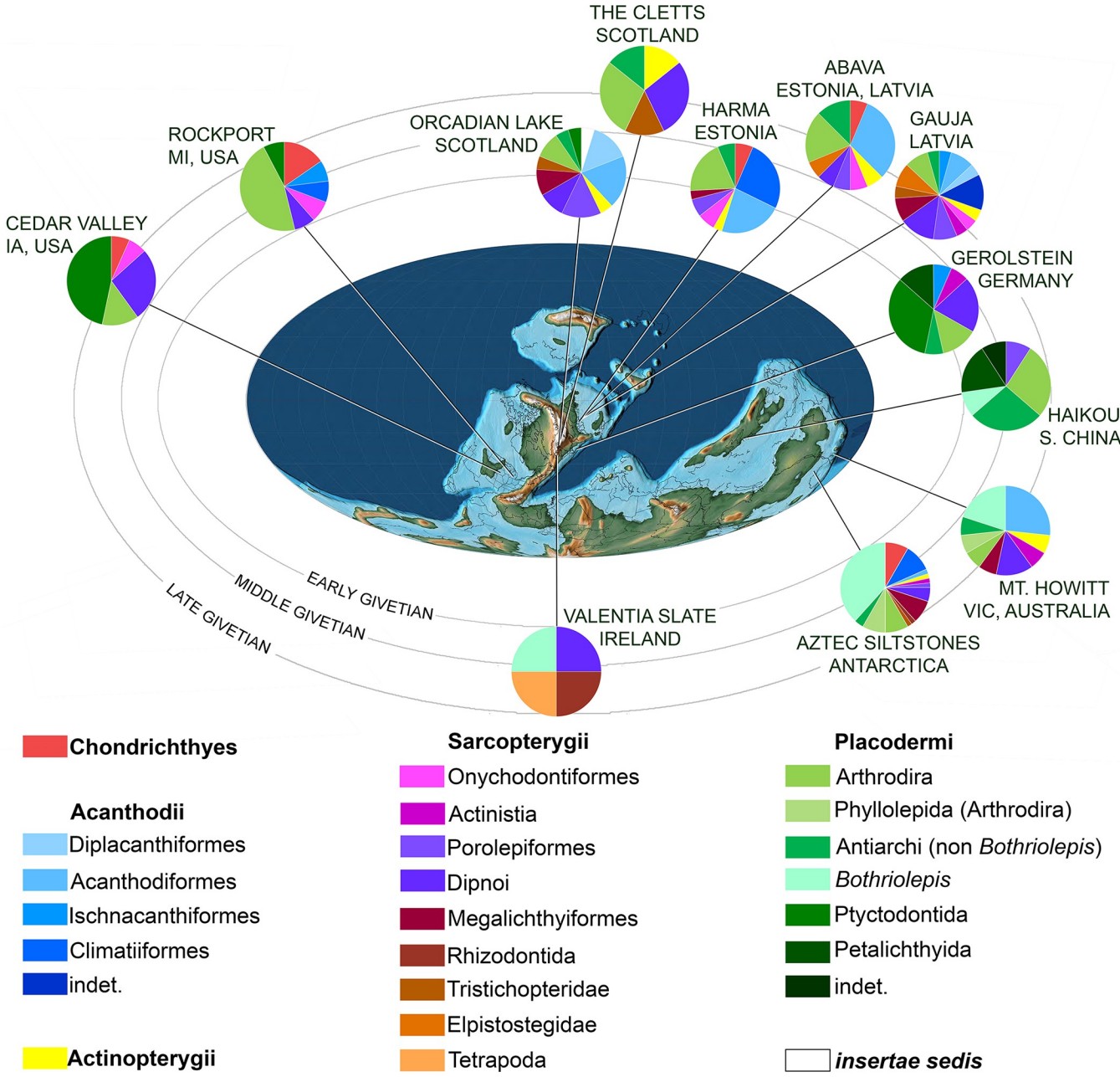

**Fig 16. Distribution and composition of Givetian vertebrate faunas based on macroremains.** Pies correspond to best known localities and associated updated taxa formerly listed in Sallan and Coates [96], with the addition of Valentia Slate formation; they are plotted on a palaeomap of the Givetian (modified after [38]; oceans in dark grey, shallow and epicontinental seas in light grey, emerged continents in white). Early Givetian localities have their pies closer to the edge of the map; late Givetian are furthest. NB. 13 unnamed and indeterminate species of *Bothriolepis* in Antarctica mentioned by Young [25] are taken in consideration in this figure.

*taylori* and *B. wilsoni* [5, 107], in the Baltic states and western Russia (East European Platform) with *B. leptocheira*, *B. leptocheira curonica*, *B. evaldi*, *B. favosa*, *B. cellulosa*, *B. maxima*, *B. obrutschewi*, *B. prima* (e.g. [5, 10]; *B. prima* and *B. obrutschewi* would be the earliest forms in the upper part of the Amata formation), and in Armenia, eastern Russia, Siberia, Timan and/ or Tuva, with *B. dorakarasugensis*, *B. jeremijevi*, *B. siberica*, *B. panderi*, *B. markovski*, *B. prima*,

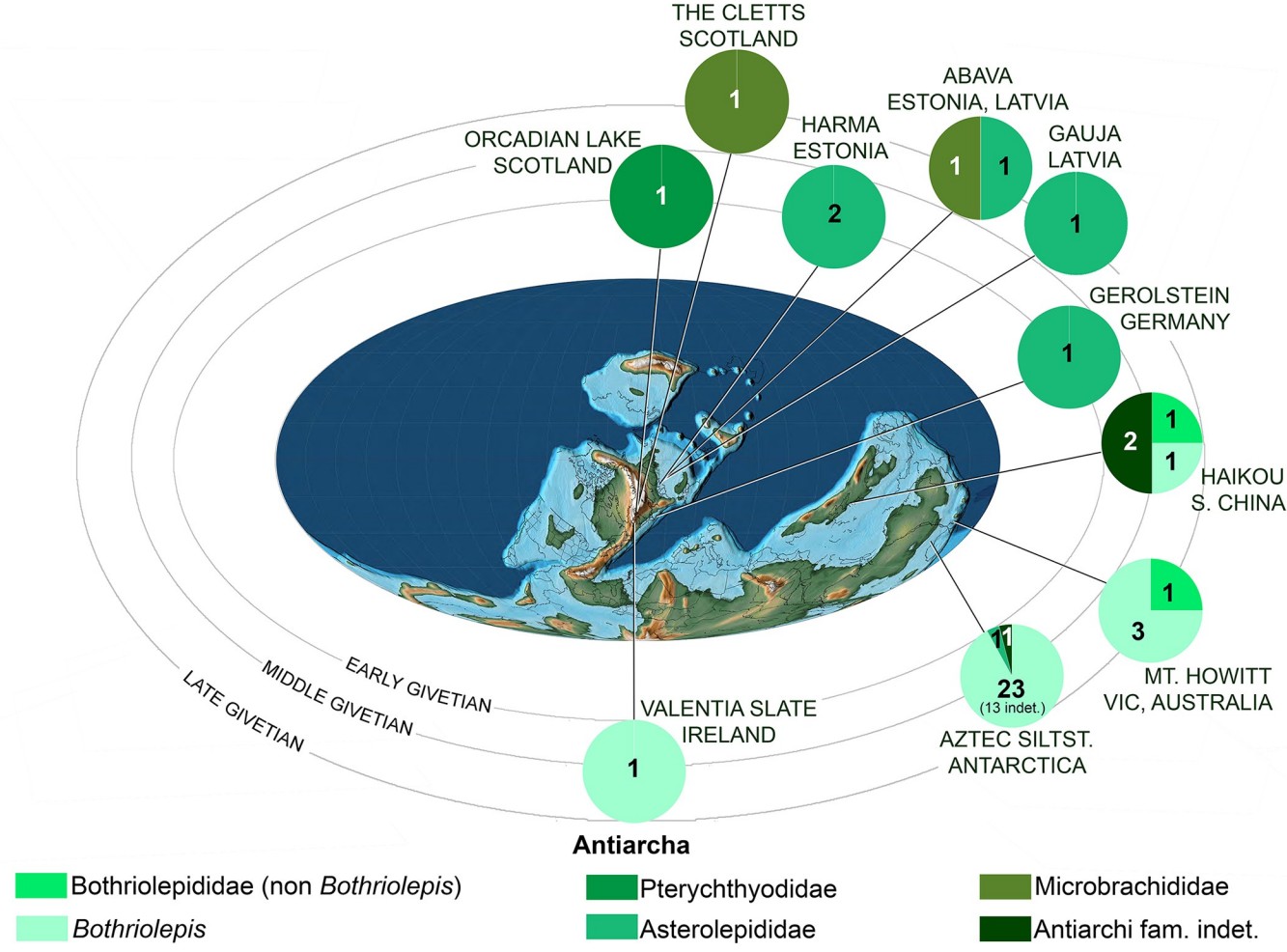

**Fig 17. Distribution and composition of Givetian Antiarcha.** Pies correspond to best known localities and associated updated taxa formerly listed in Sallan and Coates (96), with the addition of Valentia Slate formation; they are plotted on a palaeomap of the Givetian (modified after [38]; oceans in dark grey, shallow and epicontinental seas in light grey, emerged continents in white). Early Givetian localities have their pies closer to the edge of the map; late Givetian are furthest. NB. 13 unnamed and indeterminate species of *Bothriolepis* in Antarctica mentioned by Young [25] are taken in consideration in this figure.

*B. traudscholdi*, *B. maeandrina*, *B. volongensis* [5, 10, 62, 108, 109]. Westward from the Caledonian mountains in Euramerica, the genus occurs in the Frasnian of Canada (*B. rex*, *B. canadensis*, *B. traquairi*; [5, 74, 96]), in the eastern (*B. minor* and *B. virginiensis*; [5, 27, 96]) and western parts of the USA (*B. coloradensis* and *B. darbiensis*; [5, 28, 96, 110]). In the Kazakhstan palaeoblock, five species are recorded: *B. kassini*, *B. amakonyrica*, *B. nikitinae*, *B. tastenica*, and *B. sanzarensis* [57, 62, 87, 111, 112]. Indeterminate Bothriolepididae are encountered in Tarim (Xinjiang Province, China; [70]). *Briagalepis warreni* occurs in the early Frasnian of Victoria, Australia (but could possibly be dated as Givetian; see above), *Vietnamaspis trii* occurs in Vietnam, while *Grossilepis brandi* occurs in Scotland, and *Grossilepis spinosa* and *G.* tuberculata occur in the Baltic States.

**Famennian.** The distribution of the genus *Bothriolepis* remains cosmopolitan until the end of the Famennian, during which the genus becomes extinct as well as all other placoderms (S7, S8 Figs in S1 File). In the South China palaeoblock, *Bothriolepis* sp. is encountered in the Xikuangshan Formation of Lengshuijiang and Liuyang, Hunan [70, 98]. Occurrence in Gondwana is as scarce as in China, with few occurrences in the east (*B. yeungae* in Canowindra,

New South Wales, Australia; [9]), and in South Africa (*B. africana* in Witpoort; [41]). The genus appears to be the most diversified in Euramerica, especially east from the Caledonian mountains. In England and Scotland, the following species are encountered *B. alvesiensis*, *B. cristata*, *B.hayi*, *B. hickingli*, *B. hydrophila*, *B. laveroklochensis*, *B. leptocheira*, *B. macrocephala*, *B. obesa*, *B. paradoxa*, *B. stevensoni*, *B. wilsoni* [5, 10, 20, 43, 96]. In Eastern Europe the following species are encountered: *B. sosnensis*, *B. maeandrina*, *B. ciercere*, *B. jani*, *B. ornata*, *B. parvaniensis*, *B. heckeri*, *B. jazwicensis*, *B. jeremjevi* and *Bothriolepis* sp. [5, 10, 96], but also the Bothriolepididae *Livnolepis zadonica*, *Rossolepis brodensis* and *Grossilepis tuberculata* [5, 47, 65, 96]; *Tubalepis extensa* may belong to a separate family Tubalepididae [56, 57]. In the western part of Euramerica, *Bothriolepis nielseni*, *B. groenlandica* and *B. jarviki* occur in Greenland [5, 43, 96], *B. nitida* and *B. minor* occur in the eastern part of the U.S.A. [5, 96] and *B. coloradensis* in the west [5, 96, 110]. At the very end of the Famennian, *Bothriolepis* subsists in South Africa (see above) and in Belgium with *B. lohesti* and *Bothriolepis* sp., together with *Grossilepis rikiki* [113–115].

**Previous mentions of pre-Frasnian occurrence of *Bothriolepis* in Euramerica.** Below we review the occurrences of *Bothriolepis* in the pre-Frasnian terranes of Euramerica (i.e. Baltica and North America). This review is important as we advocate that the new Irish *B. dairbhrensis* sp. nov. is the most ancient occurrence of the genus in Euramerica, and that this taxon (with others) is used for important palaeogeographic implications.

Regarding Baltica, authors have moved the Givetian–Frasnian boundary between the Burtnieki, Gauja, Amata and Pļaviņas Regional Stages or their coevals in Estonia, Lithuania, Latvia, Belarus but also the western part of Russia [116–122]. Most recent works seem to consider a late Givetian age for the Gauja Formation and that the Givetian-Frasnian boundary would be situated at the base of the Amata formation [122–124].

It is also noteworthy that "the oldest remains of *Bothriolepis* in the Main Devonian Field, of probable Givetian age (Gauja Formation), were found in 1964 in borehole material from Latvia [125]. Unfortunately these remains are lost" ([10]:497). All the *Bothriolepis* material described in Lukševičs [10] come from the Main Devonian Field belonging to Upper Devonian deposits ([10]:499), with *B. prima*, *B. obrutschewi*, *B. cellulosa*, *B. panderi*, *B. traudscholdi*, *B. maxima* and *B. evaldi* coming from the Frasnian deposits, while *B. leptocheira curonica*, *B. ornata*, *B. jani*, *B. heckeri* and *B. ciercere* occur in the Famennian sequence [10].

*Bothriolepis curonica*, entered by Sallan and Coates [96] as occurring in the Givetian Gauja Formation actually refers to *B. leptocheira curonica* from the Famennian Eleja Formation (see [10]). Likewise, *Grossilepis spinosa* and *Taenolepis speciosa* rather belong to the Frasnian Pamūšis Regional Stage [10]. Only *Asterolepis ornata* from the Givetian Gauja Formation remains from Sallan and Coates's original list ([96]; supplementary information: 27).

In north America, *Bothriolepis* sp. was recorded from the mid-Givetian Rockport formation (see [96]), but the material was later re-attributed to juvenile (smaller) forms of the arthrodire *Protitanichthys rockportensis* ([126]:14). In other words, there is no pre-Frasnian occurrence of *Bothriolepis* in North America (see S3-S8 Figs in S1 File).

**Frasnian Western Gondwana.** The following forms could not be taken into account in our phylogenetic analysis because of the incompleteness of the taxa. However, their occurrence is acknowledged in our palaeogeographic discussion.

The Upper Devonian Cuche Formation of Columbia yielded some material attributed to Bothriolepis by Janvier & Villaroel described in 2000, who refer to an "elliptic shape" of the preorbital recess (thus "simple, type 1 preobital recess in our article). The inner side of the plate however shows a triangular vault which could indicate a trifid type (i.e. type 4 in our work; see [105]:pl. 1, fig. 7). A more recently described *Bothriolepis* sp. seems to have a simple

preorbital recess ([106]:fig. 4H). This would indicate the presence of at last two species of *Bothriolepis* in the Cuche Formation.

The Frasnian Campo Chico Formation in Venezuela was studied notably by Young and Moody in in 2000 [103]. The authors erected and described the species *Bothriolepis perija* from Venezuela. Rests of this species are found in "locality 139, Caño Colorado Sur, in the upper member of the Campo Chico Formation"([103]:166), which corresponds to an early to mid Frasnian age ([103]:164). They also raised concerns about the processes leading to the known spatial and temporal distribution of the Venezuelan fauna, and especially *Bothriolepis*. Such processes can only be inferred from phylogenetic analyses ([127] in [103]). *Bothriolepis* is one of the most widespread and abundant vertebrate in the Devonian world, and its occurrences span a large variety of environments (e.g. freshwater, reef or open marine, even brackish waters). Young et al. [104] suggested that *"Bothriolepis* no doubt dispersed through the sea". Interestingly, because of morphological affinities of *B. perija* with other Gondwanan rather than closer Euramerican forms, Young & Moody considered a dispersal scenario along the coasts rather than by crossing large areas of open water, and consequently concur with Rosen's definition of a continental fish for *Bothriolepis* ([103]:163).

**Notes on the stratigraphic uncertainties of some taxa.** Below we report the uncertainty of the stratigraphic information displayed in the cladograms (dashed lines in Figs 13–15). Although much larger than their real and likely more finite occurrences, these larger ranges do not infer the discussion regarding a northward dispersal of Gondwanan faunas in the Givetian.

- Regarding *B. darbiensis*, the most precise available information is Frasnian in the lower part of the Upper Devonian Darbty Fm. [5, 110].

- *B. cellulosa* occurs in the earliest Frasnian (Pļaviņas Baltic Regional stage, in [10]; Ust' Yarega Fm. in Timan, in [128, 129]).

- *B. volongensis* occurs in the Upper Part of the Rassokha Fm., dated as Frasnian [130], without further precision.

- *B. obesa occurs in the* Upper Devonian of the Jedburgh beds [5], without further precision.

- *B. niushoushanensis* occurs in zone VI in [70] dated as Eifelian.

- *B. longi* occurs in the Middle-Upper Devonian (Givetian-Frasnian) according to Johanson and Young [26]. The authors note strong affinities with Aztec Antarctic fauna, but acknowledge that this does not mean synchronicity.

- *B. hayi* is recorded in the Upper Devonian [5, 131], without further information.

- *B. hickingli* occurs in the Famennian [131], without further information.

- *B. virginiensis* occurs in the Upper Devonian, without further information

- *B. coloradensis* is noted as Frasnian (Elbert Fm.) by Denison [5, 110], but is actually Famennian (Chaffee Fm.; see [96]).

- Regarding *B. wilsoni*, we do not possess more precise available information than Upper Devonian [5]; it could at best belong to the upper part of the Upper Old Red Sandstone Lithofacies in ([131]:49).

- *B. gigantea* could be mid-Late Frasnian in age according to Becker et al. [107] in Parfitt et al. [132].

- *Lastly*, *B. alvesiensis* occurs in the Alves beds considered as mid–late Frasnian according to Becker et al. [107] in Parfitt et al. [132]. Rosebrae beds are considered as late Famennian according to Marshall in Rogers [133] (see Stephenson and Gould, 1995 [134], at http://earthwise.bgs.ac.uk/index.php/Devonian,_Grampian_Highlands); Rosebrae beds are considered as early Famennian by Sallan and Coates [96]. NB to coauthors, reviewers and editors: KEEP URL IN TEXT UNTIL PUBLICATION (OTHERWISE IT DISAPPEARS FROM THE ENDNOTE GENERATED LIST)

**Phylogeny and palaeobiogeographic implication: Proposal of an early northward dispersal scenario.** The Euramerican clade containing *B. dairbhrensis* n. sp. and *B. gigantea* resolved at the base of the tree in a Gondwanan paraphyletic sequence indicates a first northward dispersal episode from Gondwana to Euramerica of *Bothriolepis*, although it cannot be precised whether their common ancestor was Euramerican or Gondwanan (node 5, Fig 13). This first dispersion occurred in the middle Givetian, thus much earlier than what previously thought based on the sole stratigraphic occurrence of the most ancient Euramerica species of *Bothriolepis* (*B. cellulosa* in Latvia and Timan, *B. obrutshewi* in Baltic states, *B. prima* in Latvia and Armenia, for the species with the shortest and most secure stratigraphic range; see S. Fig 6 for other possible early Frasnian distribution in Euramerica).

It is noteworthy that the above hypothesis derived from the phylogenetic analysis of the sole genus *Bothriolepis* is consistent with stratigraphic data (except maybe the Chinese record), but is also supported by other vertebrate taxa (Figs 16 and 17). Preliminary observations indicate the presence of rhizodont sarcopterygians in the same Givetian Valentia Slate Formation fauna (i.e. different from the Frasnian *Sauripterus* sp. mentioned in [4]).

After this first Givetian northward dispersion, it is unsure whether some newly Euramerican forms swam back to Gondwana or whether a pre-existing pool remained. What is certain is that later Euramerican species are not directly related to the *B. dairbhrensis–gigantea* group, but instead presumably represent later dispersal waves.

The most apical half of the tree (node 18 and more apical) is mainly poorly resolved. This can be explained by several reasons. Taxonomically, the speciose aspect of the genus is problematic, and related to the validity of all species, together with the incomplete aspect of the fossil material. Biologically, this could correspond to interbreeding between closely related species (although this is obviously non-testable hypothesis). Evolutionarily, a rapid diversification can blur the relationships between species; this is likely to occur when ecological niches become newly available, intraspecies competition increases, and possibly with invasive species in a new environment devoid of predators (see [135]).

## Givetian vertebrate palaeoenvironments

**Methods.** In order to comprehend the palaeogeographic importance of the Valentia Slate vertebrate assemblage, it is important to compare its components with those of other Givetian localities (Figs 16 and 17).

Sallan & Coates [96] compiled a list of vertebrate taxa from several well-known macrofossil localities and compared their assemblage over several orders of Chondrichthyes, Acanthodii, Actinopterygii, Sarcopterygii and Placodermi. Their list is here updated (Figs 16 and 17; S5 and S6 Tables). The *Bothriolepis* material of Rockport Formation, MI, USA, was since re-attributed to juvenile smaller forms of the arthrodire *Protitanichthys rockportensis* ([126]:14). *Palaeospondylus* is considered as *incertae sedis*. The Bothriolepididae *Briagalepis warreni* is added to Mount Howitt fauna [19, 64]. For Gauja, *Asterolepis ornata* remains the only Givetian Antiarcha from their original list, as *Grossilepis spinosa*, *Taenolepis speciose* and *Bothriolepis*

*curonica* are dated as middle Frasnian [10]. *Vorobjevaia dolonodon* Young et al., 1992 [136] is added to the Megalichthyiformes of the Givetian Antarctic Aztec Siltstone [136, 137]. Sarcopterygii are slightly reorganised from the original analysis, with Sallan and Coates' 'Tetrapoda' being now divided into Elpistostegidae and Tetrapoda s.s. We have expanded this study at the family level for Antiarcha, and special attention to the genus *Bothriolepis* is emphasised at each level (Figs 16 and 17). Pie-diagrams corresponding to relative proportions for each group are then plotted on a palaeomap of the Givetian [38]. Relative ages of localities (early, middle or late Givetian) are indicated by the relative distance of the pies from the edge of the map.

**A brief characterisation of Chinese, Gondwanan and Euramerican Givetian vertebrate assemblages.**    A few immediate distinctions can be made between Gondwanan, Euramerican and Chinese faunal assemblages during the Givetian period. Gondwanan localities exhibit the most important proportion of *Bothriolepis* species compared to all other taxa (at least 20%; Mount Howitt, Victoria, Antarctica and Aztec, Antarctica; Fig 16; see S5 Table). Besides, until our current study, *Bothriolepis* was an exclusively Gondwanan and Chinese taxon for early and middle Givetian (Aztec, Mount Howitt, Haikou; Fig 16; see also other localities in S5 Fig in S1 File). The Aztec Siltstone of Antarctica is by far the richest formation in term of species number of *Bothriolepis* (9 species; Fig 17) and also contains the Rhizodontida *Aztecia* [138]; it should however be reminded that all species were not strictly contemporaneous nor from the same exact locality.

**Valentia Slate macro-vertebrate assemblage.**    Because of the important palaeogeographic significance of this taxon, an update of the Valentia Slate macrovertebrate assemblage originally given by Russell [4] should be given. Russell recognised the presence of two taxa: 1) *Bothriolepis* sp. (Placodermi, Antiarcha, diverse anatomy and the taxon yielding by far the most abundant specimens; [4]:figs 3–6, 11), and 2) one indeterminate Acanthodii (fin spine; [4]:fig. 9). *Sauripterus* (Sarcopterygii, dermal scales; [4]:figs 7–8) come from the overlying Frasnian St Finan's Formation and belong in fact to *Holoptychius*; these are not taken further into consideration in this article.

A complete review of the material published by Russell reveals that the supposed acanthodian fin spine is actually a rhizodont coronoid fang (NHM P 59686; [4]:fig. 9), probably closely related to the Gondwanan genus *Gooloogongia* [138]. A dipnoan tooth plate is also identified in the matrix of the previous specimen (both identifications were made possible by CT-scanning the specimen at NHM). Both these taxa belong to the Valentia Slate formation (locality FB1 in [4]:fig. 2), and will be the subject of a separate publication.

The updated macro-vertebrate assemblage of the Givetian Valentia Slate is as follows: 1) *Bothriolepis dairbhrensis* sp. nov., 2) Rhizodontida indet., 3) Dipnoi indet., and 4) Tetrapoda indet. (ichnofossils only so far; Fig 16).

**Comparison: Valentia slate formation fauna vs. the rest of the Givetian world.**    It is noteworthy that the macrovertebrate faunal list of Valentia is very reduced compared to those of other localities listed by Sallan and Coates [96], and it would have certainly not been considered by these authors in their study ("Only macrofossil sites containing at least five named species in three groups in two divisions were eligible for inclusion in this smaller dataset"; see [96]:sup. inf. p. 9).

But despite this overall lower diversity, the presence of the antiarch *Bothriolepis* and of a rhizondont in the middle Givetian Valentia Slate is unique enough in Givetian Euramerica to invite for a comparison between those different localities.

The Valentia Slate Formation assemblage reminds the situation in the Mount Howitt and Aztec Siltstone faunas (these three localities hold the highest proportions of *Bothriolepis* compared to all other tatxa or only Antiarcha: *Bothriolepis* in Valentia corresponds to $25\%_{total}$, $100\%_{Antiarcha}$; Aztec Siltstones has $32.9\%_{total}$ and $92\%_{Antiarcha}$; *Bothriolepis* in Mount Howit

corresponds to $20\%_{total}$ and $75\%_{Antiarcha}$; by comparison, *Bothriolepis* composes $9.1\%_{total}$ and $25\%A_{ntiarcha}$ in Haikou, and of course $0\%_{total/Antiarcha}$ in Givetian Euramerican localities; Figs 16 and 17). It should be noted that the Aztec Siltstone covers six biozones, possibly ranging from the Eifelian to early Frasnian; comparisons should hence remain careful with this locality (J. A. L., pers. obs.). The presence of *Bothriolepis* in the Euramerican Valentia Slate of Ireland is thus very reminiscent of a Gondwanan fauna. This is emphasised with the presence of an indeterminate rhizodont in Valentia Slate, recalling the occurrence of *Aztecia* in the Givetian of Antarctica and of *Gooloogongia* in the Famennian of Canowindra, New South Wales, Australia; [138, 139]), while Rhizodontida were hitherto unknown in Euramerica.

Likewise, the faunal composition of the Valentia Slate Formation differs strikingly from that of sub-contemporaneous Euramerican localities. *Bothriolepis* is the only placoderm encountered in Valentia, and whereas this genus is absent in other localities. Beyond a possible collection bias, it should be noted that none of the hundreds of collected specimens could be attributed to other groups of placoderms. The contrast is even sharper with the localities on western side of the Caledonian–Appalachian Mountains, which are devoid of antiarchs and actinopterygians, but with over 50% of placoderms, a feature shared with Gerolstein in Germany, but also Haikou in the South China Palaeoblock and the Aztec Siltstones.

The fauna in the Valentia Slate Formation also differs from other well-known Givetian localities as it lacks (so far) both actinopterygians and acanthodians, a situation also encountered in Haikou and Cedar Valley, while Gerolstein and Rockport, although lacking actinopterygians, possess acanthodians.

The early Givetian Orcadian Lake in Scotland, U.K., and Harma in Estonia as well as the late Givetian Abava and Gauja in Latvia and Estonia share a likelihood in the faunal composition by possessing a small amount of actinopterygians and chondrichthyans, between 20% and 50% of acanthodians or sarcopterygians.

The strong Gondwanan faunal affinity of the Valentia slate contrasts with its Euramerican location as well as its early Givetian stratigraphic occurrence. We thus hypothesise that a cluster of Gondwanan vertebrates dispersed northward as soon as the early Givetian in Euramerica. It is noteworthy that not all taxa of this cluster remained in Euramerica and dispersed throughout the northern supercontinent. Indeed, our phylogenetic analysis seems to point toward a common ancestor of the Irish Bothriolepis with *B. gigantea* from Scotland, and further Euramerican occurrences of *Bothriolepis* are related to later dispersal waves.

*Bothriolepis* was likely a poor demersal swimmer, unable to cross large oceanic masses. It was likely restricted to freshwater environments and continental platform (following Young and Moody's assumptions of dispersal processes of *Bothriolepis*; [103]:163). Besides, the Gondwanan terrane was likely already closer to Euramerica than previously suggested. Such early connections between the two super continents may not be the most ancient, as evidenced by other earlier vertebrate and invertebrate proxies (for a Lochkovian proximity between the two supercontinents and their related terranes, see [140–142]). Besides, it is worth mentioning that Scotese figured a land connection in the West between Euramerica and Gondwana (Figs 16 and 17; [38]).

## Conclusions and further directions

A new species of the antiarch placoderm *Bothriolepis*, *B. dairbhrensis* sp. nov., is described from the early Givetian of the Valentia Slate formation in the Iveragh Peninsula, Republic of Ireland. Despite the green schist facies metamorphism, some specimens are well preserved enough to provide confident anatomical and hence taxonomic attribution. This also invites for further prospections in this area in order to discover new elements of the palaeoenvironment,

heavily dominated by this placoderm; of particular interest are the sarcopterygian remains indicating a unique faunal assemblage for the early Middle Devonian of Euramerica.

A series of computerised phylogenetic analyses are performed for the first time in order to resolve the relationships between the best known species of the genus *Bothriolepis*: These are the pretext for a critical review of the previous phylogenetic hypotheses and classifications, together with an objective appraisal of the morphological characters sustaining these hypotheses, as well as the validity of a genus particularly speciose (about one hundred species). The different tests performed in order to assess the confidence of our own hypotheses (i.e. various outgroups, successive weighting, Bremer decay) call for a deeper handling of the problem, beyond the scope of the current article.

The stratigraphic occurrences and the updated taxonomic assemblage of the Valentia Slate formation inform how we frame our study of the new *Bothriolepis* material. The most significant results from this work are the phylogenetic relationships of the various *Bothriolepis* species and its associated biogeographic implications for vertebrate dispersal episodes. The new species *B. dairbhrensis* is sister group of *B. gigantea* from the late Frasnian of Scotland, and corresponds to a first northward dispersal wave from Gondwana. This scenario does not preclude a Chinese origin for the genus *Bothriolepis*, but rather invites for more investigations in this direction. Other Euramerican *Bothriolepis* species are not related to the clade (*B. dairbhrensis* sp. nov., *B.gigantea*), but are related to later dispersal waves.

A deeper review of the different species and associated material of the genus *Bothriolepis* is required. Such intensive work has started, notably in the Baltic states by E. Lukševičs or Eastern Gondwana (Australia and Antarctica) by Z. Johanson, J. Long and G. Young. Further revision needs to be done on Euramerican (especially Old Red Sandstone) and Chinese material. It is however expected that the diversification of *Bothriolepis* will remain blur based on the extraordinary adaptive and opportunistic nature of the genus, coupled with likely interbreeding and evolutionary convergences.

## Supporting information

**S1 Text.**
(DOCX)

**S1 File.**
(DOCX)

**S1 Table. Anatomical abbreviations.**
(XLSX)

**S2 Table. Scan settings.**
(XLSX)

**S3 Table. Data matrix completude.**
(XLSX)

**S4 Table. Taxa and characters consistency and retention indices.**
(XLSX)

**S5 Table. Givetian macrovertebrate assemblages; list updated from Sallan and Coates, 2010 [96].**
(XLSX)

**S6 Table. Givetian antiarch assemblages; list updated from Sallan and Coates, 2010 [96].**
(XLSX)

## Acknowledgments

K.H. participated in the first field missions in the 1970s and later. P.E.A., H.M.B., M. Q, G. N., and I.S. participated in the field missions in 2018; V.D. joined the team in 2019. The authors thank P. Szrek and W. Lewenstam for their help on the field, as well as the reviewers of this article.

Fossils were scanned at Oslo NHM by Øyvind Hammer, and at the NHM of London, U.K. by Vincent Fernandez. Emma Bernard granted access to the collections of the NHM, London. Nélia Castro, Oslo NHM and University, performed the elemental analysis. Christian Obrist prepared part of the material physically, under Iwan Stossel's supervision. Preliminary CT-scanning experiments were performed with the help of Caroline Öhman Mägi (Materials in Medicine, Division of Applied Materials Science, Department of Engineering Sciences, Ångström Laboratory, Uppsala University, Sweden).

Jake Leyhr lent his camera for photographing the specimens in London, Ireland and Switzerland; Jan-Ove R. Ebbestad let V.D. access the binocular at the Evolution Museum (Uppsala University, Uppsala, Sweden) to photograph the small ADL NMING:F35210.

V.D. thanks the whole Stössel family for their warm welcome during his stay in Schaffhausen, Emma Bernard, Zerina Johanson and Vincent Fernandez (Natural History Museum, London, U.K.) for their welcome in and access to the collections and facilities, Patrick Wyse Jackson and Una Farrell (Trinity College Museum, Dublin, Ireland) for granting access to the collection and loan of some specimens, Nigel Monaghan and Patrick Roycroft (National Museum of Ireland–Natural History) for managing numbers and curation of new specimens, Sifra Bijl (EBC) for the use of her workstation for computing the Bremer indices on Mesquite, and Sébastien P. Olive and Ervins Lukševičs for fruitful discussions about *Bothriolepis*.

## Author Contributions

**Conceptualization:** Vincent Dupret, Nélia Castro, Kenneth T. Higgs, Iwan Stössel, Per E. Ahlberg.

**Data curation:** Vincent Dupret.

**Formal analysis:** Vincent Dupret, Nélia Castro, Øyvind Hammer, Iwan Stössel.

**Funding acquisition:** Kenneth T. Higgs, Iwan Stössel, Per E. Ahlberg.

**Investigation:** Vincent Dupret, Kenneth T. Higgs, Iwan Stössel.

**Methodology:** Vincent Dupret, Nélia Castro, Øyvind Hammer.

**Project administration:** Vincent Dupret, Hannah M. Byrne, Martin Qvarnström, Iwan Stössel, Per E. Ahlberg.

**Resources:** Vincent Dupret, Hannah M. Byrne, Kenneth T. Higgs, Grzegorz Niedźwiedzki, Martin Qvarnström, Iwan Stössel, Per E. Ahlberg.

**Software:** Vincent Dupret, Per E. Ahlberg.

**Supervision:** Vincent Dupret, Per E. Ahlberg.

**Validation:** Vincent Dupret, Hannah M. Byrne, Nélia Castro, Øyvind Hammer, Johan A. Long, Grzegorz Niedźwiedzki, Iwan Stössel, Per E. Ahlberg.

**Visualization:** Vincent Dupret, Øyvind Hammer, Johan A. Long, Per E. Ahlberg.

**Writing – original draft:** Vincent Dupret, Johan A. Long.

**Writing – review & editing:** Vincent Dupret, Hannah M. Byrne, Kenneth T. Higgs, Johan A. Long, Grzegorz Niedźwiedzki, Martin Qvarnström, Iwan Stössel, Per E. Ahlberg.

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
