## [Decision Letter · Decision Letter 0]

14 Nov 2022

PONE-D-22-25034The Bothriolepis (Placodermi, Antiarcha) material from the Valentia Slate Formation of the Iveragh Peninsula (middle Givetian, Ireland): morphology, evolutionary and systematic considerations, phylogenetic and palaeogeographic implicationsPLOS ONE

Dear Dr. Dupret,

Thank you for submitting your manuscript to PLOS ONE. After careful consideration, we feel that it has merit but does not fully meet PLOS ONE’s publication criteria as it currently stands. Therefore, we invite you to submit a revised version of the manuscript that addresses the points raised during the review process.

We look forward to receiving your revised manuscript.

Kind regards,

Jinzhuang Xue

Academic Editor

PLOS ONE

Journal Requirements:

2. Please take this opportunity to be sure you have met all of our guidelines for new species. For proper registration of a new zoological taxon, we require two specific statements to be included in your manuscript.

a.           In the Results section, the globally unique identifier (GUID), currently in the form of a Life Science Identifier (LSID), should be listed under the new species name, for example:

Anochetus boltoni Fisher sp. nov. urn:lsid:zoobank.org:act:B6C072CF-1CA6-40C7-8396-534E91EF7FBB

Another LSID for the manuscript itself should also appear within the Nomenclature statement. You will need to contact Zoobank (zoobank.org/About) to obtain a GUID (LSID). You should receive one LSID for your manuscript and a separate, unique LSID for the new species.

b.           Please also insert the following text into the Methods section, in a sub-section to be called "Nomenclatural Acts":

The electronic edition of this article conforms to the requirements of the amended International Code of Zoological Nomenclature, and hence the new names contained herein are available under that Code from the electronic edition of this article. This published work and the nomenclatural acts it contains have been registered in ZooBank, the online registration system for the ICZN. The ZooBank LSIDs (Life Science Identifiers) can be resolved and the associated information viewed through any standard web browser by appending the LSID to the prefix "" ext-link-type="uri" xlink:type="simple">http://zoobank.org/". The LSID for this publication is: urn:lsid:zoobank.org:pub: XXXXXXX. The electronic edition of this work was published in a journal with an ISSN, and has been archived and is available from the following digital repositories: PubMed Central, LOCKSS [author to insert any additional repositories].

All PLOS ONE articles are deposited in PubMed Central and LOCKSS. If your institute, or those of your co-authors, has its own repository, we recommend that you also deposit the published online article there and include the name in your article.

Following a recent ruling by the International Commission on Zoological Nomenclature, electronic journals are now a valid format for publication of new zoological taxa. In order to ensure the valid publication of your new species, please be sure to include the updated version of Nomenclatural Acts (above). A complete explanation of our guidelines for publishing new species can be found on our website: http://www.plosone.org/static/guidelines#zoological.

“This work is funded by a Wallenberg Scholarship from the Knut and Alice Wallenberg Foundation and an ERC Advanced Grant (ERC-2020-ADG 101019613 "Tetrapod Origin"), both awarded to P.E.A. The physical preparation of the material in Switzerland was personally funded by K.H”

 “This work is funded by a Wallenberg Scholarship from the Knut and Alice Wallenberg Foundation and an ERC Advanced Grant (ERC-2020-ADG 101019613 "Tetrapod Origin"), both awarded to P.E.A.

6. We note that Figures 1, 15, 16 and 17 in your submission contain map images which may be copyrighted. All PLOS content is published under the Creative Commons Attribution License (CC BY 4.0), which means that the manuscript, images, and Supporting Information files will be freely available online, and any third party is permitted to access, download, copy, distribute, and use these materials in any way, even commercially, with proper attribution. For these reasons, we cannot publish previously copyrighted maps or satellite images created using proprietary data, such as Google software (Google Maps, Street View, and Earth). For more information, see our copyright guidelines: http://journals.plos.org/plosone/s/licenses-and-copyright.

 a. You may seek permission from the original copyright holder of Figures 1, 15, 16 and 17 to publish the content specifically under the CC BY 4.0 license. 

Additional Editor Comments:

I attach below another review report finished by Dr. Philippe Janvier (this report is informal, but very helpful, since it was not submitted through the website, but instead, was sent to us by email.).

" PONE-D-22-25034-This manuscript describes a new fauna of antiarchan placoderm fishes from the Givetian Valentia Slate Formation of Ireland, and includes an extensive review of the phylogenetic relationships of all the antiarchs known to date, as well as a new analysis of their biogeographical history. The authors are very brave to have tackled these very difficult questions, which once caused me to have many headaches! As a whole, I find this manuscript a remarkable piece, which will be extremely useful to all palaeontologists who happen to find antiarch-bearing (especially Bothriolepis-bearing) outcrops. Bothriolepids were swarming all over the world in the Devonian and I have long suspected that resolving their relationships would once become the leading thread of Devonian paleobiogeography. This is what is achieved here. Therefore, I very strongly recommend the publication of this paper." By Dr. Philippe Janvier

Reviewers' comments:

Reviewer's Responses to Questions

**Comments to the Author**

1. Is the manuscript technically sound, and do the data support the conclusions?

Reviewer #1: Yes

2. Has the statistical analysis been performed appropriately and rigorously? 

Reviewer #1: N/A

3. Have the authors made all data underlying the findings in their manuscript fully available?

Reviewer #1: Yes

4. Is the manuscript presented in an intelligible fashion and written in standard English?

Reviewer #1: Yes

5. Review Comments to the Author

Reviewer #1: This research is a thorough investigation on the status of all Bothriolepis species known to date.

The study is centered on a new species of the antiarch placoderm Bothriolepis , found in Ireland. The material , preserved in a tectonised matrix needed to be explored for some crucial specimens with new methods (e.g. Xray scanning). Without the use of such up-to-date technology, it would have been impossible to produce such a thorough study of this material which is also the oldest representative of the genus Bothriolepis in Euramerica.

The systematic description is clear and extensive. It fits with the best tradition with accurate comparisons with the other species representative of the genus . It was necessary in order to base a larger study oriented towards the phylogenetic relationships of all known Bothriolepids . This was a necessary step to compare the new result with previous proposals published by various authors and to discuss the evolutionary story of the diverse species through time and space during the Lower, Middle and Upper Devonian period worldwide.

The extensive study of the relationships between all known Bothriolepids, is done in a magistral way. Bothriolepis is a very difficult matter to study because the material is quite diverse and preserved in various ways often represented by scattered elements. As often with complex fossils some characters used to investigate the phylogenetic relationships are sometime missing and this is a point that explains all the multiple analyses illustrated in the supplementary information . This exploratory part is a very important source of knowledge . It is an innovation relative to preceding published phylogenetic results without the use of extensive computed methodology.

This contribution is an important one and I recommend its publication.

I have some remarks and details about the text ,but it can be done during the revision before publication .

- some formal latinized names are not in italic style.

- Gondwana is not an adjective . It should be "gondwanan"

- Fig 4 A and C are too dark due to the colors that should be lighter in order to avoid the blurring of surface anatomical details of the bones.

-some references bibliographic list are forgotten in text : eg (p.87) Janvier Villaroel ref[105] ; same for Young Moody [ref 103]

6. PLOS authors have the option to publish the peer review history of their article (what does this mean?). If published, this will include your full peer review and any attached files.

Reviewer #1: No

---

## [Author Response · Author response to Decision Letter 0]

15 Dec 2022

[the text below is an excerpt of the "answers to reviewers" letter attached in the submission]:

ce list is correct and up to date. 

9. Answers to reviewer 1:

9.1. About some formal Latin names that were not italicised: This is related to WORD automated format for titles. In agreement with the Editor (email 2022-11-28), these mistakes will be managed by the editor himself from the original WORD document. 

9.2. “Gondwana” vs. “Gondwanan” occurrences are corrected. 

9.3. Fig. 4A,C were modified to appear lighter (as per reviewer and editor’s wish – email 2022-11-28). 

9.4. The references mentioned by the reviewer as missing were actually cited in the text and listed in the References section.

---

## [Editor Report · Decision Letter 1]

22 Dec 2022

The Bothriolepis (Placodermi, Antiarcha) material from the Valentia Slate Formation of the Iveragh Peninsula (middle Givetian, Ireland): morphology, evolutionary and systematic considerations, phylogenetic and palaeogeographic implications

PONE-D-22-25034R1

Dear Dr. Dupret,

We're pleased to inform you that your manuscript has been judged scientifically suitable for publication and will be formally accepted for publication once it meets all outstanding technical requirements.

Kind regards,

Jinzhuang Xue

Academic Editor

PLOS ONE

Additional Editor Comments (optional):

The authors have made revisions according to the suggestions raised by the reviewers and the editor. And the manuscript is now acceptable for publication. Please note there is a small mistake: on page 16, there is figure legend of Fig. 2; but I believe it is in a wrong place. This error can be resolved when correcting the proof. This paper will be a very valuable contribution to the filed. Congratulations to the authors.
---

## [Editor Report · Acceptance letter]

5 Jan 2023

PONE-D-22-25034R1 

The *Bothriolepis* (Placodermi, Antiarcha) material from the Valentia Slate Formation of the Iveragh Peninsula (middle Givetian, Ireland): morphology, evolutionary and systematic considerations, phylogenetic and palaeogeographic implications 

Dear Dr. Dupret:

I'm pleased to inform you that your manuscript has been deemed suitable for publication in PLOS ONE. Congratulations! Your manuscript is now with our production department. 

Kind regards, 

on behalf of

Dr. Jinzhuang Xue 

Academic Editor

PLOS ONE